# Alignment-Sensitive Minimax Rates for Spectral Algorithms with Learned Kernels

## Abstract

We study spectral algorithms in the setting where kernels are learned from data. We introduce the effective span dimension (ESD), an alignment-sensitive complexity measure that depends jointly on the signal, spectrum, and noise level $\sigma^2$. The ESD is well-defined for arbitrary kernels and signals without requiring eigen-decay conditions or source conditions. We prove that for sequence models whose ESD is at most $K$, the minimax excess risk scales as $\sigma^2 K$. Furthermore, we analyze over-parameterized gradient flow and prove that it can reduce the ESD of a sequence model, which in turn moves the problem into an easier ESD class and lowers the corresponding minimax risk. This analysis suggests a general route to study how adaptive feature learning can improve generalization through signal-kernel alignment: adaptive learning procedures reshape the kernel so that the ESD decreases and the problem enters an easier ESD class. We also extend the ESD framework to linear models and RKHS regression, and we support the theory with numerical experiments. This framework provides a novel perspective on generalization beyond traditional fixed-kernel theories.

## 1 Introduction

Neural networks excel across many applications, yet a complete theoretical understanding of their efficiency remains an open problem. In the infinite-width limit, the Neural Tangent Kernel (NTK) theory approximates training dynamics as kernel regression (Jacot et al., 2018; Allen-Zhu et al., 2019), and it enables the study of generalization by leveraging the classical theory of kernel regression and Reproducing Kernel Hilbert Spaces (RKHS) (Bauer et al., 2007; Yao et al., 2007). However, the NTK theory does not explain why finite-width networks, which adapt their features during training, often outperform traditional methods (Ghorbani et al., 2020; Gatmiry et al., 2021; Karp et al., 2021; Shi et al., 2023; Wenger et al., 2023; Seleznova & Kutyniok, 2022).

A growing line of work directly studies adaptivity, i.e., learning representations or kernel properties during training (Ba et al., 2022; Kunin et al., 2024; Liu et al., 2024; Bordelon et al., 2025; Xu & Ziyin, 2025; Zhang et al., 2024). Simplified models show that learning *eigenvalues* (with eigenfunctions fixed) can align the kernel with the signal and improve performance (Li & Lin, 2024; 2025). The common thread is signal-kernel alignment: ***performance improves when the target's energy concentrates on leading eigenfunctions***. The importance of signal-kernel alignment is well-recognized in the literature (Arora et al., 2019; Woodworth et al., 2020; Kornblith et al., 2019; Radhakrishnan et al., 2024). In classical RKHS theory, signal-kernel alignment is often captured through ***source conditions*** (Engl et al., 1996; Caponnetto & De Vito, 2007). While powerful for static kernels, source conditions assume a fixed eigenbasis and become ill-defined if the kernel evolves over time as in adaptive learning. To explain the observed advantages of adaptive learning, we need a refined theoretical framework that goes beyond fixed-kernel assumptions.

In this paper, we propose the **Effective Span Dimension (ESD)**, which is a population complexity measure for the analysis of signal-kernel alignment. ESD counts the smallest number of leading eigenfunctions required so that the remaining signal energy matches the estimation variance. Unlike classical measures that ignore the signal, ESD depends on the *signal, spectrum, and noise level*. Our framework provides new theoretical insights that are absent in classical analyses. In particular, we achieve the following:

(i) We establish a sharp minimax optimal convergence rate using ESD, which not only subsumes classical rates but also extends to dynamic kernels where classical theory is silent.

(ii) We provide a mechanistic explanation for how adaptive algorithms improve generalization through ESD reduction: they modify the induced kernel to better align with the signal, shrink the ESD, and thereby move the problem into a class with lower minimax risk. This mechanism is proved for over-parameterized gradient flow and observed empirically in deep linear networks.

(iii) We extend our definitions and theory from sequence models to linear regression and kernel regression, which demonstrates the broad applicability of our framework.

Our ESD framework bridges fixed-kernel theory and adaptive learning by quantifying signal-kernel alignment. We hope it will open avenues for deeper understanding of neural networks and novel adaptive algorithms.

*Notations.* Write $a \lesssim b$ if there exists a constant $C > 0$ such that $a \leq Cb$, and write $a \asymp b$ if $a \lesssim b$ and $b \lesssim a$, where the dependence of the constants on other parameters is determined by the context. For $d \in \mathbb{N}_+$, let $[d] = \{1, 2, \ldots, d\}$; for $d = \infty$, let $[d] = \mathbb{N}_+$. $\mathbf{1}_{\{\cdot\}}$ denotes an indicator function.

## 2 BACKGROUND ON KERNEL METHODS

We first review kernel regression to provide context. Let $(\boldsymbol{x}_i, y_i)_{i=1}^n$ be i.i.d. samples from $y = f^*(\boldsymbol{x}) + \epsilon$, where $\boldsymbol{x} \sim \mu$ on a compact space $\mathcal{X}$, $\epsilon$ is an independent noise with $\mathbb{E}[\epsilon] = 0, \mathrm{Var}(\epsilon) = \sigma_0^2$. For an estimator $\hat{f}$ of the target function $f^*$, the excess risk is $\mathcal{R}(\hat{f}; f^*) = \mathbb{E}_{\boldsymbol{x} \sim \mu}\left[(\hat{f}(\boldsymbol{x}) - f^*(\boldsymbol{x}))^2\right]$.

A symmetric, positive-definite, and continuous kernel $\mathbf{k}(\cdot, \cdot) : \mathcal{X} \times \mathcal{X} \to \mathbb{R}$ induces an RKHS $\mathcal{H} \subset L^2(\mathcal{X}, \mu)$ with inner product $\langle \cdot, \cdot \rangle_{\mathcal{H}}$ and norm $\|\cdot\|_{\mathcal{H}}$ (Wahba, 1990; Schölkopf & Smola, 2002). Assuming $\mathbf{k}$ is bounded, Mercer's theorem yields

$$\mathbf{k}(\boldsymbol{x}, \boldsymbol{x}') = \sum_{j=1}^{\infty} \lambda_j \psi_j(\boldsymbol{x}) \psi_j(\boldsymbol{x}'), \quad \boldsymbol{x}, \boldsymbol{x}' \in \mathcal{X}, \tag{1}$$

where $\{\lambda_j\}_{j \geq 1}$ are eigenvalues and $\{\psi_j\}_{j \geq 1} \subset \mathcal{H}$ are eigenfunctions forming an orthonormal basis of $L^2(\mathcal{X}, \mu)$. For background, see Steinwart & Christmann (2008); Steinwart & Scovel (2012).

Kernel regression estimates $f^*$ using $f = \sum_j \beta_j \psi_j$ and regularizes via a filter of the kernel spectrum $\{\lambda_j\}$(Rosasco et al., 2005; Caponnetto & Vito, 2007; Gerfo et al., 2008). If $f^*$ satisfies the *Hölder source condition* $\sum_j \langle f^*, \psi_j \rangle^2 / \lambda_j^s \leq R_s$ for some positive constants $s$ and $R_s$ (Engl et al., 1996; Mathé & Pereverzev, 2003) and the spectrum decays polynomially $\lambda_j \asymp j^{-\gamma}$, then the minimax rate is $n^{-s\gamma/(s\gamma+1)}$ (Yao et al., 2007; Li et al., 2024; Wang et al., 2024). The choice of kernels can significantly affect the performance (Li & Lin, 2024; Zhang et al., 2024), so it is beneficial when the kernel eigenvalues align well with the expansion of the target function.

Since the kernel is usually chosen without knowing $f^*$, fixed-kernel methods may encounter misalignment. To address this limitation, adaptive methods have recently emerged. For instance, Li & Lin (2025) propose adapting kernel eigenvalues while fixing eigenfunctions. Specifically, they consider the kernel $\mathbf{k}_{\boldsymbol{a}}(x, x') = \sum_{j \geq 1} a_j^2 \psi_j(x) \psi_j(x')$ indexed by $\boldsymbol{a} = (a_j)_{j \geq 1}$ and the candidate $f = \sum_{j \geq 1} \beta_j a_j \psi_j$, where $a_j$'s and $\beta_j$'s are learned jointly via gradient flow. Such adaptation often improves performances, yet classical analyses built on fixed spectral assumptions do not explain these gains, because (a) adapted eigenvalues typically deviate from standard eigenvalue decay assumptions, and (b) it is unclear whether the classical source condition holds with respect to the adapted kernel, and if so, what the value of $s$ is. We therefore seek a refined theoretical framework that explicitly captures signal-kernel alignment and explain the gains achieved by kernel adaptation.

**Bridge to the sequence model.** We next connect the RKHS regression with the sequence model to motivate our analysis in the next section. For any $j \in \mathbb{N}_+$, define

$$\theta_j^* = \langle f^*, \psi_j \rangle, \quad z_j = n^{-1} \sum_i y_i \psi_j(\boldsymbol{x}_i), \text{ and } \xi_j = n^{-1} \sum_i \epsilon_i \psi_j(\boldsymbol{x}_i). \tag{2}$$

For large $n$, we have $n^{-1} \sum_i \psi_j(\boldsymbol{x}_i)\psi_k(\boldsymbol{x}_i) \approx \mathbb{E}[\psi_j(\boldsymbol{x})\psi_k(\boldsymbol{x})] = \mathbf{1}_{\{j=k\}}$, which implies that

$$z_j \approx \theta_j^* + \xi_j, \text{ and } \mathbb{E}[\xi_j] = 0, \mathrm{Cov}(\xi_j, \xi_k) \approx n^{-1}\sigma_0^2 \mathbf{1}_{\{j=k\}}, \quad \forall j, k \in \mathbb{N}_+. \tag{3}$$

This reduction connects RKHS regression to a sequence model where the observations are $z_j = \theta_j^* + \xi_j$ and the noise terms $\{\xi_j\}$ are uncorrelated with effective noise variance $\sigma_{\text{eff}}^2 := n^{-1}\sigma_0^2$. The error in the approximation due to finite $n$ will inflate the estimation variance compared to the sequence model. This approximation error can be controlled if $f^*$ is bounded; see Appendix B for a rigorous treatment.

## 3 Effective Span Dimension and Span Profile

To bridge existing theory and adaptive kernel methods as discussed in Section 2, we propose a novel framework to characterize the alignment between spectrum and signal. To focus on the main idea, we use the reduction in Equation (3) and first present our framework using sequence models.

**Sequence models.** A sequence model assumes observations are sampled as follows:

$$z_j = \theta_j^* + \xi_j, \quad 1 \le j \le d, \tag{4}$$

where $d \in \{\infty\} \cup \mathbb{N}_+$, $\boldsymbol{\theta}^* = (\theta_j^*)_{j=1}^d$ is a sequence of unknown parameters, $\xi_j$'s are uncorrelated random variables with mean zero and variance $\sigma^2$ (the noise level). For an estimator $\widehat{\boldsymbol{\theta}} = (\widehat{\theta}_j)_{j=1}^d$, we consider the loss $\mathcal{L}(\widehat{\boldsymbol{\theta}}; \boldsymbol{\theta}^*) = \sum_{j=1}^d (\widehat{\theta}_j - \theta_j^*)^2$ and risk $\mathcal{R}(\widehat{\boldsymbol{\theta}}; \boldsymbol{\theta}^*) = \mathbb{E}\mathcal{L}(\widehat{\boldsymbol{\theta}}; \boldsymbol{\theta}^*)$. The sequence model captures core estimation phenomena while permitting explicit analysis (Brown et al., 2002; Johnstone, 2017). In Appendix A, we use whitening to deal with correlated noise and analyze fixed-design linear regression. In Appendix B, we leverage the approximation in Equation (3) to analyze RKHS regression and random-design linear regression.

**Spectral estimators.** Given eigenvalues $\boldsymbol{\lambda} = (\lambda_j)_{j=1}^d$, spectral estimators take the form $\widehat{\theta}_j = (1 - g_\nu(\lambda_j)) z_j$, where $g_\nu(\lambda)$ is a filter such that larger $\nu$ induces more shrinkage. Some examples are:

$$\text{Ridge (R):} \quad g_\nu^{\text{R}}(\lambda) = \frac{1}{1 + \lambda/\nu}, \qquad \widehat{\theta}_j^{\text{R},\nu} = \frac{\lambda_j}{\lambda_j + \nu} z_j. \tag{5}$$

$$\text{Gradient Flow (GF):} \quad g_\nu^{\text{GF}}(\lambda) = e^{-\lambda/\nu}, \qquad \widehat{\theta}_j^{\text{GF},\nu} = \left(1 - e^{-\lambda_j/\nu}\right) z_j. \tag{6}$$

$$\text{Principal Component (PC):} \quad g_\nu^{\text{PC}}(\lambda) = \mathbf{1}_{\{\lambda < \nu\}}, \qquad \widehat{\theta}_j^{\text{PC},\nu} = \mathbf{1}_{\{\lambda_j \ge \nu\}} z_j. \tag{7}$$

For spectral estimators, the risk decomposes into squared bias $\sum_j (g_\nu(\lambda_j))^2 \theta_j^2$ and variance $\sum_j (1 - g_\nu(\lambda_j))^2 \sigma^2$, where $\nu$ controls the bias-variance trade-off. Classical analyses often assume $\boldsymbol{\theta}^*$ lies in an ellipsoid $\Theta_{\mathbf{a}} = \left\{\theta : \sum_j^\infty a_j^2 \theta_j^2 \le C^2\right\}$ and derives convergence rates for sequences with $a_i \asymp i^\alpha$ (Johnstone, 2017). Our theoretical framework aims to bypass these assumptions.

### 3.1 Effective Span Dimension

Our goal is to develop a measure that captures the interplay between signal structure $\boldsymbol{\theta}^*$, spectrum $\boldsymbol{\lambda}$, and noise variance $\sigma^2$. To start, we examine the Principal Component (PC) estimator analytically. PC operates by truncating coordinates with small eigenvalues. Its risk is composed of variance from the retained components and squared bias from those truncated. By trading variance against tail bias, PC admits the optimal truncation point. This motivates our core definition.

**Definition 3.1.** *Suppose $\{\lambda_j\}_{j \in [d]}$ are distinct, with $\pi_i$ indexing the $i$-th largest so that $\lambda_{\pi_1} > \lambda_{\pi_2} > \ldots$. We define the Effective Span Dimension (ESD) $d^\dagger$ of $\boldsymbol{\theta}^*$ w.r.t. the spectrum $\boldsymbol{\lambda}$ and variance $\sigma^2$ as*

$$d^\dagger = d^\dagger(\sigma^2; \boldsymbol{\theta}^*, \boldsymbol{\lambda}) = \min\{k \in [d] : \frac{1}{k} \sum_{i=k+1}^d (\theta_{\pi_i}^*)^2 \le \sigma^2\}.$$

Intuitively, the ESD $d^\dagger$ is the number of leading coordinates (with leading eigenvalues $\lambda_i$) that are most critical for estimation at a given noise level $\sigma^2$. It is the truncation point where the squared tail bias of the PC estimator first becomes comparable to (or less than) the estimation variance. The next theorem shows that $d^\dagger$ describes the best achievable risk for the PC estimator.

**Theorem 3.2** (Optimal PC Estimator Risk). *Let $\widehat{\boldsymbol{\theta}}^{\mathrm{PC},\nu}$ be the PC estimator for the sequence model in Equation* (4). *Denote by $\mathcal{R}_*^{\mathrm{PC}}$ the minimal possible risk over all choices of $\nu$. Let $d^\dagger = d^\dagger(\sigma^2; \boldsymbol{\theta}^*, \boldsymbol{\lambda})$ be the ESD of $\boldsymbol{\theta}^*$ w.r.t. the spectrum $\boldsymbol{\lambda}$ and the variance $\sigma^2$. It holds that*

$$(d^\dagger - 1)\,\sigma^2 \leq \mathcal{R}_*^{\mathrm{PC}} \;\leq\; 2\,d^\dagger\,\sigma^2.$$

The well-tuned PC estimator is known to be minimax rate optimal under classical assumptions in sequence models that are analogous to the polynomial eigen-decay condition and the source condition in kernel regression (see Propositions 3.11 and 4.23 of Johnstone (2017)). In contrast, Theorem 3.2 suggests that we can instead use $O(d^\dagger \sigma^2)$ to upper bound the minimax estimation error with no reliance on particular spectral decay or source conditions.

Although we motivate ESD via PC estimators, the definition of ESD itself is not tied to any specific estimator, just as the sparsity level is not tied exclusively to Lasso or best subset selection. Moreover, the following theorem confirms that $d^\dagger \sigma^2$ indeed characterizes the intrinsic difficulty of estimation.

**Theorem 3.3.** *For any $K \in [d]$, spectrum $\boldsymbol{\lambda} = \{\lambda_j\}_{j \in [d]}$, and variance $\sigma^2$, define*

$$\mathcal{F}_{K,\boldsymbol{\lambda}}^{(n)} = \left\{ \boldsymbol{\theta} \in \mathbb{R}^d : d^\dagger(\sigma^2; \boldsymbol{\theta}, \boldsymbol{\lambda}) \leq K \right\}. \tag{8}$$

*Suppose the sample $\boldsymbol{Z}$ is drawn from the sequence model in Equation* (4). *We have*

$$\inf_{\widehat{\boldsymbol{\theta}}} \sup_{\boldsymbol{\theta}^* \in \mathcal{F}_{K,\boldsymbol{\lambda}}^{(n)}} \mathcal{R}(\widehat{\boldsymbol{\theta}}, \boldsymbol{\theta}^*) \asymp K\sigma^2,$$

*where* $\inf$ *is taken over any estimator $\widehat{\boldsymbol{\theta}}$ based on $\boldsymbol{Z}$.*

Theorem 3.3 considers the minimax risk over $\mathcal{F}_{K,\boldsymbol{\lambda}}^{(n)}$, a class of distributions whose ESDs are at most $K$. We interpret $K$ as the quota for ESD: the larger $K$, the larger $\mathcal{F}_{K,\boldsymbol{\lambda}}^{(n)}$ and thus the higher the minimax risk. Theorem 3.3 highlights the usefulness of ESD: although we motivate the definition of ESD using the PC estimator, the minimax lower bound $K\sigma^2$ applies to all estimators and the ESD actually quantifies the best possible worst-case performance of *any estimator*.

We emphasize that the ESD is a population-level complexity measure for signal-spectrum alignment. It is not a tuning parameter for a specific algorithm, and it does not need to be estimated. Instead, it serves as a statistical complexity measure, playing a role analogous to sparsity levels in high-dimensional regression or smoothness indices in nonparametric statistics. While these quantities are unknown to the practitioner, they characterize the information-theoretic limits of the respective learning tasks. Estimating the ESD from data is an interesting problem on its own, and we defer it to future work because it goes beyond the goals in the current paper.

**Comparisons to other alignment measures.** Alternative alignment measures exist. The cosine similarity-based kernel-target alignment yields generalization bounds (Cortes et al., 2012; Cristianini et al., 2001), but these bounds are typically too loose to explain fast rates in adaptive kernel methods. Recently, Barzilai & Shamir (2023) extended benign-overfitting analyses (Bartlett et al., 2020b; Tsigler & Bartlett, 2023) to kernel ridge regression, which may encounter saturation effects that prevent optimal rates for overly smooth target functions.

**Comparisons to other effective dimensions.** There are some well-known measures used in the classical analysis of spectral methods. We discuss the differences between ESD and these measures.

Zhang (2005) introduces the effective dimension to quantify the complexity of any regularized method. For ridge regularization in Equation (5), the effective dimension is defined as $d_{\mathrm{eff}}(\nu) = \sum_j \frac{\lambda_j}{\lambda_j + \nu}$ (see Proposition A.1 in Zhang (2005)). $d_{\mathrm{eff}}(\nu)$ depends only on the spectrum $\boldsymbol{\lambda}$ and the regularization parameter $\nu$, but not on the signal $\boldsymbol{\theta}^*$ or the noise level $\sigma^2$. Consequently, the effective dimension is not suitable for measuring signal-spectrum alignment. Furthermore, the effective dimension, as a function of $\nu$, does not directly connect to any minimax risk.

In linear regression, Bartlett et al. (2020a) analyze the minimum-norm interpolator via the effective rank $r_k = \frac{\sum_{i>k} \lambda_{\pi_i}}{\lambda_{\pi_{k+1}}}$ (using the relationship in Equation (3)). They define the splitting index

$k^* = \min\{k \geq 0 : \sigma^2 r_k \geq b\}$ for some constant $b$ and establish risk bounds using $\sigma^2 k^*$. While $k^*$ may resemble ESD since both depend on $\boldsymbol{\lambda}$ and $\sigma^2$, they differ in two important aspects: (i) $k^*$ does not involve the signal and thus cannot measure signal-spectrum alignment; and (ii) $k^*$ is tailored to the minimum-norm estimator and does not characterize the minimax risk over a class.

Both $d_{\text{eff}}(\nu)$ and $k^*$ are **signal-agnostic**: they depend on the spectrum $\boldsymbol{\lambda}$ (and either $\nu$ or $\sigma^2$) only, and therefore remain invariant under any change in the alignment between the signal and the kernel's eigenfunctions. For instance, if adaptive training improves alignment by reordering the eigenfunctions to better align with the signal while preserving the set of eigenvalues, then both $d_{\text{eff}}(\nu)$ and $k^*$ are unchanged. In contrast, the ESD $d^\dagger(\sigma^2; \boldsymbol{\theta}^*, \boldsymbol{\lambda})$ is signal-aware, because it is defined by the bias-variance crossing for the specific $\boldsymbol{\theta}^*$. As signal-kernel alignment improves, the ESD decreases. Consequently, the ESD can mechanistically explain the generalization benefits of adaptive kernel learning, a phenomenon that signal-agnostic complexity measures like $d_{\text{eff}}(\nu)$ and $k^*$ cannot capture.

**Examples.** For the following canonical settings, the optimal PC risk satisfies

$$
\mathcal{R}_*^{\text{PC}} \asymp \begin{cases}
\min\left\{\sigma^{\frac{2s\beta}{1+s\beta}}, d\,\sigma^2\right\}, & \text{(1)}\ \lambda_i = i^{-\beta},\ \sum_{i=1}^d \lambda_i^{-s}\theta_i^{*2} \leq R,\ \beta, s > 0, \\
\min\left\{\sigma^{2-\frac{2}{\alpha}}, d\,\sigma^2\right\}, & \text{(2)}\ \theta_i^* = i^{-\alpha/2},\ \alpha > 1,\ \{\lambda_i\}\downarrow, \\
\min\left\{d\,\sigma^2, \log\left(d\sigma^2/\log\left(d\sigma^2\right)\right)\right\}, & \text{(3)}\ d < \infty,\ \theta_i^* = i^{-1/2},\ \{\lambda_i\}\downarrow, \\
d\min\left\{d^{-\alpha}, \sigma^2\right\}, & \text{(4)}\ d < \infty,\ 0 < \alpha < 1,\ \theta_i^* = i^{-\alpha/2},\ \{\lambda_i\}\downarrow,
\end{cases}
\tag{9}
$$

where $\{\lambda_i\}\downarrow$ means $\lambda_i$ is decreasing. Details and proofs are deferred to Appendix D.3.

In Setting (1), we may take $\sigma^2 = \sigma_0^2/n$ in view of Equation (3), and then the upper bound becomes $\sigma_0^2 \min\left(n^{-\frac{s\beta}{1+s\beta}}, d/n\right)$, which matches the well-known optimal rate under the source condition and the polynomial eigen-decay condition in the case when $d = \infty$. When $d < \infty$, there is a phase transition around $d_0 \asymp n^{\frac{1}{1+s\beta}}$: if $d \lesssim d_0$, the upper bound is $d\sigma_0^2/n$; if $d \gtrsim d_0$, the upper bound is the same as if $d = \infty$.

Appendix C.1 illustrates a sparse signal example where the ESD provides a quantitative comparison of two different spectra while the existing measures like $d_{\text{eff}}(\nu)$ and $k^*$ do not. These examples suggest that the notion of ESD allows us not only to recover classical results but also to explore new settings where the classical framework is inapplicable.

## 3.2 SPAN PROFILE

The definition of ESD explicitly depends on the noise level $\sigma^2$, which distinguishes it from other complexity measures in the literature. The dependence on $\sigma^2$ reflects the bias-variance trade-off nature of ESD: as $\sigma^2$ decreases, more coordinates can be unbiasedly estimated while controlling the overall variance, thereby more bias is removed. To focus on the alignment between a given signal $\boldsymbol{\theta}^*$ and a spectrum $\boldsymbol{\lambda}$, we examine the ESD by varying the noise level.

**Definition 3.4.** *We define the span profile of $\boldsymbol{\theta}^*$ w.r.t. the spectrum $\boldsymbol{\lambda}$ as $\mathbf{D}_{\boldsymbol{\theta}^*, \boldsymbol{\lambda}} : \tau \mapsto d^\dagger(\tau; \boldsymbol{\theta}^*, \boldsymbol{\lambda})$.*

The span profile $\mathbf{D}_{\boldsymbol{\theta}^*, \boldsymbol{\lambda}}$ is a well-defined object that depends only on $\boldsymbol{\theta}^*$ and the ordering of $\boldsymbol{\lambda}$, and it summarizes how $\sigma^2$ affects the ESD. Theorem 3.2 suggests that for two spectra $\boldsymbol{\lambda}^{(1)}$ and $\boldsymbol{\lambda}^{(2)}$, we can compare their alignments with the signal by the ratio of $r(\tau) = \mathbf{D}_{\boldsymbol{\theta}^*, \boldsymbol{\lambda}^{(1)}}(\tau)/\mathbf{D}_{\boldsymbol{\theta}^*, \boldsymbol{\lambda}^{(2)}}(\tau)$ for small $\tau$, because, if this ratio is very small (and in particular if the limit is 0 for $\tau \to 0$), then a kernel method using $\boldsymbol{\lambda}^{(1)}$ can achieve a smaller risk than one that uses $\boldsymbol{\lambda}^{(2)}$. Such comparisons are not as convenient in classical theory. See Appendix C for more illustrations.

A closely related object is the *trade-off function* of $\boldsymbol{\theta}^*$ relative to $\boldsymbol{\lambda}$, which is defined as

$$
\mathbf{H}_{\boldsymbol{\theta}^*, \boldsymbol{\lambda}}(k) = \frac{1}{k} \sum_{i=k+1}^d \left(\theta_{\pi_i}^*\right)^2 = \frac{1}{k} \sum_{i:\, \lambda_i < \lambda_{\pi_k}} \left(\theta_i^*\right)^2,\ k \in [d].
\tag{10}
$$

The quantity $\sigma^{-2}\mathbf{H}_{\boldsymbol{\theta}^*, \boldsymbol{\lambda}}(k)$ equals the bias-variance ratio of the PC estimator using the $k$ leading coordinates. Properties of span profiles and trade-off functions are summarized as follows.

**Proposition 3.5.** *(1) Both $\mathbf{D}_{\boldsymbol{\theta}^*,\boldsymbol{\lambda}} : \tau \mapsto [d]$ and $\mathbf{H}_{\boldsymbol{\theta}^*,\boldsymbol{\lambda}} : [d] \mapsto [0,\infty)$ are nonincreasing. (2) For any $\tau$, it holds that $\mathbf{D}_{\boldsymbol{\theta}^*,\boldsymbol{\lambda}}(\tau) = \min\{k \in [d] : \mathbf{H}_{\boldsymbol{\theta}^*,\boldsymbol{\lambda}}(k) \leq \tau\}$. (3) For two spectra $\boldsymbol{\lambda}^{(1)}$ and $\boldsymbol{\lambda}^{(2)}$, if $\mathbf{H}_{\boldsymbol{\theta}^*,\boldsymbol{\lambda}^{(1)}}(k) \leq \mathbf{H}_{\boldsymbol{\theta}^*,\boldsymbol{\lambda}^{(2)}}(k)$ for all $k \in [d]$, then $\mathbf{D}_{\boldsymbol{\theta}^*,\boldsymbol{\lambda}^{(1)}}(\tau) \leq \mathbf{D}_{\boldsymbol{\theta}^*,\boldsymbol{\lambda}^{(2)}}(\tau), \quad \forall \tau > 0$.*

Property (3) in Proposition 3.5 suggests that the faster $\mathbf{H}_{\boldsymbol{\theta}^*,\boldsymbol{\lambda}}(\cdot)$ decreases, the better the spectrum $\boldsymbol{\lambda}$ aligns with the signal $\boldsymbol{\theta}^*$. In the extreme case where the ordering of $\lambda_i$ matches the ordering of $|\theta_i^*|^2$, the decay of $\mathbf{H}_{\boldsymbol{\theta}^*,\boldsymbol{\lambda}}(\cdot)$ is the fastest, which leads to the most favorable span profile.

**Extensions.** To save space, we defer the extensions to linear models and kernel regression to Appendices A and B, respectively. For the kernel regression model in Equations (1) and (2), we define the ESD of $f^*$ w.r.t. the kernel $\mathbf{k}$ and the effective noise variance $\sigma_{\text{eff}}^2 := (\sigma_0^2 + \|f^*\|_\infty^2)/n$ as

$$d^\dagger(\sigma_{\text{eff}}^2; f^*, \mathbf{k}) = \min\{k \in \mathbb{N}_+ \cup \{\infty\} : \mathbf{H}_{\boldsymbol{\theta}^*,\boldsymbol{\lambda}}(k) \leq \sigma_{\text{eff}}^2\}.$$

## 4 MINIMAX OPTIMAL CONVERGENCE RATES

When using the span profile to characterize the signal-spectrum alignment, it is of interest to establish the optimal convergence rates. Since the setting where $d = d_n$ is finite and grows along with $n$ can be studied using Theorem 3.3 for every finite $n$, we focus on the case where $d = \infty$ and the spectrum $\boldsymbol{\lambda}$ is given with ordering denoted by $\{\pi_j\}$ such that $\lambda_{\pi_1} > \lambda_{\pi_2} > \ldots$. In this asymptotic analysis, $n$ is growing and the noise variance $\sigma^2$ is set to be $\sigma_0^2/n$ where $\sigma_0^2$ is fixed.

We begin by defining a class of populations whose span profile is bounded by a sequence of quotas $\boldsymbol{K} = \{K_n\}_{n=1}^\infty$. This leads to the following class of parameters:

$$\mathcal{F}_{\boldsymbol{K},\boldsymbol{\lambda}} := \left\{ \boldsymbol{\theta} \in \mathbb{R}^\infty : \mathbf{D}_{\boldsymbol{\theta},\boldsymbol{\lambda}}\left(\frac{\sigma_0^2}{n}\right) \leq K_n, \quad \forall n \geq n_0 \text{ for some } n_0 \right\}. \tag{11}$$

For each $\boldsymbol{\theta} \in \mathcal{F}_{\boldsymbol{K},\boldsymbol{\lambda}}$, the sequence model in Equation (4) with $\boldsymbol{\theta}^* = \boldsymbol{\theta}$ and $\sigma^2 = \sigma_0^2/n$ will have an ESD no greater than $K_n$. For a sample $\boldsymbol{Z}^{(n)}$ from this sequence model and any estimator $\widehat{\boldsymbol{\theta}}$ based on $\boldsymbol{Z}^{(n)}$, we aim to determine the convergence rate of the following minimax risk:

$$\inf_{\widehat{\boldsymbol{\theta}}} \sup_{\boldsymbol{\theta} \in \mathcal{F}_{\boldsymbol{K},\boldsymbol{\lambda}}} \mathcal{R}(\widehat{\boldsymbol{\theta}}, \boldsymbol{\theta}). \tag{12}$$

We emphasize that $\boldsymbol{K}$ is a *model-class descriptor*. It is not a parameter of the distribution, but rather describes a condition on the distribution. For example, the sparsity assumption in high-dimensional regression states that $\|\boldsymbol{\beta}\|_0 \leq s$, so $s$ describes a class of distributions; yet $s$ is not a parameter of the distribution. Our minimax result requires a regularity condition on the quota sequence $\boldsymbol{K}$. Let $\bar{K} := \sup\{K_n\} \in \mathbb{N} \cup \{\infty\}$. For any $k \in [\bar{K}]$, let $M_k := \max\{n : K_n = k\}$ (the largest $n$ such that $K_n = k$).

**Condition 4.1.** *(1) $K_{n+1} - K_n \leq 1$ for all $n$ sufficiently large. (2) For all $k \in [\bar{K}]$, it holds that $(k+1)/M_{k+1} \leq k/M_k$.*

Condition 4.1 ensures that $K_n$ does not grow faster than $n$, and the ratio sequence $\{k/M_k\}$ is nonincreasing. Condition 4.1 is easily satisfied by common growth laws.

**Example 4.2.** *(1) Suppose $K_n \asymp n^a$ where $0 < a < 1$. For any $k$, we have $M_k \asymp k^{1/a}$. Since $k/k^{1/a}$ is decreasing, Condition 4.1 holds.*

*(2) Suppose $K_n \asymp (\log n)^b$ where $b > 0$. For any $k$, we have $M_k \asymp e^{k^{1/b}}$. Since $k/e^{k^{1/b}}$ is decreasing, Condition 4.1 holds.*

The next theorem provides a lower bound on the minimax risk in Equation (12).

**Theorem 4.3.** *Suppose Condition 4.1 holds for a quota sequence $\boldsymbol{K} = \{K_n\}_{n=1}^\infty$. Let $c_0 = 1/4$. If $\boldsymbol{Z}^{(n)}$ is drawn from the sequence model with $\boldsymbol{\theta}^* = \boldsymbol{\theta}$ and $\sigma^2 = \sigma_0^2/n$, it holds that*

$$\inf_{\widehat{\boldsymbol{\theta}}} \sup_{\boldsymbol{\theta} \in \mathcal{F}_{\boldsymbol{K},\boldsymbol{\lambda}}} \mathcal{R}(\widehat{\boldsymbol{\theta}}, \boldsymbol{\theta}) \geq c_0 \sigma_0^2 \frac{K_n}{n}.$$

Theorem 4.3 shows that given a quota sequence $\boldsymbol{K}$, no estimator can, uniformly over the class $\mathcal{F}_{\boldsymbol{K},\boldsymbol{\lambda}}$, achieve a faster convergence rate of risk than $\sigma_0^2 K_n/n$. On the other hand, Theorem 3.2 (using $\sigma^2 = \sigma_0^2/n$) provides a matched upper bound on the risk of the optimal PC estimator, which is $2\sigma_0^2 K_n/n$. We thus conclude that the minimax optimal rate over $\mathcal{F}_{\boldsymbol{K},\boldsymbol{\lambda}}$ is $\sigma_0^2 K_n/n$.

Our theory suggests that the ESD is an essential quantity, and the span profile provides a useful characterization of the attainable error rate for spectral methods. Our theory does not invoke any source condition or eigenvalue-decay condition and is more general than classical analysis:

1. The ESD framework recovers classical minimax rates. For spectra with polynomial decay $\lambda_j \asymp j^{-\gamma}$ and signals satisfying source conditions with smoothness index $s$, the ESD quota is $K_n \asymp n^{\frac{1}{1+s\gamma}}$. Consequently, our rate $\sigma_0^2 K_n/n$ recovers the standard optimal rate $n^{-\frac{s\gamma}{1+s\gamma}}$ (see Example D.3 for the full derivation).

2. The ESD framework can provide sharp rates for irregular classes where standard theory is silent. The next example presents a case where the minimax convergence rate is faster than the standard rate in classical analysis based on source conditions.

**Example 4.4.** *Let $b \geq 1$ be a constant and $K_n = \lceil (\log n)^{1/b} \rceil$ for $n \in \mathbb{N}_+$. Suppose $\{\lambda_j\}_{j=1}^\infty$ is decreasing and $\theta_{j+1}^* = \sqrt{\sigma_0^2 \left[ je^{-j^b} - (j+1)e^{-(j+1)^b} \right]}$ for $j \geq 1$ and $\theta_1^* = 0$. Then, $\boldsymbol{\theta}^* \in \mathcal{F}_{\boldsymbol{K},\boldsymbol{\lambda}}$ and the optimal rate is $\sigma_0^2 (\log n)^{1/b} n^{-1}$. In contrast, the traditional convergence rate based on the source condition is $\sigma_0^2 n^{-\alpha/(1+\alpha)}$ for arbitrary $\alpha > 0$, which is not sharp.*

## 4.1 TRACKING DYNAMIC ALIGNMENT

The key advantage of the ESD framework over classical theories is its utility in analyzing adaptive learning, where the main difficulty is that the signal-kernel alignment changes during training.

As discussed in Section 2, the polynomial eigenvalue decay condition ($\lambda_j \asymp j^{-\gamma}$) fails to hold and the Hölder source condition $\sum_j \langle f^*, \psi_j \rangle^2 / \lambda_j^s \leq R_s$ is ill-defined in adaptive learning. In adaptive learning, the learned kernel's eigenvalues and eigenfunctions evolve, making it analytically intractable to track the smoothness index $s$ and radius $R_s$ along the training trajectory.

In contrast, ESD is valid for evolving eigenfunctions. For any time-dependent kernel path $(\mathbf{k}_t)_{t \geq 0}$, the quantity $d^\dagger \left( \sigma_{\text{eff}}^2; f^*, \mathbf{k}_t \right)$ remains a well-defined population measure of complexity at each training time $t$, even when both eigenvalues and eigenfunctions evolve. This allows us to mathematically track the improvement in kernel-signal alignment during training, a dynamic that classical notions cannot easily describe. This *pathwise ESD* framework is further discussed in Appendix C.2.

Using the pathwise ESD, we can provide a mechanistic explanation for the generalization benefits of feature learning. An adaptive learning algorithm aligns the kernel's leading eigenfunctions with the signal, so that a signal that is poorly aligned with a large ESD at initialization may become well-aligned with a small ESD after training. By reducing the ESD, the algorithm actively moves the problem from a model class with high minimax risk to one with low minimax risk.

The above mechanistic explanation is supported by our minimax theorems (Theorem 3.3 and Theorem B.6). We rigorously establish the ESD reduction in Theorem 5.2 for the adaptive sequence model studied by Li & Lin (2024), where we prove the ESD is guaranteed to decrease over time. This result is obtained in a stylized model and should be viewed as a tractable case study of dynamic signal-kernel alignment. We expect similar ESD reduction to appear for other adaptive algorithms, including finite-width neural networks trained with stochastic gradient descent, but the dynamics of learned kernels in those settings are often analytically intractable and establishing such guarantees remains an open problem. Nonetheless, we provide empirical validation using a four-layer deep linear network in Appendix C.2, which demonstrates that ESD tracks the decay of prediction risk even when eigenfunctions evolve.

## 5 ADAPTIVE EIGENVALUES VIA OVER-PARAMETERIZED GRADIENT FLOW

This section will investigate the benefits of learning eigenvalues via over-parameterized gradient flow (OP-GF) in sequence models (Li & Lin, 2024) through the lens of ESDs.

Inspired by the over-parameterized nature of deep neural networks, Li & Lin (2024) parameterized $\theta_j = a_j b_{j,1} \cdots b_{j,D} \beta_j$, where $D$ stands for the number of layers and $(a_j, b_{j,i}, \beta_j)$ are parameters to be learned. The gradient flow w.r.t. the empirical loss $L = \frac{1}{2} \sum_j (\theta_j - y_j)^2$ is given by

$$\dot{a}_j = -\nabla_{a_j} L, \quad \dot{b}_{j,i} = -\nabla_{b_{j,i}} L \ (i \in [D]), \quad \dot{\beta}_j = -\nabla_{\beta_j} L, \tag{13}$$

$$a_j(0) = \lambda_j^{1/2}, \quad b_{j,i}(0) = b_0 > 0, \ (i \in [D]), \quad \beta_j(0) = 0, \ j \in [d],$$

where $\lambda_j$'s are the initial eigenvalues and $b_0$ is the common initialization of all $b_{j,i}$. At time $t$, the learned eigenvalues are given by $\tilde{\lambda}_j(t) = (a_j(t) b_{j,1}(t) \cdots b_{j,D}(t))^2$ and the OP-GF estimates are $\hat{\theta}_j^{OP}(t) = \tilde{\lambda}_j^{\frac{1}{2}}(t) \beta_j(t)$ for $j \in [d]$. Li & Lin (2024) consider infinite-dimensional sequence models with a polynomial decay condition on the initial eigenvalues and establish upper bounds on the risk of the OP-GF estimator with proper early stopping.

Here we study the dynamics of eigenvalues in OP-GF and how it changes the ESD. At time $t$, the learned eigenvalues are $\tilde{\boldsymbol{\lambda}}(t) := (\tilde{\lambda}_j(t))_{j \in [d]}$, and the ESD is $d^\dagger(t) = d^\dagger(\sigma^2; \boldsymbol{\theta}^*, \tilde{\boldsymbol{\lambda}}(t))$. We aim to show that under some regularity conditions, OP-GF can adjust the ordering of eigenvalues $\tilde{\boldsymbol{\lambda}}(t)$ to reduce the ESD $d^\dagger(t)$, which leads to a better signal-spectrum alignment.

We begin with some notations for the sequence model in Equation (4). Following the asymptotic framework in Section 4, we set $\sigma^2 = \frac{\sigma_0^2}{n}$ and $\sigma_0 = 1$ without loss of generality. Denote $\tilde{d} = \sum_{i=1}^d \lambda_i$ (i.e., sum of initial eigenvalues). Let $\pi_t^{-1}(i)$ denote the rank of $\tilde{\lambda}_i(t)$ at time $t$.

**Assumption 5.1.** *We assume (1) Each noise $\xi_j$ in Equation (4) is sub-Gaussian with variance proxy bounded by $C_{proxy}\sigma^2$. (2) Let $\varepsilon = 2C_{proxy}^{-1/2} n^{-1/2} \sqrt{\ln n \tilde{d} \cdot \ln n}$ and $\varepsilon' = 2C_{proxy}^{-1/2} n^{-1/2} \sqrt{\ln n}$. Define $S := \{j \in [d] : |\theta_j^*| > \varepsilon\}$. We have $|S| \leq n$. (3) $\inf_{j \in S} \lambda_j > n^{-\delta}$ for some $\delta \in (0, 1)$.*

**Theorem 5.2.** *Suppose that Assumption 5.1 holds and the initialization in Equation (13) is $b_0 = c_B D^{\frac{D+1}{D+2}} \varepsilon^{\frac{1}{D+2}}$. Define $t_2 = C \cdot D^{\frac{D}{D+2}} (\varepsilon)^{-\frac{2D+2}{D+2}}$. There exist some constants $c, C, C_M, C_{max}, C_\eta, c_\eta, c_B$, and $c'$, such that with probability larger than $1 - 4/n$, we have*

$$d^\dagger(t_2) \leq d^\dagger(t_1)$$

*for any $t_1 \in [0, t_2)$ if the followings hold:*

> *1. For any $j \in S$, we have $M \leq |\theta_j^*|$, where $M := C_M \varepsilon$;*
>
> *2. For any $j \in S^c$, we have $|\theta_j^*| \leq \tilde{\sigma}$, where $\tilde{\sigma} = c' \varepsilon$.*
>
> *3. For any $i, j \in S$, let $\eta_{i,j} := |\theta_i^*| - |\theta_j^*|$. At least one of the followings hold: (a) $\eta_{i,j} \leq 0$, (b) $\eta_{i,j} \geq C_\eta \varepsilon$ and $|\theta_i^*| \leq C_{max} M$, or (c) $\frac{|\theta_i^*|}{|\theta_j^*|} > (1 + \frac{c_\eta}{D})$.*
>
> *4. At time $t_1$, define two subsets of $S^c$: $A_1 := \{i \in S^c : \pi_{t_1}^{-1}(i) < d^\dagger(t_1), \lambda_i < c \cdot D^{-\frac{D}{D+2}} \cdot M^{\frac{2}{D+2}}\}$ and $B_1 := \{i \in S^c : \pi_{t_1}^{-1}(i) > d^\dagger(t_1)\}$, and define a subset of $S$: $B_2 := \{i \in S : \pi_{t_1}^{-1}(i) > d^\dagger(t_1)\}$. It holds that $|B_2| + \min[(|A_1| - |B_2|)_+, |B_1|] \leq |B_2|(C_M/c')^2$.*

Theorem 5.2 shows that OP-GF reduces the ESD. The proof is highly technical, but the underlying idea is simple: the gradient flow dynamics increase the eigenvalues much more rapidly along directions where the signal energy $|\theta_j^*|^2$ is large and slower along directions where the signal is small. This dynamics explicitly reorders the spectrum to match the signal structure.

Li & Lin (2024) have analyzed the risk of the OP-GF algorithm, showing that the generalization is improved during training, but they did not provide a complexity measure to explain why the risk improves. Our work fills this gap by using ESD as the explanatory variable to show that the risk improvement results from the reduction of ESD. Initially, the learning problem lies in a model class with a large quota $K_{t_1} = d^\dagger(t_1)$ and as the ESD reduces, the problem lies in a model class with a smaller quota $K_{t_2} = d^\dagger(t_2)$. Since the minimax risk scales as $\tilde{O}(\sigma^2 K)$, the ESD reduction suggests that the learning problem has been moved from a harder class to an easier class. This explains the potential for generalization improves over time.

## 6 NUMERICAL EXPERIMENTS

**Data Generation.** We utilize the **misalignment** setting in Li & Lin (2024) to specify a $d$-dimensional sequence model. We fixed the eigen-decay rate $\gamma > 0$, the signal decay rate $p > 0$, and the number of nonzero signals $J$. Given any misalignment parameter $q \geq 1$, we set eigenvalues as $\lambda_j = j^{-\gamma}, j \in [d]$, and set the true nonzero parameters as $\theta^*_{\ell(j)} = C \cdot j^{-\frac{p+1}{2}}$, where $\ell(j) = [j^q]$ and $j \leq J$. Here all other elements of $\theta^*_j$ are zero and $d \geq J^q$ so $\|\theta^*\|_0 = J$. The observations are sampled as $y_i \sim N(\theta^*_i, \sigma^2)$. This setting provides a flexible way to control the alignment between the signal structure and the spectrum. When $q = 1$, the ordering of $\theta^*$ align perfectly with the ordering of $\lambda$. As $q$ increases, more nonzero elements of $\theta^*$ are located on the tail where the eigenvalues are smaller, and more large eigenvalues are associated with zero signals, creating a worse signal-spectrum alignment.

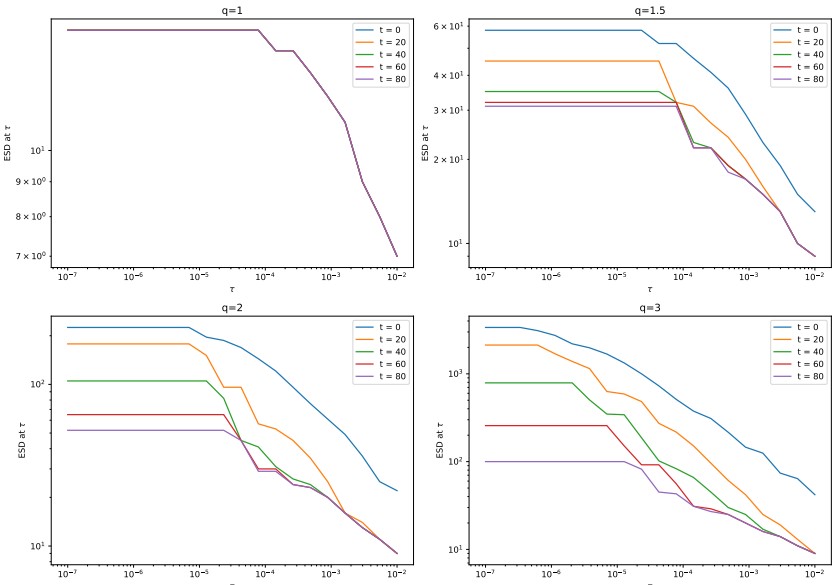

Figure 1: Evolution of span profiles during the training of an over-parameterized gradient flow. The misalignment level $q$ varies from 1 to 3. Fixed parameters are $n = 10000$, $\sigma_0 = 1$, $d = 5000$, $J = 15$, $p = 2.5$, and $\gamma = 1$.

**Evolution of Span Profile** The first experiment visualizes the span profile of the signal w.r.t. the learned spectrum at various stages in the OP-GF process with $D = 0$. Given a sample, we approximate the gradient flow in Equation (13) by discrete-time gradient descent and obtain the solution $\{(a_j(t), \beta_j(t))_{t \geq 0} : j \in [d]\}$. The trained eigenvalue sequence $\tilde{\lambda}(t)$ at time $t$ is given by $\tilde{\lambda}_j(t) = a_j^2(t)$ for $j \in [d]$. Here we focus on time points before the optimal stopping time. Figure 1 illustrates the evolution of the span profile w.r.t. the learned spectrum for different training times $t$ and various values of $q$.

When $q = 1$ (Top-Left panel), the span profiles at different training times $t$ are nearly identical. This is because the initial spectrum already aligns perfectly with the signal and there is no room for improvement. For $q > 1$ (Top-Right, Bottom-Left, Bottom-Right panels), we observe that as the training time $t$ increases 0 to 80, the span profile shifts downwards. This suggests that the training process refines the alignment between the spectrum and the signal. In addition, the reduction in the span profile is more significant for $q = 3$ compared to $q = 1.5$, because $q = 3$ corresponds to a greater initial misalignment between the signal and the spectrum, rendering the improvement from OP-GF more substantial.

**Evolution of ESD and Estimation Error of PC Estimators** We next empirically investigate the evolution of the ESD $d^\dagger$ and the estimation error as well as the impact of layers $D$. At any time $t$, we compute the ESD $d^\dagger(t)$ based on the learned eigenvalue sequence $\tilde{\lambda}(t)$ $\tilde{\lambda}(t)$ and also the PC estimate $\widehat{\theta}(t)$ based on $\tilde{\lambda}(t)$, with number of components determined by $d^\dagger(t)$. Theorems 3.2 and 3.3

suggest that the PC estimator tuned by the ESD can achieve the minimax risk rate, so we expect $\widehat{\boldsymbol{\theta}}(t)$ to perform well.

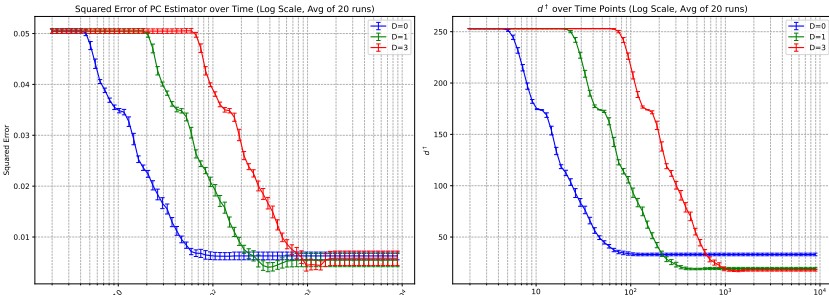

Figure 2: Averaged squared error of the tuned PC estimator and ESD as a function of the training time. Each average is computed based on 20 replications and each error bar represents a standard deviation.

The empirical evaluation involved 20 Monte Carlo repetitions. Figure 2 displays the averaged $d^{\dagger}(t)$ and the averaged estimation error of $\widehat{\boldsymbol{\theta}}(t)$ in Figure 2 as a function of training time $t$. We observe that both the ESD and the squared error of the tune PC estimator exhibit a general decay trend over training time $t$. Furthermore, for the shallow model with $D = 0$ (with no $b_{i,j}$ parameters), the initial decrease in ESD and MSE occurs earlier compared to the deeper models with $D = 1$ or $D = 3$. However, with sufficient training iterations, the deeper models with $D = 1$ or $D = 3$ can achieve lower ESD values than the shallow model with $D = 0$. These findings suggest that increased model depth ($D > 0$) may facilitate a better adaptation of the spectrum, and thus lead to lower estimation error. This observation offers a perspective on the benefits of depth in spectral learning, but a comprehensive study for general models is left for future research.

# 7 DISCUSSION

This paper introduces the effective span dimension (ESD) and span profile to analyze the interplay between the signal structure and the kernel spectrum. Our framework moves beyond classical static assumptions relative to a fixed kernel (e.g., source conditions and polynomial eigenvalue decay) and offers a dynamic, noise-dependent perspective on signal complexity. Unlike traditional source conditions, the ESD is more flexible and remains applicable when the spectrum itself is learned from data.

**Quantifying adaptivity.** Like the sparsity level in high-dimensional statistics, the ESD is a population quantity for theoretical analyses rather than an input to training. It serves as a quantitative target for adaptive algorithms on the population level: by comparing the ESD of a particular signal w.r.t. different kernels, we can determine which kernel permits better generalization for this signal.

**Connecting adaptivity and generalization.** Our framework clarifies why adaptive learning methods often outperform classical fixed-kernel approaches. As detailed in Section 4.1, a fixed kernel $\boldsymbol{\lambda}^{(0)}$ forces the signal into a class $\mathcal{F}_{\mathbf{K}^{(0)}}$ characterized by high minimax risk. By contrast, successful adaptation modifies the kernel to reduce the span profile $D_{\boldsymbol{\theta}^*, \boldsymbol{\lambda}^{(a)}}$ of the same signal w.r.t. the adapted kernel spectrum $\boldsymbol{\lambda}^{(a)}$. This adaptation places the signal into a class $\mathcal{F}_{\mathbf{K}^{(a)}}$ with a smaller quota $\mathbf{K}^{(a)}$, which implies lower minimax risk.

**Analysis of ESD dynamics**. The theoretical analysis of ESD dynamics in Theorem 5.2 utilizes a tractable proxy model to rigorously study the OP-GF. This provides a mechanistic prototype where feature learning is formalized as ESD reduction. In numerical studies, the ESD dynamics can be empirically measured in various learning settings. However, establishing rigorous theoretical results for specific adaptive algorithms remains an important open problem.

In summary, the ESD framework provides a novel view of generalization that connects classical kernel methods with modern adaptive learning. We expect to relate this framework to learned representations in neural networks to explain their superior generalization performance.

## REPRODUCIBILITY STATEMENT

There is no new datasets used. We will release anonymized source code in the supplementary material, which enables reproduction of all experiments.

## LARGE LANGUAGE MODELS STATEMENT

Large language models were used to improve the clarity of the manuscript and to facilitate experiments.

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

# A  EXTENSION TO CORRELATED NOISE AND FIXED-DESIGN LINEAR MODEL

This section extends the concepts of ESD and span profile, developed in Section 3 for the sequence model, to the setting of fixed-design linear regression. In addition, we demonstrate how the minimax optimal prediction risk in this setting can be characterized using the span profile, paralleling the analysis in Section 4.

## A.1  STRATEGY OF REDUCTION

Before introducing the linear model, it is helpful to outline our general transformation strategy in the context of a sequence model with correlated noise. Suppose $d \in \mathbb{N}_+$ and the observations are

$$\mathbf{Z} = \boldsymbol{\theta}^\star + \boldsymbol{\xi}, \quad \boldsymbol{\xi} \sim \mathcal{N}_d\left(0, \sigma^2 \boldsymbol{\Sigma}_\xi\right) \tag{14}$$

where $\boldsymbol{\Sigma}_\xi \in \mathbb{R}^{d \times d}$ is known, symmetric, and positive definite. For a correlated sequence model, it is usually of interest to measure the estimation error using the squared Mahalanobis distance defined as $L(\boldsymbol{\theta}^\star, \widehat{\boldsymbol{\theta}}^\star) = (\widehat{\boldsymbol{\theta}}^\star - \boldsymbol{\theta}^\star)^\top \boldsymbol{\Sigma}_\xi^{-1} (\widehat{\boldsymbol{\theta}}^\star - \boldsymbol{\theta}^\star)$.

Let $\mathbf{L} = \boldsymbol{\Sigma}_\xi^{-1/2}$ be a symmetric square root of $\boldsymbol{\Sigma}_\xi^{-1}$. Define the whitened observation and transformed parameters as

$$\tilde{\mathbf{Z}} = \mathbf{L}\mathbf{Z}, \quad \tilde{\boldsymbol{\theta}}^\star = \mathbf{L}\boldsymbol{\theta}^\star.$$

It follows that $\tilde{\mathbf{Z}} = \tilde{\boldsymbol{\theta}}^\star + \tilde{\boldsymbol{\xi}}$, where $\tilde{\boldsymbol{\xi}} = \mathbf{L}\boldsymbol{\xi} \sim \mathcal{N}_d\left(0, \sigma^2 \mathbf{I}_d\right)$. Accordingly, any estimator $\widehat{\boldsymbol{\theta}}$ for $\boldsymbol{\theta}^\star$ is equivalent to the estimator $\widehat{\tilde{\boldsymbol{\theta}}} := \mathbf{L}\widehat{\boldsymbol{\theta}}$ for $\tilde{\boldsymbol{\theta}}^\star$, whose squared loss is $\|\widehat{\tilde{\boldsymbol{\theta}}} - \tilde{\boldsymbol{\theta}}^\star\|^2 = (\widehat{\boldsymbol{\theta}}^\star - \boldsymbol{\theta}^\star)^\top \boldsymbol{\Sigma}_\xi^{-1} (\widehat{\boldsymbol{\theta}}^\star - \boldsymbol{\theta}^\star) = L(\boldsymbol{\theta}^\star, \widehat{\boldsymbol{\theta}}^\star)$.

Therefore, the transformed model is equivalent to the standard sequence model with uncorrelated noise in Equation (4) and the estimation is equivalent to the estimation therein. Consequently, the ESD and span profile for the model in Equation (14) can be naturally defined using the original definitions for the transformed model. Specifically, for the correlated-noise model, we define the ESD w.r.t. $\Sigma_\xi$ as

$$d^\dagger_{\boldsymbol{\Sigma}_\xi}(\tau; \boldsymbol{\theta}^\star) := d^\dagger\left(\tau; \tilde{\boldsymbol{\theta}}^\star = \mathbf{L}\boldsymbol{\theta}^\star\right).$$

Note that for the risk $\mathbb{E}L(\boldsymbol{\theta}^\star, \widehat{\boldsymbol{\theta}}^\star)$, our minimax risk characterization still applies, i.e., the minimax risk scales as $K\sigma^2$ across all distributions whose ESD $d^\dagger_{\boldsymbol{\Sigma}_\xi}(\sigma^2; \boldsymbol{\theta}^\star)$ is bounded by $K$.

The relationship between Euclidean distance and Mahalanobis distance satisfies that

$$\lambda_{\min}(\boldsymbol{\Sigma}_\xi) L(\boldsymbol{\theta}^\star, \widehat{\boldsymbol{\theta}}^\star) \leq \|\boldsymbol{\theta}^\star - \widehat{\boldsymbol{\theta}}^\star\|^2 \leq \lambda_{\max}(\boldsymbol{\Sigma}_\xi) L(\boldsymbol{\theta}^\star, \widehat{\boldsymbol{\theta}}^\star),$$

so the minimax risk in terms of $\mathbb{E}\|\boldsymbol{\theta}^\star - \widehat{\boldsymbol{\theta}}^\star\|^2$ can still be characterized sharply when the condition number of $\boldsymbol{\Sigma}_\xi$ is bounded.

This strategy of reducing a complex model to a simple model will be used in our analysis of the linear model and in RKHS regression in Appendix B.

## A.2  LINEAR MODEL

Consider the following fixed-design linear regression model:

$$\mathbf{Y} = \mathbf{X}\boldsymbol{\beta}^* + \boldsymbol{\epsilon}, \tag{15}$$

where $\mathbf{Y} \in \mathbb{R}^n$ is the vector of observations, $\mathbf{X} \in \mathbb{R}^{n \times p}$ is the fixed-design matrix of rank $r \leq \min(n, p)$, $\boldsymbol{\beta}^* \in \mathbb{R}^p$ is the unknown vector of true coefficients, and $\boldsymbol{\epsilon} \in \mathbb{R}^n$ is the noise vector. We assume the components of $\boldsymbol{\epsilon}$ are uncorrelated with mean zero and variance $\sigma_0^2$. For this model, we consider the loss (in-sample prediction error) $\mathcal{L}(\widehat{\boldsymbol{\beta}}; \boldsymbol{\beta}^*) = \frac{1}{n}\|\mathbf{X}\left(\widehat{\boldsymbol{\beta}} - \boldsymbol{\beta}^*\right)\|^2$ and risk $\mathcal{R}(\widehat{\boldsymbol{\beta}}; \boldsymbol{\beta}^*) = \mathbb{E}\mathcal{L}(\widehat{\boldsymbol{\beta}}; \boldsymbol{\beta}^*)$. For random design linear regression, we treat it as a special case of RKHS regression and discuss it in Appendix B.

To connect this model to the sequence model analysis presented earlier, we utilize the Singular Value Decomposition (SVD) of the design matrix $\mathbf{X}$ as follows:

$$\frac{1}{\sqrt{n}}\mathbf{X} = \mathbf{U}\mathbf{S}\mathbf{V}^\top, \tag{16}$$

where $\mathbf{U} \in \mathbb{R}^{n \times n}$ and $\mathbf{V} \in \mathbb{R}^{p \times p}$ are orthogonal matrices, and $\mathbf{S} \in \mathbb{R}^{n \times p}$ is a rectangular diagonal matrix with non-negative singular values $s_1 \geq s_2 \geq \cdots \geq s_r > 0$ on its diagonal, and $s_j = 0$ for $j > r$.

For any matrix $\mathbf{A}$ and subsets $R$ and $T$, we write $\mathbf{A}_{\cdot,R}$ for the submatrix formed by the columns of $\mathbf{A}$ with indices in $R$, and write $\mathbf{A}_{T,\cdot}$ for the submatrix formed by the rows of $\mathbf{A}$ with indices in $T$.

Multiplying the model in Equation (15) by $\frac{1}{\sqrt{n}}\mathbf{U}_{\cdot,[r]}^\top$, we obtain a $r$-dimensional transformed model:

$$\mathbf{Z} = \boldsymbol{\theta}^* + \boldsymbol{\xi}, \tag{17}$$

where we have defined $\mathbf{Z} = \frac{1}{\sqrt{n}}\mathbf{U}_{\cdot,[r]}^\top\mathbf{Y}$, $\boldsymbol{\theta}^* = \frac{1}{\sqrt{n}}\mathbf{U}_{\cdot,[r]}^\top\mathbf{X}\boldsymbol{\beta}^* = \mathbf{S}_{[r],\cdot}\mathbf{V}^\top\boldsymbol{\beta}^*$, and $\boldsymbol{\xi} = \frac{1}{\sqrt{n}}\mathbf{U}_{\cdot,[r]}^\top\boldsymbol{\epsilon}$. Since $\mathbf{U}$ is orthogonal, the transformed noise vector $\boldsymbol{\xi}$ still has uncorrelated components with mean zero and variance $\sigma_{\text{eff}}^2 := \sigma_0^2/n$.

The transformed model in Equation (17) is analogous to the sequence model in Equation (4), where the signal is $\boldsymbol{\theta}^*$ and the noise variance for each component is $\sigma_{\text{eff}}^2$. The "spectrum" relevant to this problem is derived from the singular values of $\mathbf{X}$. Specifically, we define the eigenvalues as $\lambda_j = s_j^2$ for $j = 1, \ldots, r$, and $\lambda_j = 0$ for $j > r$. Let $\{\pi_k\}_{k=1}^r$ denote the indices corresponding to the eigenvalues sorted in descending order, $\lambda_{\pi_1} \geq \lambda_{\pi_2} \geq \cdots \geq \lambda_{\pi_r} > 0$.

For any estimator $\widehat{\boldsymbol{\beta}}$ for the linear model Equation (15), define $\widehat{\boldsymbol{\theta}} = \mathbf{S}_{[r],\cdot}\mathbf{V}^\top\widehat{\boldsymbol{\beta}}$. We can then write the prediction risk as $n^{-1}\mathbb{E}\|\mathbf{X}\widehat{\boldsymbol{\beta}} - \mathbf{X}\boldsymbol{\beta}^*\|^2 = \mathbb{E}\|\mathbf{U}\mathbf{S}\mathbf{V}^\top\widehat{\boldsymbol{\beta}} - \mathbf{U}\mathbf{S}\mathbf{V}^\top\boldsymbol{\beta}^*\|^2 = \mathbb{E}\|\widehat{\boldsymbol{\theta}} - \boldsymbol{\theta}^*\|^2$. Conversely, given an estimator $\widehat{\boldsymbol{\theta}}$ for the sequence model Equation (17), we can define $\widetilde{\boldsymbol{\beta}} = \mathbf{V}\mathbf{S}^\dagger\widehat{\boldsymbol{\theta}}$ where $\mathbf{S}^\dagger \in \mathbb{R}^{p \times r}$ is a diagonal matrix whose diagonal elements are $\{1/s_j\}_{j \in [r]}$. It is easy to check that $\mathbf{S}_{[r],\cdot}\mathbf{V}^\top\widetilde{\boldsymbol{\beta}} = \widehat{\boldsymbol{\theta}}$. Therefore, we establish an equivalence between the model Equation (17) and the model Equation (15).

The usual ridge regression estimator for the linear model Equation (15) is given by

$$\widehat{\boldsymbol{\beta}}_\nu = \left(\mathbf{X}^\top\mathbf{X} + \nu\,\mathbf{I}_p\right)^{-1}\mathbf{X}^\top\mathbf{Y},$$

which transforms into

$$\widehat{\boldsymbol{\theta}}_\nu = \mathbf{S}_{[r],\cdot}\mathbf{V}^\top\widehat{\boldsymbol{\beta}}_\nu = \left(\mathbf{I}_r - g_\nu(\text{Diag}\,(\lambda_1, \ldots, \lambda_r))\right)\mathbf{Z}, \qquad \text{where } g_\nu(\lambda) = \frac{1}{\lambda/\nu + 1}.$$

In the above expression, we have used the identity that $\mathbf{S}_{[r],\cdot}\mathbf{S}_{[r],\cdot}^\top = \text{Diag}\,(s_1^2, \ldots, s_r^2)$ and $g_\nu(\cdot)$ is applied element-wise. If we replace $g_\nu(\lambda)$ by other functions as discussed in Section 3, we recover other spectral methods.

### A.3 ESD FOR LINEAR MODELS

We can now adapt the definitions from Section 3 to linear models.

**Definition A.1** (ESD for Linear Regression). *Suppose the SVD of the design matrix $\mathbf{X}$ is given in Equation (16). The Effective Span Dimension (ESD) of $\boldsymbol{\beta}^*$ with respect to the design $\mathbf{X}$ and the per component variance $\sigma_0^2/n$ is defined as*

$$d^\dagger = d^\dagger(\sigma_0^2/n; \boldsymbol{\beta}^*, \mathbf{X}) = \min\{k \in [r] : \mathbf{H}_{\boldsymbol{\theta}^*, \boldsymbol{\lambda}}(k) \leq \sigma_0^2/n\},$$

*where $\boldsymbol{\theta}^* = \mathbf{S}_{\cdot,[r]}\mathbf{V}^\top\boldsymbol{\beta}^*$ and $\lambda_j = s_j^2$.*

The Principal Component Regression (PCR) estimator for $\boldsymbol{\beta}^*$ corresponds to the Principal Component (PC) estimator in the transformed space Equation (17). Specifically, for any $k \in [r]$, define $\widehat{\boldsymbol{\beta}}^{\mathrm{PC},k} = \frac{1}{\sqrt{n}} \mathbf{V} \mathbf{S}_k^\dagger \mathbf{U}_{\cdot,[r]}^\top \mathbf{Y}$, where $\mathbf{S}_k^\dagger \in \mathbb{R}^{p \times r}$ is a diagonal matrix whose diagonal elements are $\left\{ \frac{1}{s_j} \mathbf{1}_{\{s_j \geq s_{\pi_k}\}} \right\}$. In the $\mathbf{Z}$ space, this means

$$\hat{\theta}_j^{\mathrm{PC},k} = \mathbf{1}\{s_j \geq s_{\pi_k}\} Z_j, \quad j \in [r]. \tag{18}$$

Analogous to Theorem 3.2, the minimal prediction risk achievable by PCR over $k$ is characterized by the ESD.

**Proposition A.2** (Optimal PCR Prediction Risk). *Let $\widehat{\boldsymbol{\beta}}^{\mathrm{PC},k}$ be the PCR estimator using the first $k$ principal components. Let $\mathcal{R}_*^{\mathrm{PC}}$ be the minimal possible prediction risk over $k \in [r]$, i.e., $\mathcal{R}_*^{\mathrm{PC}} = \min_{k \in [r]} \mathcal{R}(\widehat{\boldsymbol{\beta}}^{\mathrm{PC},k}; \boldsymbol{\beta}^*)$. It holds that*

$$(d^\dagger - 1)\sigma_0^2/n \leq \mathcal{R}_*^{\mathrm{PC}} \leq 2d^\dagger \sigma_0^2/n,$$

*where $d^\dagger = d^\dagger(\sigma_0^2/n; \boldsymbol{\beta}^*, \mathbf{X})$ is the ESD defined in Definition A.1.*

Proposition A.2 directly follows from Theorem 3.2 and its proof is omitted. This result shows that the optimal prediction risk for PCR is determined by the ESD $d^\dagger$, which measures the effective number of principal components needed to balance the bias-variance trade-off.

We can further extend the minimax analysis from Theorem 3.3. Let $K$ be a quota on ESD. Define a class of coefficient vectors based on this quota:

$$\mathcal{B}_K^{(n)} = \left\{ \boldsymbol{\beta}^* \in \mathbb{R}^p : d^\dagger(\sigma_0^2/n; \boldsymbol{\beta}^*, \mathbf{X}) \leq K \right\}, \tag{19}$$

This class contains signals whose ESD relative to the design $\mathbf{X}$ is controlled by $K$. We can establish the minimax optimal rate for prediction over this class.

**Theorem A.3** (Minimax Prediction Risk for Linear Regression). *Suppose $K \leq r$. For the linear model Equation (15) with noise variance $\sigma_0^2$, the minimax prediction risk over the class $\mathcal{B}_K^{(n)}$ defined in Equation (19) satisfies:*

$$\inf_{\widehat{\boldsymbol{\beta}}} \sup_{\boldsymbol{\beta}^* \in \mathcal{B}_K^{(n)}} \mathcal{R}(\widehat{\boldsymbol{\beta}}; \boldsymbol{\beta}^*) \asymp \sigma_0^2 \frac{K}{n}.$$

The proof of Theorem A.3 is essentially the same as that of Theorem 3.3 and is omitted.

Through this extension, the span profile framework connects the optimal prediction performance in fixed-design linear regression to the alignment between the signal structure (transformed via the design matrix) and the spectrum derived from the design matrix's singular values.

### A.4 NUMERICAL ILLUSTRATION

This section illustrates the ESD in fixed-design linear models in two examples. Throughout, we fix the noise variance at $\sigma_0^2 = 1$, the sample size at $n = 300$, and the dimension at $p = 400$.

**Experimental setup** The baseline design matrix $\mathbf{X}_0$ is randomly generated with covariance matrix $\Sigma = \mathrm{Diag}\{\lambda_j\}_{j \in [p]}$ and then held fixed. We consider two cases:

1. *Geometric decay spectrum and polynomial decay signal*: $\lambda_j \propto 0.95^j$ and $\beta_j^* = j^{-0.2}$;

2. *Logarithmic decay spectrum and signal*: $\lambda_j = 1/\log(j+1)$ with $\beta_j^* = 1/\log(j+1)$.

The response will be generated from $\boldsymbol{Y} = \mathbf{X}_0 \boldsymbol{\beta}^* + \boldsymbol{\epsilon}$ with random noise $\boldsymbol{\epsilon}$.

We are interested in the ESD and the minimum risk for different transformations of the design matrix. For this purpose, we introduce a class of non-orthogonal column transformations indexed by $\alpha > 0$ as follows:

$$\mathbf{A}(\alpha) = \mathrm{diag}\big(\exp\{\alpha\, t_j\}\big), \quad t_j = (j-1)/(p-1) - 1/2, \quad j \in [p].$$

The transformed design $\mathbf{X}(\alpha) = \mathbf{X}_0 \mathbf{A}(\alpha)$ and the correspondingly transformed coefficient vector is $\boldsymbol{\beta}(\alpha) = \mathbf{A}(\alpha)^{-1} \boldsymbol{\beta}^*$. These transformations will change the ordering of the spectrum, from well-aligned to misaligned. We are interested in the following at each $\alpha$:

- Effective Span Dimension: $d^\dagger(\alpha) = d^\dagger\big(\sigma_0^2/n; \boldsymbol{\beta}(\alpha), \mathbf{X}(\alpha)\big)$;

- Minimal PCR risk: $\mathcal{R}_*(\alpha) = \min_k \mathbb{E}\big[n^{-1}\|\mathbf{X}(\alpha)\widehat{\boldsymbol{\beta}}_k - \mathbf{X}(\alpha)\boldsymbol{\beta}(\alpha)\|^2\big]$, where $\widehat{\boldsymbol{\beta}}_k$ is the $k$-component principal-component estimator based on $(\boldsymbol{Y}, \mathbf{X}(\alpha))$.

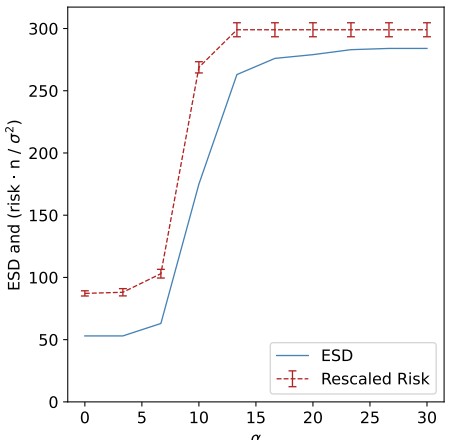 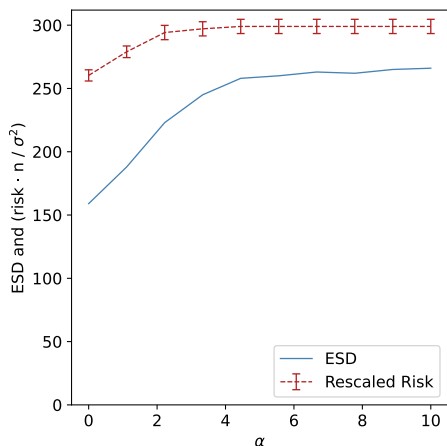

Figure 3: **Oracle PCR risk versus Effective Span Dimension** for (a) geometric eigen-decay and (b) logarithmic eigen-decay. The dashed line plots Risk $\times n/\sigma_0^2$; the solid line is $d^\dagger(\alpha)$. The risk is computed based on 20 replications and the error bar represents the standard deviation.

Figure 3 plots $d^\dagger(\alpha)$ (solid) and the rescaled oracle risk defined as $n\mathcal{R}_*(\alpha)/\sigma_0^2$ (dashed) against $\alpha$. The two curves coincide over the entire path, which empirically verifies the bound in Proposition A.2 that $\mathcal{R}_*(\alpha) \asymp \frac{\sigma_0^2}{n} d^\dagger(\alpha)$. As $\alpha$ grows, the diagonal stretch $\mathbf{A}(\alpha)$ shifts signal energy towards directions that carry smaller singular values. This raises $d^\dagger$, and consequently, the achievable risk.

This experiment illustrates that ESD, rather than raw spectral decay, is the pivotal measure governing learnability.

## B  EXTENSION TO RKHS REGRESSION

This section extends the concepts of Effective Span Dimension (ESD) and span profile, developed in Section 3, to the setting of RKHS regression. Since our goal is to develop a population level complexity measure, we focus on the simple case where the eigenfunctions of the kernel are fully known and computable, and leave a thorough analysis in future studies.

### B.1  RKHS REGRESSION

We recall the standard random-design kernel regression model from Section 2:

$$y_i = f^*(x_i) + \epsilon_i, \quad \epsilon_i \text{ are i.i.d. with } \mathbb{E}[\epsilon_i] = 0,\ \mathrm{Var}(\epsilon_i) = \sigma_0^2, \quad i = 1, \dots, n, \qquad (20)$$

where $x_i \overset{\text{i.i.d.}}{\sim} \mu$, and $f^* \in L^2(\mathcal{X}, \mu)$ is the target function. We use a kernel $\mathbf{k}(\cdot, \cdot)$ with Mercer decomposition $\mathbf{k}(x, x') = \sum_{j=1}^\infty \lambda_j \psi_j(x) \psi_j(x')$, where $\{\psi_j\}_{j=1}^\infty$ form an orthonormal basis for $L^2(\mathcal{X}, \mu)$ and $\boldsymbol{\lambda} = \{\lambda_j\}_{j\geq 1}$ is the sequence of eigenvalues not necessarily sorted. For simplicity, we assume there are no ties among the eigenvalues and let $\pi$ be the permutation that sorts them in descending order, so that $\lambda_{\pi_1} > \lambda_{\pi_2} > \cdots > \lambda_{\pi_k} > \dots$. The coefficients of $f^*$ in this basis are $\theta_j^* = \langle f^*, \psi_j \rangle_{L^2(\mu)}$. An estimator $\hat{f}$ has risk $\mathcal{R}(\hat{f}; f^*) = \mathbb{E}\|\hat{f} - f^*\|_{L^2(\mu)}^2$. If we define $\hat{\theta}_j = \langle \hat{f}, \psi_j \rangle_{L^2(\mu)}$, we can also write the risk as

$$\mathcal{R}(\hat{f}; f^*) = \mathbb{E}\|\hat{f} - f^*\|_{L^2(\mu)}^2 = \mathbb{E}\Big[\sum_{j=1}^\infty \big(\hat{\theta}_j - \theta_j^*\big)^2\Big] = \sum_{j=1}^\infty \mathbb{E}\big(\hat{\theta}_j - \theta_j^*\big)^2. \qquad (21)$$

The expression in Equation (21) suggests that we can equivalently estimate each $\theta_j^*$ separately using the transformed observation $z_j = n^{-1} \sum_i y_i \psi_j(x_i)$ for any $j \geq 1$ as introduced in Equation (2).

Here we kindly remind the reader that in RKHS, we use subscript $i$ for samples and subscript $j$ for eigen-coordinates. The subscript $j$ aligns with the use of notation in the sequence model, where we use indices $j$ to denote coordinates.

**Inflated variance in Equation (3).** Before we introduce the definition of ESD, we demonstrate that the approximation in Equation (3) can be made exact by increasing the variance to absorb the approximation error.

For each $j \geq 1$, we can write the transformed observation as

$$
\begin{aligned}
z_j &= \frac{1}{n} \sum_{i=1}^n y_i \, \psi_j(x_i) \\
&= \frac{1}{n} \sum_{i=1}^n \left( f^*(x_i) + \epsilon_i \right) \psi_j(x_i) \\
&= \frac{1}{n} \sum_{i=1}^n \left( \sum_{k \geq 1} \theta_k^* \psi_k(x_i) \right) \psi_j(x_i) \; + \; \frac{1}{n} \sum_{i=1}^n \epsilon_i \, \psi_j(x_i) \\
&= \sum_{k \geq 1} \theta_k^* \underbrace{\left( \frac{1}{n} \sum_{i=1}^n \psi_k(x_i) \psi_j(x_i) \right)}_{:= G_{kj}} \; + \; \underbrace{\frac{1}{n} \sum_{i=1}^n \epsilon_i \, \psi_j(x_i)}_{= \xi_j} \\
&= \sum_{k \geq 1} G_{kj} \, \theta_k^* \; + \; \xi_j,
\end{aligned}
$$

where $G_{kj} := \frac{1}{n} \sum_{i=1}^n \psi_k(x_i) \psi_j(x_i)$ are entries of the empirical feature correlation matrix. For $\xi_j$, we have $\mathbb{E}(\xi_j \mid \{x_i\}_{i \in [n]}) = 0$ and $\mathbb{E}(\xi_j \xi_k \mid \{x_i\}_{i \in [n]}) = n^{-1} \sigma_0^2 G_{jk}$.

Since $x_i \overset{\text{i.i.d.}}{\sim} \mu$ and $\{\psi_j\}$ are orthonormal in $L^2(\mathcal{X}, \mu)$, we have $\mathbb{E}[G_{kj}] = \mathbf{1}_{\{k=j\}}$. Hence $\mathbb{E}[z_j] = \theta_j^*$.

We may further decompose $z_j$ as follows

$$
z_j - \theta_j^* = (G_{jj} - 1)\theta_j^* + \left( \sum_{k \geq 1, k \neq j} G_{kj} \theta_k^* \right) + \xi_j = \Delta_j + \xi_j,
$$

where we have defined $\Delta_j = (G_{jj} - 1)\theta_j^* + \left( \sum_{k \geq 1, k \neq j} G_{kj} \theta_k^* \right)$. This term does not appear in the sequence model; its randomness arises solely from the random covariates $x_i$. As $n \to \infty$, this term vanishes because $G_{jj} = n^{-1} \sum_i \psi_j(x_i)^2 \to 1$ and $G_{kj} = n^{-1} \sum_{i=1}^n \psi_j(x_i) \psi_{j'}(x_i) \to 0$ for $k \neq j$. Furthermore, since $\mathbb{E}(\xi_j \mid \{x_i\}_{i \in [n]}) = 0$, we have

$$
\mathbb{E}(\Delta_j \xi_j) = \mathbb{E}(\Delta_j \mathbb{E}(\xi_j \mid \{x_i\}_{i \in [n]})) = 0.
$$

The presence of $\Delta_j$ effectively inflates the variance in $z_j$ to $\sigma_0^2/n + \mathrm{Var}(\Delta_j)$. One can show that $\mathrm{Var}(\Delta_j) = \mathrm{Var}(f^*(x)\psi_j(x))$, which is bounded by $\|f^*\|_\infty^2$. This is how we will control the impact of $\Delta_j$ in the following development.

## B.2 ESD FOR RKHS REGRESSION

We start by analyzing the counterpart of the PC estimator, the Kernel Principal Component Projection Estimator (KPCPE), defined as

$$
\hat{f}_k^{\mathrm{PC}}(x) := \sum_{j : \lambda_j \geq \lambda_{\pi_k}} z_j \psi_j(x), \tag{22}
$$

where $k$ is the number of leading eigenfunctions to be included.

The risk $\mathcal{R}_k := \mathbb{E}\|\hat{f}_k^{\mathrm{PC}} - f^*\|_{L^2(\mu)}^2$ decomposes into squared bias $B(k)$ and variance $V(k)$. Since $\mathbb{E}[\psi_{j'}(x_i)\psi_j(x_i)] = \mathbf{1}_{\{j=j'\}}$, we have $\mathbb{E}[z_j] = \theta_j^*$. Therefore, the bias is due to truncation as

$$B(k) = \sum_{j=k+1}^{\infty} (\theta_j^*)^2. \tag{23}$$

The integrated variance is $V(k) = \sum_{j:\lambda_j \geq \lambda_{\pi_k}} \mathrm{Var}(z_j)$. Using the law of total variance, we have

$$\mathrm{Var}(z_j) = \frac{1}{n}\mathrm{Var}(y_i\psi_j(x_i)) = \frac{1}{n}(\sigma_0^2 + \tau_j^2), \quad \text{where} \quad \tau_j^2 := \mathrm{Var}(f^*(x)\psi_j(x)). \tag{24}$$

The term $\tau_j^2$ arises from the randomness of the design $x$. To ensure $V(k)$ grows at the rate of $k/n$, we need to uniformly bound the design-induced variance $\tau_j^2$. To illustrate the idea, we assume $f^*$ is bounded in the sense that $|f^*(X)| \leq \|f^*\|_\infty$, $\mu$-almost surely. Here, $\|f^*\|_\infty$ denotes the essential supremum of $|f^*|$ w.r.t. the measure $\mu$.

**Assumption B.1** (Bounded target). *$f^* \in L^\infty(\mathcal{X}, \mu)$ and $\|f^*\|_\infty = \operatorname{ess\,sup} |f^*|$.*

Assumption B.1 is very mild: for compact $\mathcal{X}$, if $f$ is continuous, then $f$ is bounded.

Under Assumption B.1, $\tau_j^2 \leq \mathbb{E}[f^*(x)^2\psi_j(x)^2] \leq \|f^*\|_\infty^2$. Subsequently, the variance is bounded by $V(k) \leq \frac{k}{n}(\sigma_0^2 + \|f^*\|_\infty^2)$. This motivates us to define the effective noise variance per component as

$$\sigma_{\mathrm{eff}}^2 := \frac{\sigma_0^2 + \|f^*\|_\infty^2}{n}. \tag{25}$$

The effective noise variance $\sigma_{\mathrm{eff}}^2$ includes the term $\|f^*\|_\infty^2/n$, which inflates the noise compared to an idealized sequence model.

We can now adapt the definitions from Section 3.1 using the effective noise variance $\sigma_{\mathrm{eff}}^2$.

**Definition B.2** (ESD for RKHS Regression). *The Effective Span Dimension (ESD) of $f^*$ with respect to the kernel $\mathbf{k}$ and the effective noise variance $\sigma_{\mathrm{eff}}^2 = (\sigma_0^2 + \|f^*\|_\infty^2)/n$ is defined as*

$$d^\dagger = d^\dagger(\sigma_{\mathrm{eff}}^2; f^*, \mathbf{k}) = \min\{k \in \mathbb{N}_+ \cup \{\infty\} : \mathbf{H}_{\boldsymbol{\theta}^*, \boldsymbol{\lambda}}(k) \leq \sigma_{\mathrm{eff}}^2\}, \tag{26}$$

*where $\boldsymbol{\theta}^* = \{\theta_j^*\}_{j \geq 1}$ and $\mathbf{H}_{\boldsymbol{\theta}^*, \boldsymbol{\lambda}}(k)$ is defined as in Equation* (10).

The risk of the KPCPE estimator can be bounded using this ESD.

**Proposition B.3** (Optimal KPCPE Risk Bound). *Let $\hat{f}_k^{\mathrm{PC}}$ be the KPCPE estimator defined in Equation* (22). *Let $\mathcal{R}_*^{\mathrm{PC}} = \min_{k \geq 1} \mathcal{R}(\hat{f}_k^{\mathrm{PC}}; f^*)$. Under Assumption B.1, it holds that:*

$$(d^\dagger - 1)\frac{\sigma_0^2}{n} \leq \mathcal{R}_*^{\mathrm{PC}} \leq 2d^\dagger\sigma^2 = 2d^\dagger\frac{\sigma_0^2 + \|f^*\|_\infty^2}{n}, \tag{27}$$

*where $d^\dagger = d^\dagger(\sigma_{\mathrm{eff}}^2; f^*, \mathbf{k})$ is the ESD from Definition B.2. In particular, if $\|f^*\|_\infty^2 \lesssim \sigma_0^2$, we can conclude that $\mathcal{R}_*^{\mathrm{PC}} \asymp d^\dagger\sigma_0^2/n$.*

We comment that Zhang et al. (2023) have established the minimax optimality of the well-tuned PC estimator. Proposition B.3 suggests that the risk of the well-tuned PC estimator scales as $d^\dagger/n$. Therefore, we essentially express the minimax rate therein using the ESD without reliance on the classical eigen-decay conditions or source conditions.

We can also extend the minimax framework in Section 4 to RKHS regression. Let $K$ be a quota on the ESD, and let $C_0$ be a constant. Define the class based on the span profile as follows:

$$\mathcal{F}_{K,\mathbf{k}}^{(n)} = \{f^* \in L^2(\mathcal{X}, \mu) \cap L^\infty(\mathcal{X}, \mu) : \|f^*\|_\infty \leq \sigma_0 C_0, \quad d^\dagger(\bar{\sigma}^2/n; f^*, \mathbf{k}) \leq K\}, \tag{28}$$

where $\bar{\sigma}^2 = \sigma_0^2(1 + C_0^2)$. We further impose the following assumption on the spectrum.

**Assumption B.4.** *The kernel $\mathbf{k}$ is said to be $(K, n)$-regular if there are some constants $c_1 \in (0, 1)$ and $C_1$ such that $\sum_{i \leq c_1 K} \lambda_{\pi_i}^{-1} \leq C_1 n$.*

**Theorem B.5** (Minimax Risk over Span Profile Classes). *If $\mathbf{k}$ is $(K, n)$-regular, then the minimax risk over $\mathcal{F}_{K,\mathbf{k}}^{(n)}$ satisfies:*

$$\inf_{\hat{f}} \sup_{f^* \in \mathcal{F}_{K,\mathbf{k}}^{(n)}} \mathcal{R}(\hat{f}; f^*) \asymp \frac{\sigma_0^2 K}{n}, \tag{29}$$

*where the infimum is over all estimators $\hat{f}$.*

Combining Theorem B.5 and Proposition B.3, the optimally tuned KPCPE estimator is minimax rate optimal over $\mathcal{F}_{K,\mathbf{k}}^{(n)}$, with rate $\sigma_0^2 K/n$.

The KPCPE serves as a simple benchmark for spectral methods. This analysis, via the ESD, characterizes the performance of the optimally tuned KPCPE based directly on the properties of the specific signal $f^*$ (via $\theta^*$) and kernel spectrum $\boldsymbol{\lambda}$, without requiring standard assumptions like source conditions or polynomial eigenvalue decay. Therefore, we consider the ESD evaluated at the design-adjusted noise level $\sigma_{\text{eff}}^2$ as a key measure of statistical complexity in RKHS regression. In summary, the span profile framework provides a unified perspective on the generalization performance of spectral methods across a variety of models.

**Minimax convergence rates.** Following the framework in Section 4, we can quantify a class of populations using a quota sequence $\boldsymbol{K} = \{K_n\}_{n=1}^{\infty}$. For some $n_0 \in \mathbb{N}_+$, define

$$\mathcal{F}_{\boldsymbol{K},\mathbf{k}} = \{f^* \in L^2(\mathcal{X}, \mu) \cap L^{\infty}(\mathcal{X}, \mu) : \|f^*\|_{\infty} \le \sigma_0 C_0, \quad d^{\dagger}(\bar{\sigma}^2/n; f^*, \mathbf{k}) \le K_n, \forall n \ge n_0\}, \tag{30}$$

where $\bar{\sigma}^2 = \sigma_0^2(1 + C_0^2)$. For a sample $\{(x_i, y_i)\}_{i=1}^n$ drawn from the model in Equation (20) and any estimator $\hat{f}$, we aim to determine the optimal convergence rate of the following minimax risk:

$$\inf_{\hat{f}} \sup_{f^* \in \mathcal{F}_{\boldsymbol{K},\mathbf{k}}} \mathcal{R}(\hat{f}, f^*). \tag{31}$$

We have the following result.

**Theorem B.6.** *Suppose Condition 4.1 holds for a quota sequence $\boldsymbol{K} = \{K_n\}_{n=1}^{\infty}$. Furthermore, suppose $\mathbf{k}$ is $(K_n, n)$-regular for all $n \ge n_0$. If $\{(x_i, y_i)\}_{i=1}^n$ is drawn from the model in Equation (20), it holds that*

$$\inf_{\hat{f}} \sup_{f^* \in \mathcal{F}_{\boldsymbol{K},\mathbf{k}}} \mathcal{R}(\hat{f}, f^*) \asymp \bar{\sigma}^2 \frac{K_n}{n}.$$

Assumption B.4 is a mild condition. The following is an example where we use Theorem B.6 to recover the minimax convergence rate derived under the classical polynomial eigen-decay conditions and source conditions.

**Example B.7.** *Suppose $\mathbf{k}$ admits spectrum such that $\lambda_{\pi_i} \asymp i^{-\beta}$ with $\beta > 0$. Let $K_n = \lfloor n^{\frac{1}{1+s\beta}} \rfloor$ for any $s \ge 1$. It is easy to see that*

$$\sum_{i \le c_1 K_n} \lambda_{\pi_i}^{-1} \asymp (c_1 K_n)^{\beta+1} \asymp n^{\frac{\beta+1}{s\beta+1}} \lesssim n.$$

*Therefore, the minimax optimal rate for the class $\mathcal{F}_{\boldsymbol{K},\mathbf{k}}$ is $\frac{\bar{\sigma}^2 K_n}{n} \asymp \sigma_0^2 n^{-\frac{s\beta}{1+s\beta}}$, which is the same as the optimal rate given by the source condition with smoothness parameter $s$.*

**Remark B.8.** *In Section 2, we simplify our discussion by assuming that $\mathbf{k}$ is positive definite. In practice, positive semi-definite (PSD) kernels may also be used. In the case where $\mathbf{k}$ has rank $d < \infty$, spectral algorithms inevitably induce a systematic (squared) bias*

$$\Delta_{f^*,\mathbf{k}} = \|f^*\|_{L^2(\mu)}^2 - \sum_{j \in [d]} (\theta_{\pi_i}^*)^2,$$

*regardless the regularization parameter. In that case, we may modify the definition of ESD by adding the systematic bias into the summation to the sum of squared tails.*

*Specifically, for a PSD kernel $\mathbf{k}$ of rank $d < \infty$, we modify the definition of ESD in Definition B.2 as*

$$\bar{d}^{\dagger} = \bar{d}^{\dagger}(\sigma_{\text{eff}}^2; f^*, \mathbf{k}) = \min \left\{ j \in [d] : \frac{1}{j} \left[ \Delta_{f^*,\mathbf{k}} + \sum_{i=j+1}^{d} (\theta_{\pi_i}^*)^2 \right] \le \sigma_{\text{eff}}^2 \right\}.$$

*Again, $\sigma_{\text{eff}}^2 \bar{d}^{\dagger}$ characterizes the risk of the well-tuned PC estimator as in Proposition B.3*

### B.3 CONNECTION TO RANDOM DESIGN LINEAR REGRESSION

The analysis in this section also covers an important case: random-design linear regression (as opposed to fixed-design linear regression). This is because linear regression can be viewed as an RKHS regression w.r.t. any positive-definite linear kernels $\mathbf{k}(x, x') = x^\top \mathbf{K} x'$ for $x, x' \in \mathbb{R}^p$, where $\mathbf{K} \in \mathbb{R}^{p \times p}$ is positive definite.

Let the support $\mathcal{X}$ be a compact subset of $\mathbb{R}^p$. Suppose $\boldsymbol{\Sigma}_x = \mathbb{E}_{x \sim \mu}(xx^\top)$ is positive definite and $\mathbf{L}$ is a symmetric square root of $\boldsymbol{\Sigma}_x$. Let the eigen-decomposition of $\mathbf{LKL}$ be

$$\mathbf{LKL} = \sum_{j=1}^{p} \lambda_j v_j v_j^\top = \mathbf{V\Lambda V}^\top,$$

where $\mathbf{V}$ is the matrix with columns formed by $v_j$ and $\boldsymbol{\Lambda} = \mathrm{Diag}(\lambda_1, \ldots, \lambda_p)$.

Define $\boldsymbol{\Psi} = \mathbf{L}^{-1}\mathbf{V}$. Then $\boldsymbol{\Psi}^\top \boldsymbol{\Sigma}_x \boldsymbol{\Psi} = \mathbf{V}^\top \mathbf{V} = \mathbf{I}_p$. Furthermore, we have

$$\mathbf{K} = \mathbf{L}^{-1}\mathbf{V\Lambda V}^\top \mathbf{L}^{-1} = \boldsymbol{\Psi\Lambda\Psi}^\top,$$

Suppose the columns of $\boldsymbol{\Psi}$ are $\psi_j$. We can then write

$$\mathbb{E}_{x \sim \mu}\left[\langle \psi_j, x \rangle \langle \psi_j, x \rangle\right] = \left(\boldsymbol{\Psi}^\top \boldsymbol{\Sigma}_x \boldsymbol{\Psi}\right)_{jk} = \mathbf{1}_{\{j=k\}}, \quad \forall j, k \in [p],$$

so $\{\psi_j\}$ is an orthonormal system in $L^2(\mathcal{X}, \mu)$. Furthermore, the kernel can be expressed as

$$\mathbf{k}(x, x') = x^\top \mathbf{K} x' = \sum_{j=1}^{p} \lambda_j \langle \psi_j, x \rangle \langle \psi_j, x' \rangle.$$

Hence, $(\{\lambda_i\}, \{\psi_j\})$ is the eigenpair for $\mathbf{k}$.

For linear regression where $y = f^*(x) + \epsilon$ and $f^*(x) = \langle \beta^*, x \rangle$. Define $\theta^* = \boldsymbol{\Psi}^{-1}\beta^* = \mathbf{V}^\top \mathbf{L}\beta^*$. We can write

$$f^*(x) = \langle \beta^*, x \rangle = \langle \boldsymbol{\Psi}^{-1}\beta^*, \boldsymbol{\Psi}^\top x \rangle = \sum_{j=1}^{p} \theta_j^* \langle \psi_j, x \rangle.$$

It is also clear that $\|f^*\|_\infty \leq \sup_{x \in \mathcal{X}} \langle \beta^*, x \rangle \leq \|\beta^*\|_2 C_\mathcal{X} < \infty$, where $C_\mathcal{X}$ is finite and depends on $\mathcal{X}$. We can define the effective noise level $\sigma_{\mathrm{eff}}^2 = n^{-1}(\sigma_0^2 + \|f^*\|_\infty^2)$.

Therefore, with respect to the kernel $\mathbf{k}$ and the basis $\{\psi_j\}$, we define the ESD exactly as in Definition B.2 using the coefficients $\{\theta_j^*\}$ and eigenvalues $\{\lambda_j\}$.

### B.4 NUMERICAL ILLUSTRATION

This section provides numerical validation of the relationship between the ESD and the optimally tuned KPCPE risk, mirroring the setup for linear models in Appendix A.4. We use the cosine basis eigenfunctions $\psi_j(x) = \sqrt{2}\cos(2\pi j x)$ on the domain $[0, 1]$ with inputs sampled as $x_i \overset{\text{i.i.d.}}{\sim} \mathrm{Unif}[0, 1]$. The sample size is fixed at $n = 400$, and for numerical purpose, we consider the first $J = 800$ eigenfunctions. The noise variance is set as $\sigma_0^2 = 1$.

**Experimental Setup:** We set the baseline kernel eigenvalue spectrum as $\lambda_{j,0} = j^{-1.1}$ and the fixed signal coefficients as $\theta_j^* = j^{-4}$. To study the impact of misalignment between the kernel spectrum and the signal, we introduce a severity parameter $\alpha \geq 0$ and define the modified eigenvalue spectrum as

$$\lambda_j(\alpha) = \lambda_{j,0}\exp\left(\alpha t_j\right), \quad t_j = \frac{j-1}{D-1} \text{ for } j \leq D, \text{ and } t_j = 0 \text{ otherwise},$$

with $D = 80$. As $\alpha$ increases, the leading $D$ eigenvalues become progressively magnified, with the largest index having the most significant increase. Consequently, the modified kernel places more emphasis on directions that receive less signal energy, so the optimal KPCPE selects more principal components.

As the severity parameter $\alpha$ grows, only the first $D$ eigenvalues are changed while the rest of the spectrum is untouched. Among the changed ones, the leading eigenvalues are magnified by a smaller

constant, so that the resulting kernel has its leading subspaces being on the directions in which the signal has less of its energy and thus increases the misalignment.

For each $\alpha$ in a specified grid, we compute two quantities:

- The Effective Span Dimension

$$d_{\text{eff}}^{\dagger}(\alpha) = d^{\dagger}\left(\sigma_{\text{eff}}^2; f^*, \lambda(\alpha)\right), \quad \sigma_{\text{eff}}^2 := \frac{\sigma_0^2 + \|f^*\|_\infty^2}{n},$$

where $\|f^*\|_\infty = \operatorname{ess\,sup}_x |f^*(x)|$ is approximated by evaluating $f^*$ on a dense grid of input points and taking the maximum absolute value.

- The optimally tuned KPCPE risk

$$\mathcal{R}_*(\alpha) = \min_k \mathbb{E}\|\hat{f}_k^{\text{PC}}(\alpha) - f^*\|_{L^2(\mu)}^2,$$

where the estimator $\hat{f}_k^{\text{PC}}(\alpha)$ is computed using the spectrum $\lambda(\alpha)$ and the expectation is estimated by averaging prediction error over $B = 10$ Monte Carlo replications.

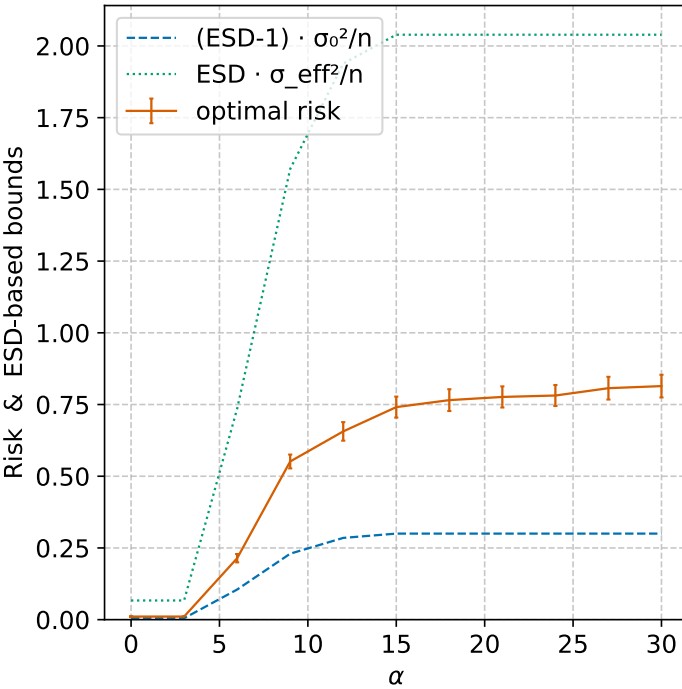

Figure 4: Effective Span Dimension and Optimal KPCPE risk. The dashed line plots Risk $\times n/\sigma_0^2$; the solid line is $d^{\dagger}(\alpha)$. The risk is computed based on 20 replications and the error bar represents the standard deviation.

Figure 4 plots the empirically computed optimal KPCPE risk (orange solid line) alongside the theoretical lower bound $(d^{\dagger} - 1)\sigma_0^2/n$ (blue dashed line) and upper bound $2d^{\dagger}\sigma_{\text{eff}}^2$ (green dotted line). The empirical risk consistently lies between the two theoretical curves, confirming the validity of the bounds derived in Proposition B.3.

As the severity parameter $\alpha$ increases, the resulting spectral perturbation shifts energy into higher-index eigenfunctions. This inflates the ESD and consequently the minimal achievable risk.

Overall, this experiment demonstrates that our span profile framework provides an accurate and robust characterization of generalization performance in RKHS regression, consistent with earlier observations made for the sequence and linear regression models.

## C MEASURING ALIGNMENT VIA ESD

We illustrate how the notion of ESD can be used to measure the alignment between the signal and the kernel.

### C.1 AN EXAMPLE COMPARING SIGNAL-SPECTRUM ALIGNMENT

The following simple example illustrates how to compare signal-spectrum alignment across the spectra discussed in Section 3.2.

Suppose $\boldsymbol{\theta}^*$ is $s$-sparse with support $S \subset [d]$ and $s = |S| \ll d$. Consider the following two spectra with the same set of eigenvalues but different allocations:

(1) The $k$ largest eigenvalues of $\boldsymbol{\lambda}^{(1)}$ are located on $S$;
(2) The $k$ largest eigenvalues of $\boldsymbol{\lambda}^{(2)}$ are located on $S^c = [d] \setminus S$.

Intuitively, $\boldsymbol{\lambda}^{(1)}$ aligns better with $\boldsymbol{\theta}^*$ than $\boldsymbol{\lambda}^{(2)}$. However, a quantitative analysis is not straightforward without using the notion of ESD.

First, we note that the effective dimensions (Zhang, 2005) is the same for both spectra because they share the same sets of eigenvalues. Similarly, the covariance-splitting index $k^*$ (Bartlett et al., 2020a) is the same for both spectra. Thus, these signal-agnostic complexity measures do not distinguish signal-spectrum alignment between the two spectra.

Next, we consider the ESD and the span profile. Rigorously, we can show for any $\tau$,

$$\mathbf{D}_{\boldsymbol{\theta}^*, \boldsymbol{\lambda}^{(1)}}(\tau) \leq s, \text{ and } \mathbf{D}_{\boldsymbol{\theta}^*, \boldsymbol{\lambda}^{(2)}}(\tau) \geq \min\left(d - s, \|\boldsymbol{\theta}^*\|^2/\tau\right).$$

Hence, for sufficiently small $\tau$, their ratio

$$r(\tau) = \mathbf{D}_{\boldsymbol{\theta}^*, \boldsymbol{\lambda}^{(1)}}(\tau)/\mathbf{D}_{\boldsymbol{\theta}^*, \boldsymbol{\lambda}^{(2)}}(\tau) \leq s/(d - s) \ll 1.$$

In view of Theorem 3.3, this suggests that the minimax risk under $\boldsymbol{\lambda}^{(1)}$ is substantially lower than under $\boldsymbol{\lambda}^{(2)}$ when the noise level is small. Therefore, spectral estimators using $\boldsymbol{\lambda}^{(1)}$ are preferred.

### C.2 PATHWISE ESD FOR LEARNED KERNELS

Section 5 analyzes eigenvalue learning because OP-GF admits tractable dynamics under a fixed eigenbasis. This is a limitation of that specific analysis, not of the ESD concept. In fact, ESD applies to general representation learning. We illustrate how decreases in ESD explain minimax risk reduction for learned kernels, whether adaptation acts through eigenvalues, eigenfunctions, or both.

Let $\mathbf{k}_t$ be the kernel learned at training time $t$, with eigenvalues $\{\lambda_j(t)\}$ (sorted in decreasing order) and eigenfunctions $\{\psi_j^{(t)}\}$ that are orthonormal in $L^2(\mathcal{X}, \mu)$. To understand how the signal-kernel alignment evolves, we define the pathwise ESD as

$$d^\dagger(t) := d^\dagger(\sigma^2; f^*, \mathbf{k}_t), t \geq 0,$$

where we have followed Definition B.2 to define $d^\dagger(\sigma^2; f^*, \mathbf{k}_t)$ as the ESD of $f^*$ w.r.t. the kernel $\mathbf{k}_t$ using $\theta_j^{*,(t)} = \langle f^*, \psi_j^{(t)} \rangle$ and $\sigma^2 = n^{-1}(\sigma_0^2 + \|f^*\|_\infty^2)$.

Let $\mathbf{H}_t(k) := \frac{1}{k} \sum_{i>k} \left[\theta_i^{*,(t)}\right]^2$. If training aligns the leading eigenfunctions $\psi_j^{(t)}$ better with $f^*$, then $\{\theta_j^{*,(t)}\}$ concentrate more on the leading indices, and thus $\mathbf{H}_t(k)$ decreases for all $k$, which implies a decrease in $d^\dagger(t)$.

**Experiment on Deep Linear Networks.** To demonstrate this pathwise perspective, we simulate a random-design linear regression.

Each covariate coordinate is drawn independently from $\{\pm 1\}$, so $\boldsymbol{\Sigma}_x = \mathbb{E}(XX^\top) = \mathbf{I}_p$ and $\|X\|_\infty = 1$. We set $p = 900$, and specify the true parameters as follows: $\beta^*$ follows a power-law

decay with $\beta_j^* = j^{-1.1}$ for $1 \leq j \leq 200$ and $\beta_j^* = 0$ for $j > 200$. The response is $Y = \langle \beta^*, X \rangle + \varepsilon$ with $\varepsilon \sim N(0, \sigma_0^2)$ and $\sigma_0 = 0.1$.

We draw $n = 1000$ samples and train a deep *linear network* with $D = 4$ hidden affine layers without bias using full-batch Adam with learning rate $10^{-4}$. The hidden weight matrices of the network are $\mathbf{W}_\ell(t) \in \mathbb{R}^{p \times p}$ for $\ell = 1, \ldots, D$ (using a near-identity initialization), and the weight of the final linear layer is $w(t) \in \mathbb{R}^p$.

The estimated function at time $t$ is given by $f_t(x) = w(t)^\top \mathbf{A}(t)x$, where $\mathbf{A}(t) := \mathbf{W}_D(t) \cdots \mathbf{W}_1(t)$. We form the learned kernel $\mathbf{k}_t(x, x') = \langle \mathbf{A}(t)x, \mathbf{A}(t)x' \rangle = x^\top \mathbf{G}_t x'$, where $G_t := \mathbf{A}(t)^\top \mathbf{A}(t)$. We then follow the derivation in Appendix B.3 and define the ESD $d^\dagger(t)$ of $f^*$ w.r.t. the kernel $\mathbf{k}_t$. Since $\|X\|_\infty = 1$ $\mu$-a.s., we have $\|f^*\|_\infty^2 = \|\beta^*\|_1^2$; this is used in computing the effective noise level.

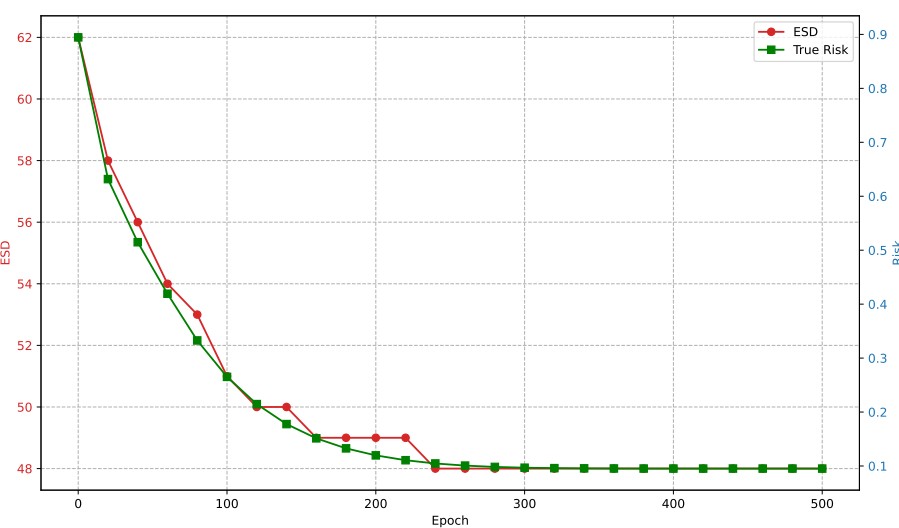

Figure 5: Pathwise ESD and risk under a learned kernel using a 4-layer linear network.

Figure 5 shows that adaptive representation learning progressively reduces the ESD $d^\dagger(t)$ over training time, along with the true risk. This confirms that ESD captures the evolving alignment between signal and kernel.

## D PROOF

### D.1 PROOFS OF RESULTS ON ESD OF SEQUENCE MODELS

*Proof of Theorem 3.2.* For any $\nu > 0$, define

$$k_\Lambda(\nu) = \#\{\, j : \lambda_j \geq \nu \,\},$$

which counts how many eigenvalues exceed the threshold $\nu$. The KPCR estimator sets

$$\hat{\theta}_i^\nu = \mathbf{1}_{\{\lambda_i \geq \nu\}} \, z_i, \ i \in [d].$$

Its squared bias and variance are given by

$$B^{\mathrm{PC}}(v) = \sum_{i:\lambda_i < v} (\theta_i^*)^2, \quad V^{\mathrm{PC}}(v) = \sum_{i:\lambda_i \geq v} \sigma^2 = k_\Lambda(v) \, \sigma^2.$$

For any threshold $v$, we can reparameterize the bias and variance using $k = k_\Lambda(v)$ as

$$B^{\mathrm{PC}}(k) = \sum_{i=k+1}^{d} (\theta_{\pi_i}^*)^2, \quad \text{and} \quad V^{\mathrm{PC}}(k) = k \, \sigma^2, k = 0, 1, \ldots, d.$$

The function $B^{\mathrm{PC}}(k)$ decreases in $k$, while $V^{\mathrm{PC}}(k)$ increases in $k$. The risk function is given by $\mathcal{R}^{\mathrm{PC}}(k) = B^{\mathrm{PC}}(k) + V^{\mathrm{PC}}(k)$.

For any integer $k \geq 1$, we have

$$\mathcal{R}^{\mathrm{PC}}(k) \;=\; k \left( \sigma^2 \;+\; \frac{1}{k} \sum_{i:\lambda_i < \lambda_{\pi_k}} (\theta_i^*)^2 \right).$$

**Upper bound**   For $k = d^\dagger$, we have $\frac{1}{k} \sum_{i:\lambda_i < \lambda_{\pi_k}} (\theta_i^*)^2 \leq \sigma^2$. By definition of the optimal risk, we have

$$\mathcal{R}_*^{\mathrm{PC}} \;\leq\; \mathcal{R}^{\mathrm{PC}}(d^\dagger) \;\leq\; 2\, d^\dagger\, \sigma^2.$$

Therefore, the upper bound is proved.

**Lower bound**   Without loss of generality, assume $d^\dagger \geq 2$. For any $k \leq d^\dagger - 1$, we have

$$\mathcal{R}^{\mathrm{PC}}(k) \geq B^{\mathrm{PC}}(k) = \sum_{i=k+1}^{d} \left( \theta_{\pi_i}^* \right)^2 \geq \sum_{i=d^\dagger}^{d} \left( \theta_{\pi_i}^* \right)^2 > (d^\dagger - 1)\sigma^2,$$

where the last inequality comes from the definition of $d^\dagger$. For any $k \in [d^\dagger, d]$, we have

$$\mathcal{R}^{\mathrm{PC}}(k) \geq k\sigma^2 \geq d^\dagger \sigma^2.$$

Therefore, the lower bound is proved. $\qquad\qquad\square$

## D.2   PROOF ON MINIMAX RESULTS

*Proof of Theorem 4.3.*   Throughout the proof, the quota sequence is fixed. Recall the definition of $M_k$ in Condition 4.1. Define $\psi(k) = \sigma_0^2 \frac{k}{M_k}$ for any $k \in \bar{K}$. Also define $\mathbf{S}_{\boldsymbol{\theta},\boldsymbol{\lambda}}(k) = k\mathbf{H}_{\boldsymbol{\theta},\boldsymbol{\lambda}}(k)$.

We can express $\mathcal{F}_{\boldsymbol{K},\boldsymbol{\lambda}}$ as follows.

**Lemma D.1.**   *Under Condition 4.1, we have*

$$\mathcal{F}_{\boldsymbol{K},\boldsymbol{\lambda}} \;=\; \left\{ \boldsymbol{\theta} \in \mathbb{R}^\infty : \mathbf{S}_{\boldsymbol{\theta},\boldsymbol{\lambda}}(k) \leq \psi(k) \text{ for all } k \in [\bar{K}] \right\}.$$

*Proof of Lemma D.1.*   Observe the relation that

$$\boldsymbol{\theta} \in \mathcal{F}_{\boldsymbol{K},\boldsymbol{\lambda}} \iff \mathbf{D}_{\boldsymbol{\theta},\boldsymbol{\lambda}}\!\left( \frac{\sigma_0^2}{n} \right) \leq K_n, \;\; \forall n \geq 1 \iff \mathbf{S}_{\boldsymbol{\theta},\boldsymbol{\lambda}}(K_n) \leq \sigma_0^2 \frac{K_n}{n}, \;\; \forall n \geq 1.$$

By (1) of Condition 4.1, we have

$$\mathbf{S}_{\boldsymbol{\theta},\boldsymbol{\lambda}}(K_n) \leq \sigma_0^2 \frac{K_n}{n}, \;\; \forall n \geq 1 \iff \mathbf{S}_{\boldsymbol{\theta},\boldsymbol{\lambda}}(k) \leq \psi(k) \text{ for all } k \in [\bar{K}].$$

Therefore, we can rewrite

$$\mathcal{F}_{\boldsymbol{K},\boldsymbol{\lambda}} \;=\; \left\{ \boldsymbol{\theta} \in \mathbb{R}^\infty : \mathbf{S}_{\boldsymbol{\theta},\boldsymbol{\lambda}}(k) \leq \psi(k) \text{ for all } k \in [\bar{K}] \right\}.$$

$\qquad\qquad\square$

Fix any $n$. Define $\delta = \sqrt{c\sigma_0^2/n}$ with the constant $c = \frac{1}{4} \wedge \tau$, where $\tau$ comes from Condition 4.1. Consider assigning nonzero signals on the block $B_n = \{\pi_1, \ldots, \pi_{K_n}\}$ to construct a subset of populations.

Specifically, we define the collection of hypercubes vertices $\mathcal{V} = \{-1, 1\}^{K_n}$. For every vertex $v \in \mathcal{V}$, define a parameter vector $\boldsymbol{\theta}^{(v)} = (\theta_j^{(v)})_{j=1}^d$ as follows:

$$\theta_{\pi_i}^{(v)} = \delta\, v_i, \text{ for } i = 1, \ldots, K_n, \qquad \text{and } \theta_j^{(v)} = 0 \text{ for } j \notin B_n. \tag{32}$$

There are $2^{K_n}$ such vectors $\{\boldsymbol{\theta}^{(v)}\}$, and they satisfy the following property.

**Lemma D.2.** *For any $v \in \mathcal{V}$, the parameter vector $\boldsymbol{\theta}^{(v)}$ constructed in Equation (32) lies in $\mathcal{F}_{K,\lambda}$.*

*Proof of Lemma D.2.* For any $k \in [\bar{K}]$, if $k \geq K_n$, then $\mathbf{S}_{\boldsymbol{\theta}^{(v)},\boldsymbol{\lambda}}(k)$ is 0.

If $1 \leq k \leq K_n - 1$, then $\mathbf{S}_{\boldsymbol{\theta}^{(v)},\boldsymbol{\lambda}}(k) \leq K_n \delta^2 = c\sigma_0^2 K_n/n$. Denote $k_0 = K_n - 1$ and $L = 1 + M_{k_0}$. By definition of $M_{k_0}$, we have $n \geq L$. Since $k \leq k_0$, we have $L \geq 1 + M_k$. We have

$$
\begin{aligned}
\frac{K_n}{n} &\leq \frac{K_n}{L} \\
&= \frac{k_0 + 1}{1 + M_{k_0}} \\
&\leq 2\frac{k_0}{M_{k_0}} \\
&\leq 2\frac{k}{M_k},
\end{aligned}
\tag{33}
$$

where the second last inequality is because $(1 + k_0)/(1 + m) \leq 2k_0/m \Leftrightarrow m + k_0 m \leq 2k_0 + 2k_0 m$ and the last inequality is due to (2) of Condition 4.1. Since $2c < 1$, we see that $\sigma_0^2 c K_n/n \leq \psi(k)$ for all $k < K_n$.

In either case, we have $\mathbf{S}_{\boldsymbol{\theta}^{(v)},\boldsymbol{\lambda}}(k) \leq \psi(k)$ for all $k \in [\bar{K}]$, and thus $\boldsymbol{\theta}^{(v)} \in \mathcal{F}_{K,\lambda}$. $\qquad\square$

For each $v \in \mathcal{V}$, let $P_v$ be the sampling distribution of the sequence model in Equation (4) with $\boldsymbol{\theta}^* = \boldsymbol{\theta}^{(v)}$, $\sigma^2 = \sigma_0^2/n$, and $\{\xi_j\}_{j\in[d]}$ being i.i.d. from $N(0, \sigma^2)$. Let $\rho$ be Hamming distance on $\mathcal{V}$. If $v$ and $w \in \mathcal{V}$ differ in exactly one coordinate (i.e., $\rho(v, w) = 1$), then

- $\|\boldsymbol{\theta}^{(v)} - \boldsymbol{\theta}^{(w)}\|^2 \geq (2\delta)^2$, and

- the Kullback-Leibler divergence between $P_v$ and $P_w$ satisfies $\mathrm{KL}(P_v \| P_w) = \frac{1}{2\sigma^2}(2\delta)^2 = 2c \leq \frac{1}{2}$, and by the Pinsker's inequality, $\|P_v \wedge P_w\| = 1 - \mathrm{TV}(P_v, P_w) \geq 1 - \sqrt{\mathrm{KL}(P_v \| P_w)/2} \geq 1/2$.

By Assouad's Lemma (Lemma 2 in Yu (1997)), for any estimator $\widehat{\boldsymbol{\theta}}$ based on a sample $Y^{(n)}$ drawn from $P_v$, we have

$$
\sup_{v\in\mathcal{V}} \mathbb{E}_v \|\widehat{\boldsymbol{\theta}} - \boldsymbol{\theta}^{(v)}\|^2 \geq K_n \frac{(2\delta)^2}{4} = c\sigma_0^2 \frac{K_n}{n}.
$$

$\qquad\square$

*Proof of Theorem 3.3.* The upper bound is given by Theorem 3.2, so we only need to prove the lower bound. The main idea is the same as the proof for Theorem 4.3.

Let $\delta = \sqrt{c\sigma_0^2/n}$ with the constant $c = \frac{1}{4}$. Let $B_n = \{\pi_1, \ldots, \pi_{K_n}\}$. We define the collection of hypercubes vertices $\mathcal{V} = \{-1, 1\}^K$. For every vertex $v \in \mathcal{V}$, define a parameter vector $\boldsymbol{\theta}^{(v)} = (\theta_j^{(v)})_{j=1}^d$ as in Equation (32). There are $2^K$ such vectors $\{\boldsymbol{\theta}^{(v)}\}$. For each $v \in \mathcal{V}$, let $P_v$ be the sampling distribution of the sequence model in Equation (4) with $\boldsymbol{\theta}^* = \boldsymbol{\theta}^{(v)}$, $\sigma^2 = \sigma_0^2/n$, and $\{\xi_j\}_{j\in[d]}$ being i.i.d. normal. Let $\rho$ be Hamming distance on $\mathcal{V}$. The rest of the proof is identical to that of Theorem 4.3 and is omitted. $\qquad\square$

## D.3 DETAILS OF EXAMPLES IN EQUATION (9)

We provide the details of Equation (9) for illustration of the concepts of ESD and span profile through several examples.

**Example D.3** (Polynomial spectrum with source condition). *Assume $\lambda_i = i^{-\beta}$ for some $\beta > 0$ and the source condition $\sum_{i=1}^{d} \lambda_i^{-s} \theta_i^{*2} \leq R$ with $s > 0$. The trade-off function satisfies*

$$\mathbf{H}_{\boldsymbol{\theta}^*, \boldsymbol{\lambda}}(k) = \frac{1}{k} \sum_{i>k} \theta_i^{*2} \leq \frac{\lambda_k^s}{k} \sum_{i>k} \lambda_i^{-s} \theta_i^{*2} \leq R\, k^{-(1+s\beta)},$$

*which follows that $\mathbf{D}_{\boldsymbol{\theta}^*, \boldsymbol{\lambda}}(\sigma^2) \lesssim [\sigma^2]^{-\frac{1}{1+s\beta}}$. Since $\mathbf{D}_{\boldsymbol{\theta}^*, \boldsymbol{\lambda}}(\sigma^2) \leq d$, the optimal risk of PC estimator satisfies*

$$\mathcal{R}_*^{\mathrm{PC}} \lesssim \min\left( [\sigma^2]^{\frac{s\beta}{1+s\beta}}, d\sigma^2 \right).$$

In Example D.3, we note that for $\sigma^2 = \sigma_0^2/n$, the upper bound becomes $\sigma_0^2 \min\left( n^{-\frac{s\beta}{1+s\beta}}, d/n \right)$. When $d = \infty$, this upper bound matches the well-known optimal rate under the source condition and the polynomial eigen-decay condition. When $d < \infty$, there is a phase transition around $d_0 \asymp n^{\frac{1}{1+s\beta}}$: if $d \lesssim d_0$, the upper bound is $d\sigma_0^2/n$; if $d \gtrsim d_0$, the upper bound is the same as if $d = \infty$. Using the span profile, we can extend classical results to finite-dimensional models and reveal new phenomena.

**Example D.4** (Polynomial signals ($\alpha > 1$)). *Suppose $\theta_i^* = i^{-\alpha/2}$ for some constant $\alpha > 1$, and $\{\lambda_i\}_1^d$ are decreasing. By an integral approximation, we can get $\mathbf{H}_{\boldsymbol{\theta}^*, \boldsymbol{\lambda}}(k) \leq \frac{1}{\alpha-1} k^{-\alpha}$. Therefore, we have $\mathbf{D}_{\boldsymbol{\theta}^*, \boldsymbol{\lambda}}(\sigma^2) \lesssim [\sigma^2]^{-\frac{1}{\alpha}}$. The optimal risk of PC estimator satisfies*

$$\mathcal{R}_*^{\mathrm{PC}} \leq 2\sigma^2 \mathbf{D}_{\boldsymbol{\theta}^*, \boldsymbol{\lambda}}(\sigma^2) \lesssim \min\left( [\sigma^2]^{1-\frac{1}{\alpha}}, d\sigma^2 \right).$$

**Example D.5** (Polynomial signals ($\alpha = 1$)). *Suppose $d < \infty$, $\theta_i^* = i^{-1/2}$, and $\{\lambda_i\}_1^d$ are decreasing. We show in the supplementary material that for some constant $C$, if $d\sigma^2 \leq e$, then $\mathcal{R}_*^{\mathrm{PC}} \leq C d\sigma^2$, and if $d\sigma^2 > e$, then $\mathcal{R}_*^{\mathrm{PC}} \leq C \log\left( d\sigma^2 / \log\left( d\sigma^2 \right) \right)$.*

**Example D.6** (Polynomial signals ($\alpha < 1$)). *Suppose $d < \infty$, $\theta_i^* = i^{-1/2}$, and $\{\lambda_i\}$ is decreasing. We show in the supplementary material that $\mathcal{R}_*^{\mathrm{PC}} \lesssim d \min\left( d^{-\alpha}, \sigma^2 \right)$.*

These examples suggest that using our framework of span profile, we are able not only to recover classical results but also to extend it to various settings where the classical framework is inapplicable.

*Details of Example D.5.* We have $\mathbf{H}_{\boldsymbol{\theta}^*, \boldsymbol{\lambda}}(k) \leq k^{-1} \int_k^d \frac{1}{x} dx = k^{-1}\left( \log d - \log k \right)$.

By dropping the term $\log k$ in the numerator, it is easy to see that a sufficient condition for $\mathbf{H}_{\boldsymbol{\theta}^*, \boldsymbol{\lambda}}(k) \leq \sigma^2$ is given by $k \geq \sigma^{-2} \log(d)$. Therefore, we have $\mathbf{D}_{\boldsymbol{\theta}^*, \boldsymbol{\lambda}}(\sigma^2) \leq \lceil \sigma^{-2} \log(d) \rceil$.

The upper bound can be improved. Suppose $A > 1$ satisfies $d\sigma^2 \leq A \log A$. If $k \geq \sigma^{-2} \log A$, then

$$\frac{k}{d} \geq \frac{\log A}{d\sigma^2} \geq \frac{d\sigma^2/A}{d\sigma^2} = \frac{1}{A},$$

which follows that $\mathbf{H}_{\boldsymbol{\theta}^*, \boldsymbol{\lambda}}(k) \leq k^{-1} \log A \leq \sigma^2$. Therefore,

$$\mathbf{D}_{\boldsymbol{\theta}^*, \boldsymbol{\lambda}}(\sigma^2) \leq \min\left( d, \lceil \sigma^{-2} \log A \rceil \right).$$

By elementary calculus, if $y > e$, the solution to $x \log x = y$ satisfies that $x \in (e, y)$, and thus $\log x \in (1, \log(y))$, which implies $x > y/\log(y)$ and thus $x < y/\log(y/\log(y)) = y/\left( \log y - \log\log y \right) < 2y/\log(y)$.

If $d\sigma^2 \leq e$, we can take $A = e$ and conclude

$$\mathcal{R}_*^{\mathrm{PC}} \leq 2\sigma^2 \mathbf{D}_{\boldsymbol{\theta}^*, \boldsymbol{\lambda}}(\sigma^2) \lesssim d\sigma^2.$$

If $d\sigma^2 > e$, then $\log\left( d\sigma^2 \right) > 1$ and we can take $A = 2d\sigma^2/\log\left( d\sigma^2 \right)$, which implies that

$$\mathcal{R}_*^{\mathrm{PC}} \leq 2\sigma^2 \mathbf{D}_{\boldsymbol{\theta}^*, \boldsymbol{\lambda}}(\sigma^2) \lesssim \log\left( d\sigma^2 \right) - \log\left( \log\left( d\sigma^2 \right) \right).$$

$\square$

*Detail of Example D.6.* By an integral approximation, we see that

$$\mathbf{H}_{\boldsymbol{\theta}^*, \boldsymbol{\lambda}}(k) \asymp k^{-1}\left( d^{1-\alpha} - k^{1-\alpha} \right).$$

**Case 1:** $\sigma^2 d^\alpha < 2$. We have the default bound $\mathbf{D}_{\boldsymbol{\theta}^*,\boldsymbol{\lambda}}(\sigma^2) \leq d$.

**Case 2:** $\sigma^2 d^\alpha \geq 2$. If $k \geq d^{1-\alpha}/\sigma^2$, then $\mathbf{H}_{\boldsymbol{\theta}^*,\boldsymbol{\lambda}}(k) \leq \sigma^2$. Therefore, we have $\mathbf{D}_{\boldsymbol{\theta}^*,\boldsymbol{\lambda}}(\sigma^2) \leq \lceil d^{1-\alpha}/\sigma^2 \rceil$, which is not larger than $\lceil d/2 \rceil$.

Combining both cases, we have $\mathbf{D}_{\boldsymbol{\theta}^*,\boldsymbol{\lambda}}(\sigma^2) \lesssim d \min(1/(d^\alpha \sigma^2), 1)$. Multiplying by $2\sigma^2$ on both sides, we have

$$\mathcal{R}_*^{\mathrm{PC}} \leq 2\sigma^2 \mathbf{D}_{\boldsymbol{\theta}^*,\boldsymbol{\lambda}}(\sigma^2) \lesssim d \min\left(d^{-\alpha}, \sigma^2\right).$$

$\square$

### D.4    DETAIL OF EXAMPLE 4.4

Let $f(x) = \sigma_0^2 x e^{-x^b}$. Then $(\theta_{j+1}^*)^2 = f(j) - f(j+1)$ for $j \geq 1$. Since $\theta_1^* = 0$, for any $k \geq 1$, the tail sum is

$$\sum_{j=k+1}^{\infty} (\theta_j^*)^2 = \sum_{j=k}^{\infty}(f(j) - f(j+1)) = f(k) = \sigma_0^2 k e^{-k^b},$$

since $f(N) \to 0$. As $\{\lambda_j\}$ is assumed to be decreasing, the trade-off function is $\mathbf{H}_{\boldsymbol{\theta}^*,\boldsymbol{\lambda}}(k) = \frac{1}{k}\sum_{j=k+1}^{\infty}(\theta_j^*)^2 = \sigma_0^2 e^{-k^b}$.

For any $n \geq 3$, let $k = K_n$. By definition of the ceiling function, $k \geq (\log n)^{1/b}$, which implies $k^b \geq \log n$, and thus $e^{k^b} \geq n$. Then, $\mathbf{H}_{\boldsymbol{\theta}^*,\boldsymbol{\lambda}}(k) = \sigma_0^2 e^{-k^b} \leq \sigma_0^2/n$. By Proposition 3.5, we have $\mathbf{D}_{\boldsymbol{\theta}^*,\boldsymbol{\lambda}}(\sigma_0^2/n) \leq k = K_n$.

Since this holds for all sufficiently large $n$, we conclude that $\boldsymbol{\theta}^* \in \mathcal{F}_{\boldsymbol{K}}$. Theorem 4.3 guarantee the optimal convergence rate is $\Theta(\sigma_0^2 K_n/n) = \Theta(\sigma_0^2 (\log n)^{1/b}/n)$.

Lastly, we consider the standard source condition that for some $s > 0$, there is some constant $R_s$ such that

$$\sum_{j=1}^{\infty} \lambda_j^{-s} (\theta_j^*)^2 \leq R_s. \tag{34}$$

Let's assume a polynomial eigenvalue decay $\lambda_j \asymp j^{-\gamma}$ for some $\gamma > 0$. Let $S$ be the left hand side of Equation (34). Since $\theta_1^* = 0$, we have

$$S = \sum_{j=2}^{\infty}(j^{-\gamma})^{-s}(\theta_j^*)^2 = \sum_{j=2}^{\infty} j^{s\gamma}(\theta_j^*)^2$$

$$= \sum_{k=1}^{\infty}(k+1)^{s\gamma}(\theta_{k+1}^*)^2.$$

Using $(\theta_{k+1}^*)^2 = f(k) - f(k+1)$ with $f(x) = \sigma_0^2 x e^{-x^b}$:

$$S = \sum_{k=1}^{\infty}(k+1)^{s\gamma}(f(k) - f(k+1)).$$

Using summation by part, we have

$$S = (1+1)^{s\gamma} f(1) - \lim_{N\to\infty}(N+1)^{s\gamma} f(N+1) + \sum_{k=1}^{\infty}((k+2)^{s\gamma} - (k+1)^{s\gamma})f(k+1).$$

Since $\lim_{N\to\infty}(N+1)^{s\gamma} N e^{-N^b} = 0$ for $b \geq 1$, the limit term vanishes. $f(1) = \sigma_0^2 e^{-1}$. The difference term $(k+2)^{s\gamma} - (k+1)^{s\gamma} > 0$. $f(k+1) = \sigma_0^2(k+1)e^{-(k+1)^b} > 0$. The sum $\sum_{k=1}^{\infty}((k+2)^{s\gamma} - (k+1)^{s\gamma})f(k+1)$ converges because $f(k+1)$ decays faster than any polynomial grows. Specifically, $(k+2)^{s\gamma} - (k+1)^{s\gamma} \approx s\gamma k^{s\gamma-1}$, and the sum $\sum k^{s\gamma-1}(k+1)e^{-(k+1)^b}$ converges. Therefore, $S$ converges for any $s > 0$ and any $\gamma > 0$.

The classical theory predicts a rate of $n^{-\frac{s\gamma}{s\gamma+1}}$. Since the source condition holds for arbitrarily large $s$, the classical rate can be made arbitrarily close to $n^{-1}$. However, this $n^{-1}$ rate ignores the logarithmic factor $(\log n)^{1/b}$ present in the true optimal rate $\Theta(\sigma_0^2 (\log n)^{1/b}/n)$. Thus, the traditional convergence analysis based on the source condition is not sharp for this signal.

# E  PROOFS FOR RESULTS IN APPENDIX B

*Proof of Proposition B.3.* **Upper bound**: Take $k = d^\dagger$, and we have $B(k) = k\mathbf{H}_{f^*,\lambda}(k) \leq k\sigma^2$. The variance $V(k) = \sum_{j=1}^{k}(\sigma_0^2 + \tau_j^2)/n \leq k(\sigma_0^2 + \sigma_{f,4}^2)/n = k\sigma^2$. Thus $\mathcal{R}_*^{\mathrm{PC}} \leq \mathcal{R}_k = B(k) + V(k) \leq 2k\sigma^2 = 2d^\dagger\sigma^2$.

**Lower bound**: Let $k^*$ be the optimal tuning parameter. If $k^* \geq d^\dagger$, then $\mathcal{R}_* \geq d^\dagger\sigma_0^2/n$. If $k^* \leq d^\dagger - 1$, by definition of ESD, we have $\mathcal{R}_* \geq B(k^*) \geq B(d^\dagger - 1) \geq (d^\dagger - 1)\sigma^2 \geq (d^\dagger - 1)\sigma_0^2/n$.

$\square$

*Proof of Theorem B.5.* **Upper bound**: The upper bound follows the proof of the upper bound in Proposition B.3. To see this, we note that since $f^* \in \mathcal{F}_{K,\boldsymbol{\lambda},n}$, we have $\|f\|_\infty^2 \leq \sigma_0^2 C_0^2$. Therefore, $\sigma^2 \leq \bar{\sigma}^2/n$. We can then apply the argument in Proposition B.3 with $\sigma^2$ replaced by $\bar{\sigma}^2/n$.

**Lower bound**: We establish the lower bound using Assouad's method.

Let $m = \lfloor c_1 K \rfloor$. Consider the first $m$ eigenfunctions $\{\psi_{\pi_j}\}_{j\leq m}$ corresponding to the largest eigenvalues $\{\lambda_{\pi_j}\}_{j\leq m}$. Define the collection of hypercubes vertices $\mathcal{V} = \{-1, 1\}^m$. For every vertex $v \in \mathcal{V}$, define a function

$$f^{(v)}(x) = \gamma \sum_{j=1}^{m} v_j \psi_{\pi_j}(x), \tag{35}$$

where the amplitude $\gamma$ is to be chosen. Since $\mathbf{k}$ is $(K, n)$-regular, we have

$$f^{(v)}(x)^2 \leq \gamma^2 \sum_{j\leq m} \lambda_{\pi_j}^{-1} \sum_{j\leq m} \lambda_j \psi_{\pi_j}^2(x) \leq \gamma^2 C_1 n\kappa^2, \tag{36}$$

where $\kappa^2 = \sup_x \mathbf{k}(x, x) < \infty$ by assumption.

We choose

$$\gamma^2 = n^{-1} \min\left(\frac{\bar{\sigma}^2}{4(1 + C_0^2)}, \frac{\sigma_0^2 C_0^2}{C_1 \kappa^2}\right).$$

It then follows that $\|f^{(v)}\|_\infty^2 \leq \sigma_0^2 C_0^2$.

For each $v \in \mathcal{V}$, let $P_v$ be the sampling distribution of $\{z_i = (x_i, y_i)\}_{i\leq n}$ from the regression model Equation (20) with $f^* = f^{(v)}$. Let $\rho$ be the Hamming distance on $\mathcal{V}$. If $v$ and $w \in \mathcal{V}$ differ in exactly one coordinate (i.e., $\rho(v, w) = 1$), then

- $\|f^{(v)} - f^{(w)}\|_{L^2(\mu)}^2 \geq (2\gamma)^2$, and

- the Kullback-Leibler divergence between $P_v$ and $P_w$ satisfies $\mathrm{KL}\left(P_v \,\|\, P_w\right) = \frac{n}{2\sigma_0^2}(2\gamma)^2 \leq \frac{1}{2}$, where the last equation is due to the definition of the constant $c$. By the Pinsker's inequality, $\|P_v \wedge P_w\| = 1 - \mathrm{TV}(P_v, P_w) \geq 1 - \sqrt{\mathrm{KL}\left(P_v \,\|\, P_w\right)/2} = 1/2$.

By Assouad's Lemma (Lemma 2 in Yu (1997)), for any estimator $\widehat{f}$ based on a sample $\{z_i = (x_i, y_i)\}_{i\leq n}$ drawn from $P_v$, we have

$$\sup_{v\in\mathcal{V}} \mathbb{E}_v \|\widehat{\boldsymbol{\theta}} - \boldsymbol{\theta}^{(v)}\|^2 \geq m\frac{(2\gamma)^2}{4} = c\frac{\sigma_0^2 K}{n},$$

where $c$ is a constant that depends on $C_0, \kappa, c_1, C_1$.

$\square$

*Proof of Theorem B.6.* Since

$$\mathcal{F}_{\mathbf{K},\mathbf{k}} = \bigcap_{n \geq n_0} \mathcal{F}^{(n)}_{K_n,\mathbf{k}},$$

the upper bound $\bar{\sigma}^2 K_n/n$ is immediately implied by Theorem B.5.

The lower bound follows the same argument as in the proof of Theorem 4.3, but replace the construction of parameter vectors in Equation (32) by the construction of functions in Equation (35). Following the proof for Theorem 4.3, we use Condition 4.1 to ensure the constructed functions all belong to $\mathcal{F}_{\mathbf{K},\mathbf{k}}$. Then the lower bound is given using Assouad's Lemma as in the proof of Theorem B.5. Below, we provide the details for completeness.

Mercer's theorem yields

$$\mathbf{k}\left(\boldsymbol{x}, \boldsymbol{x}'\right) = \sum_{j=1}^{\infty} \lambda_j \psi_j(\boldsymbol{x}) \psi_j\left(\boldsymbol{x}'\right), \quad \boldsymbol{x}, \boldsymbol{x}' \in \mathcal{X}, \tag{37}$$

where $\{\psi_j\}_{j \geq 1}$ is a $L^2(\mathcal{X}, \mu)$-orthonormal eigenbasis. Without loss of generality, assume $\lambda_j$ is sorted decreasingly.

Fix $n$ and set $m := \lfloor c_1 K_n \rfloor$ where $c_1$ comes from Assumption B.4.

For a sign vector $v = (v_j)_{j \leq m} \in \{-1, +1\}^m$, define the sequence of coefficients as

$$\theta_j^{(v)} := \begin{cases} \gamma\, v_j, & j \leq m, \\ 0, & j > m, \end{cases} \qquad f_v(x) := \sum_{j \geq 1} \theta_j^{(v)} \psi_j(x) = \gamma \sum_{j \leq m} v_j\, \psi_j(x).$$

Since $\mathbf{k}$ is $(K_n, n)$-regular, Equation (36) holds and reads as

$$f^{(v)}(x)^2 \leq \gamma^2 C_1 n \kappa^2.$$

If $\gamma_2 C_1 n \kappa^2 \leq \sigma_0^2 C_0^2$, then $\|f^{(v)}\|_\infty^2 \leq \sigma_0^2 C_0^2$. Furthermore, if $m\gamma^2 \leq (2n)^{-1} \sigma_0^2 K_n$, we can the same argument in Lemma D.2 (in particular, using Condition 4.1 to derive Equation (33)) to show that $f^{(v)} \in \mathcal{F}_{\mathbf{K},\mathbf{k}}$.

We choose

$$\gamma^2 = n^{-1} \min\left(\sigma_0^2, \frac{\bar{\sigma}^2}{4(1 + C_0^2)}, \frac{\sigma_0^2 C_0^2}{C_1 \kappa^2}\right),$$

which implies $f^{(v)} \in \mathcal{F}_{\mathbf{K},\mathbf{k}}$.

We then follows the same argument in the proof of lower bound in Theorem B.5 to obtain

$$\sup_{v \in \mathcal{V}} \mathbb{E}_v \left\|\widehat{\boldsymbol{\theta}} - \boldsymbol{\theta}^{(v)}\right\|^2 \geq m \frac{(2\gamma)^2}{4} = c \frac{\sigma_0^2 K}{n},$$

where $c$ is a constant that depends on $C_0, \kappa, c_1, C_1$.

$\square$

## F  Proofs for Result on overparameterized gradient flow

In this section, we prove Theorem 5.2. The high-level idea is as follows: To show the ESD decreases, it is enough to show that the squared signal tail sorted by the learned eigenvalues at the new time is smaller than that at the old time. The key idea is to study how the gradient flow changes the eigenvalues depending on the signal's strength. Our analysis reveals that eigenvalues associated with the strong signal coordinates will often grow much faster than those associated with weak ones. Consequently, more of the largest learned eigenvalues correspond to the strong signals. This implies that the signal energy is concentrated in the top principal components of the learned kernel, which reduces the signal tail and thus reduces the ESD.

We first remark that for any $j \in [d]$, due to the same initialization $b_{j,k} = b_0$ for all $k$, one can prove that throughout the time $b_{j,k}$ (for all $k$) have the same value $b_j$. Therefore, we can rewrite the over-parameterization as $\theta_j = a_j b_j^D \beta_j$, and consider the following gradient flow

$$\begin{aligned}
\dot{a}_j &= -\nabla_{a_j} L_j = b_j^D \beta_j (z_j - \theta_j), \\
\dot{b}_j &= -\nabla_{b_j} L_j = D a_j b_j^{D-1} \beta_j (z_j - \theta_j), \\
\dot{\beta}_j &= -\nabla_{\beta_j} L_j = a_j b_j^D (z_j - \theta_j), \\
a_j(0) &= \lambda_j^{\frac{1}{2}} > 0, \quad \boldsymbol{b}(0) = b_0 > 0, \quad \boldsymbol{\beta}(0) = 0,
\end{aligned} \tag{38}$$

where $L_j = \frac{1}{2}(z_j - \theta_j)^2$.

## F.1 PROOF OF THEOREM 5.2

Recall that we define these sets as follows:

$$\begin{aligned}
A_1 &:= \{i : \pi_{t_1}^{-1}(i) < d^\dagger(t_1), \lambda_i < c \cdot D^{-\frac{D}{D+2}} \cdot M^{\frac{2}{D+2}}, |\theta_i^*| < \tilde{\sigma}\}; \\
A_2 &:= \{i : \pi_{t_1}^{-1}(i) < d^\dagger(t_1), |\theta_i^*| > M\}; \\
A_3 &:= \{i : \pi_{t_1}^{-1}(i) < d^\dagger(t_1), |\theta_i^*| < \tilde{\sigma}\} \setminus A_1.
\end{aligned}$$

and

$$B_1 := \{i : \pi_{t_1}^{-1}(i) > d^\dagger(t_1), |\theta_i^*| < \tilde{\sigma}\}; \quad B_2 := \{i : \pi_{t_1}^{-1}(i) > d^\dagger(t_1), |\theta_i^*| > M\}.$$

where $\tilde{\sigma}^2 \le \min\{\frac{|B_1|}{C_{B_1}}\varepsilon^2, c'\varepsilon^2\}$, $C_{B_1} = \min\{1 \vee (|A_1| - |B_1|), |B_2|\}$ and $c'$ is a constant $\le 1$. Also recall from the assumption of Theorem 5.2 that suppose $|\theta_i^*| > |\theta_j^*| > M$, then if $C_{max} M > |\theta_i^*|$, $\eta_{i,j} = |\theta_i^*| - |\theta_j^*| > C_\eta \varepsilon'$; otherwise, $\frac{|\theta_i^*|}{|\theta_j^*|} > (1 + \frac{c_\eta}{D})$.

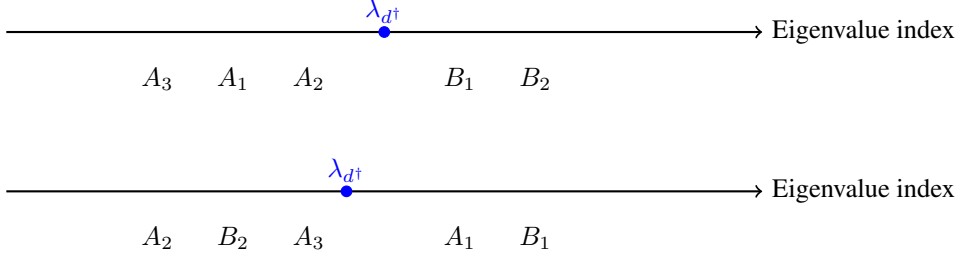

Throughout the proof, we assume all the events $\{E_k\}$ in Lemma F.9 hold. We divide the proof into several parts.

**Part 1: Very small eigenvalues can be ignored**. From Assumption 5.1, we have $\inf_{j \in S} \lambda_j > n^{-\delta}$, where $\delta$ is a constant. For $i \in R$, if $\lambda_i < n^{-(2.1\delta \vee 5)}$, Proposition F.2 implies that at $t_2$, we have

$$\tilde{\lambda}_i(t_2) < n^{1.1} \cdot \lambda_i^{0.99} < n^{-\delta}.$$

If $B_2$ is empty, then all the signals in $R$ is 0 by the definition of $\tilde{\sigma}$. Otherwise, for $\tilde{\lambda}_i < n^{-\delta}$, and $\lambda_{\pi_{t_1}(d^\dagger(t_1))} > n^{-\delta}$. According to the monotonicity of eigenvalues in Lemma F.4, any the index $i$ such that $\lambda_i < n^{-(2.1\delta \vee 5)}$ can not rank among the first $d^\dagger(t_1)$ at time $t_2$, i.e., it makes no difference to the variation of $d^\dagger$ from $t_1$ to $t_2$.

**Part 2: Analysis for $B_2$**

If $j \in B_2$, by Proposition F.1, we have $|\theta_j(t_2) - \theta_j^*| < \varepsilon' \le \frac{1}{C_M} M$. We apply Equation (50) to get $\beta^2 = a^2 - \lambda < a^2$ and $\beta^2 = D^{-1}(b^2 - b_0^2) < D^{-1} b^2$, and thus

$$|\theta_j(t)| = a_j(t) b_j^D(t) |\beta_j(t)| \le a_j(t) b_j^D(t) \cdot a_j^{\frac{1}{D+1}}(t) \cdot D^{-\frac{D}{2(D+1)}} b_j^{\frac{D}{D+1}}(t) = D^{-\frac{D}{2(D+1)}} \left[ a_j(t) b_j^D(t) \right]^{\frac{D+2}{D+1}}.$$

Therefore, at time $t = t_2$, we have for some constants $c$ and $C$ that

$$D^{-\frac{D}{2(D+1)}} \left[a_j(t)b_j^D(t)\right]^{\frac{D+2}{D+1}} \geq |\theta_j(t)| \geq (1-c)M, \implies a_j(t)b_j^D(t) \geq C \cdot D^{\frac{D}{2(D+2)}} M^{\frac{D+1}{D+2}}.$$

It follows that $\tilde{\lambda}_j(t_2) \geq cD^{\frac{D}{(D+2)}} M^{\frac{2(D+1)}{D+2}}$. Moreover, for $\lambda_j$ and $D^{-1}b_0^2$ that are much smaller than $c \cdot D^{-\frac{D}{D+2}} \cdot M^{\frac{2}{D+2}}$, we use Equation (50) to obtain

$$\beta_j^2 \asymp a_j^2 \asymp D^{-1}b_j^2 \asymp D^{-\frac{D}{D+2}} M^{\frac{2}{D+2}}. \tag{39}$$

**Part 3: $A_1$ and $B_2$ will exchange**

In the following, suppose $i \in A_1$ and $j \in B_2$. We will prove $\tilde{\lambda}_i(t_2) < \tilde{\lambda}_j(t_2)$ by contradiction.

If $\tilde{\lambda}_i(t_2) \geq \tilde{\lambda}_j(t_2)$, then by Proposition F.1, we have

$$|\theta_j(t_2) - \theta_j^*| < 2\varepsilon'.$$

By Lemma F.4, we have

$$|\theta_i(t_2)| < |\theta_i^*| + \kappa_i.$$

We have $|\theta_j| > C_D|\theta_i|$ where $C_D > 1 + \frac{c}{D}$ for $M = C_M\varepsilon > C_M|\kappa_i|$. At $t_2$, the following holds:

$$|\beta_j(t_2)| > C_D|\beta_i(t_2)|.$$

It follows that $a_i^2(t_2)b_i^{2D}(t_2) > a_j^2(t_2)b_j^{2D}(t_2)$. Combined with Equation (50), we have

$$\frac{\beta_i^2(t_2) + \lambda_i}{\beta_j^2(t_2) + \lambda_j} > \left(\frac{D\beta_j^2(t_2) + b_0^2}{D\beta_i^2(t_2) + b_0^2}\right)^D > \left(\frac{C_D}{1+\delta}\right)^D = (1 + c(C_D - 1))^D. \tag{40}$$

Recall that $|\beta_i(t_2)| < \frac{1}{C_D}|\beta_j(t_2)|$ and $\beta_j^2(t_2) > CD^{-\frac{2}{D+2}} M^{\frac{2}{D+2}}$. If we choose the constant $C_D$ such that $(1 + c(C_D - 1))^D$ is large enough, the inequality Equation (40) will implies $\lambda_i$ larger than its upper bound in the definition of set $A_1$. (We can let $C_D = 1 + c \cdot \frac{1}{D}$.)

The contraction shows that $\tilde{\lambda}_i(t_2) < \tilde{\lambda}_j(t_2)$ for any $i \in A_1, j \in B_2$. If the sets $A_1$ and $B_2$ are not empty when $t = t_1$, then from $t_1$ to $t_2$, the elements of set $B_2$ will be arranged before those of set $A_1$ according to the eigenvalue index. We only need $\lambda_i < c \cdot D^{-\frac{D}{D+2}} M^{\frac{2}{D+2}}$.

For the same reason, the elements of set $A_2$ will be arranged before those of set $A_1$ at $t_2$.

**Part 4: $A_2$ and $B_2$ will be monotonously nonincreasing**

In the following, W.L.O.G we assume $\theta^*, \theta^* > 0$. We prove that given $i \in A_2, j \in B_2$, if $\theta_i^* > \theta_j^*$, we have $\tilde{\lambda}_i(t_2) > \tilde{\lambda}_j(t_2)$.

If $\theta_i^* - \theta_j^* > C_\eta\varepsilon$, we have $z_i > z_j$. Then by Proposition F.1, we have at $t_2$

$$|\theta_i(t_2) - \theta_i^*| < 2\varepsilon', \quad |\theta_j(t_2) - \theta_j^*| < 2\varepsilon'.$$

If $\lambda_i \geq \lambda_j$, by Equation (38) and monotonicity, we always have $\tilde{\lambda}_i(t) > \tilde{\lambda}_j(t)$.

Now, consider the case where $\lambda_i < \lambda_j$. By the definition of $A_1$ and $B_2$, we have $\lambda_j < c \cdot D^{-\frac{D}{D+2}} \cdot M^{\frac{2}{D+2}}$. Next, we use proof by contradiction. Note that

$$\frac{\theta_i(t_2)}{\theta_j(t_2)} = \frac{\tilde{\lambda}_i^{\frac{1}{2}}(t_2)\beta_i(t_2)}{\tilde{\lambda}_j^{\frac{1}{2}}(t_2)\beta_j(t_2)} \geq C_D.$$

If $\tilde{\lambda}_i(t) \leq \tilde{\lambda}_j(t)$, then $\beta_i(t_2) > C_D\beta_j(t_2)$. By Equation (39), both $\beta_i(t_2)$ and $\beta_j(t_2)$ are much larger then $D^{-\frac{1}{2}}b_0$. It then follows that

$$\left(\frac{D\beta_i^2(t_2) + b_0^2}{D\beta_j^2(t_2) + b_0^2}\right)^D > C^*$$

where $C^*$ is a constant that can be made large enough by choosing $C_D$. Therefore, using the same reason as in Equation (40), we can get

$$\frac{\beta_i^2(t_2) + \lambda_i}{\beta_j^2(t_2) + \lambda_j} < \frac{1}{C^*},$$

which is impossible because $\beta_i(t_2) > C_D \beta_j(t_2)$ and $\beta_i^2(t_2) > C\lambda_j$.

**Part 5: $B_1$ will stay behind $A_2$, $A_3$ and $B_2$**

In the following, let $j \in B_1$. By $y = \theta^* + \xi$, we have $|z_j| < \tilde{\sigma} + \varepsilon'$. By Equation (53) and $|\theta_j| < |z_j|$, we have

$$|\beta_j(t_2)| \leq \left( D^{-D/2} |z_j| \right)^{1/(D+2)}.$$

Then by $\tilde{\lambda}_j(t_2) = (\beta_j^2(t_2) + \lambda_j)(D\beta_j^2(t_2) + b_0^2)^D < (1 + \frac{1}{c \cdot D})^D b_0^{2D}(\beta_j^2(t_2) + \lambda_j)$. Then if $\lambda_j < c \cdot D^{-D/(D+2)} \varepsilon^{2/(D+2)}$, given any $i \in A_3 \cup A_2 \cup B_2$, $\tilde{\lambda}_i(t_2) > \tilde{\lambda}_j(t_2)$. Otherwise, we have $\lambda_j > C \cdot D^{-D/(D+2)} \varepsilon^{2/(D+2)}$, then we have $\tilde{\lambda}_j(t_2) < (1 + \delta)\tilde{\lambda}_j(t_1)$, then by the definition of $d^\dagger$, at least $d^\dagger$ eigenvalues will larger than $\tilde{\lambda}_j(t_2)$.

**Part 6: $A_3$ will be ahead of $A_1$**

by the same reason between $B_1$ and $A_1$, for given $j \in A_3$, $\tilde{\lambda}_j(t_2) = (\beta_j^2(t_2) + \lambda_j)(D\beta_j^2(t_2) + b_0^2)^D < (1 + \frac{1}{c \cdot D})^D b_0^{2D}(\beta_j^2(t_2) + \lambda_j)$, and $i \in A_1$, $\lambda_i < c \cdot D^{-D/(D+2)} \varepsilon^{2/(D+2)}$, combined with

$$|\beta_i(t_2)| \leq \left( D^{-D/2} |z_i| \right)^{1/(D+2)}.$$

then we have $\tilde{\lambda}_j(t_2) > \tilde{\lambda}_i(t_2)$ for $c$ is small and $C$ is large enough.

**Part 7: Ordering of the spectrum at $t_2$ and $d^\dagger(t_1) \geq d^\dagger(t_2)$**

To show $d^\dagger(t_2) \leq d^\dagger(t_1)$, it suffices to show $\mathbf{H}_{\boldsymbol{\theta}^*, \tilde{\boldsymbol{\lambda}}(t_2)}(d^\dagger(t_1)) < \mathbf{H}_{\boldsymbol{\theta}^*, \tilde{\boldsymbol{\lambda}}(t_2)}(d^\dagger(t_1))$, which is equivalent to prove the following difference

$$\sum_{i:\pi_{t_1}^{-1}(i) > d^\dagger(t_1)} |\theta_i^*|^2 - \sum_{i:\pi_{t_2}^{-1}(i) > d^\dagger(t_1)} |\theta_i^*|^2 \tag{41}$$

is nonnegative.

We will make use of $|A_1| + |A_2| + |A_3| = d^\dagger(t_1)$ and consider two possible cases.

- Case 1: $|B_2| \leq |A_1|$. Since $B_2 \cup A_2$ is ahead of $A_1 \cup B_1$, we can see that the eigenvalue of the last element of $B_2 \cup A_2$ is among the top $d^\dagger$ ones. Because $A_3$ is ahead of $A_1 \cup B_1$, so only some of $A_1$ is swapped to the later part. Also some of $B_1$ may arise ahead some of $A_1$. Therefore, to analyze the ordering of eigenvalues $\tilde{\boldsymbol{\lambda}}(t_2)$, we define

$$\begin{aligned} B_{11} &= \{i \in B_1 : \pi_{t_2}^{-1}(i) \leq d^\dagger(t_1)\}. \\ A_{11} &= \{i \in A_1 : \pi_{t_2}^{-1}(i) > d^\dagger(t_1)\}. \end{aligned} \tag{42}$$

  Here $A_{11}$ contains all the elements that move from the top $d^\dagger(t_1)$ part to the later part, while $B_2$ and $B_{11}$ are the elements that move from the later part to the top $d^\dagger(t_1)$ part. Therefore, we have $|A_{11}| = |B_2| + |B_{11}|$. Let $C_{B_1} := \min\{(|A_1| - |B_2|)_+, |B_1|\}$. We have $|B_{11}| \leq C_{B_1}$.

  W.L.O.G., we can write divide $A_{11}$ into two subsets such that $|A_{111}| = |B_2|$ and $|A_{112}| = |B_{11}|$. We can then write Equation (41) as

$$\|\theta_{B_2}^*\|_2^2 + \|\theta_{B_{11}}^*\|_2^2 - \|\theta_{A_{111}}^*\|_2^2 - \|\theta_{A_{112}}^*\|_2^2. \tag{43}$$

  The exchange between $A_1$ and $B_2$ yields

$$\|\theta_{B_2}^*\|_2^2 - \|\theta_{A_{111}}^*\|_2^2 \geq |B_2|(M^2 - \tilde{\sigma}^2), \tag{44}$$

  and

$$\|\theta_{B_{11}}^*\|_2^2 - \|\theta_{A_{112}}^*\|_2^2 \geq -|B_{11}|\tilde{\sigma}^2 \geq -C_{B_1}\tilde{\sigma}^2.$$

Note the assumption of Theorem 5.2 that $(|B_2| + C_{B_1})\tilde{\sigma}^2 \leq |B_2|M^2$. We add the last two inequalities Equation (43) and appendix F.1 together to get

$$\sum_{i:\pi_{t_1}(i)>d^\dagger(t_1)} |\theta_i^*|^2 - \sum_{i:\pi_{t_2}(i)>d^\dagger(t_1)} |\theta_i^*|^2 \geq 0,$$

where $\geq$ becomes $>$ if $B_2 \neq \emptyset$. By the definition of $d^\dagger$, $d^\dagger(t_2) \leq d^\dagger(t_1)$.

- Case 2: $|B_2| > |A_1|$. If the eigenvalue of the last element of $B_2 \cup A_2$ is among the top $d^\dagger$, we follow the exact same proof in Case 1.

  Now suppose that the eigenvalue of the last element of $B_2 \cup A_2$ is in the later part. In this case, all elements $B_1$ and $A_1$ are in the later part. We first identify all the elements that fall in the later part at time $t_2$: in addition to all elements of $B_1$, the following

$$B_{21} := \{i \in B_2 : \pi_{t_2}^{-1}(i) > d^\dagger\},$$
$$A_{11} := \{i \in A_1 : \pi_{t_2}^{-1}(i) > d^\dagger\},$$
$$A_{21} := \{i \in A_2 : \pi_{t_2}^{-1}(i) > d^\dagger\},$$
$$A_{31} := \{i \in A_3 : \pi_{t_2}^{-1}(i) > d^\dagger\}.$$

Note that at time $t_1$, the elements in the later part are in $B_1$, $B_{21}$, and $B_{22} := B_2 \setminus B_{21}$. Therefore, Equation (41) can be written as

$$\|\theta_{B_{22}}\|^2 - \|\theta_{A_{11}}\|^2 - \|\theta_{A_{21}}\|^2 - \|\theta_{A_{31}}\|^2. \tag{45}$$

By definition of $B_2$, $A_1$, and $A_3$, each squared element in $B_2$ is larger than that of both $A_1$ and $A_3$. In addition, since $A_2$ and $B_2$ will be monotonously nonincreasing, for any element in $B_{22}$, its squared signal will be no less than that of any element in $\theta_{A_{21}}$. Therefore, we conclude that Equation (45) is nonnegative and will be positive if $B_{22}$ is not empty.

## F.2 Generalized signal results by dynamic equation analysis

**Proposition F.1** (Shrinkage monotonicity and shrinkage time). *Suppose all the events $\{E_k\}$ in Lemma F.9 hold. Let $\varepsilon = 2(C_{proxy})^{-1/2}\sqrt{\frac{\ln n\tilde{d}}{n} \cdot \ln n}, \varepsilon' = 2(C_{proxy})^{-1/2}\sqrt{\frac{\ln n}{n}}$.*

*For any $j \in S$ (as defined in Assumption 5.1), we have*

$$|\theta_j^* - \theta_j(t)| < 2\varepsilon'. \quad \forall t \geq t(\varepsilon) \tag{46}$$

*where $t(\varepsilon) = C\, b_0^{-D}\, \varepsilon^{-1} \ln n$ for some absolute constant $C$.*

*Proof.* When all the events $E_k$ in Lemma F.9 hold, we have

$$\|\xi_S\|_\infty \leq \varepsilon'. \tag{47}$$

Consider $j \in S$. We have $\theta_j^* \geq 8\varepsilon'$(We let $C_M \geq 8$). By taking $\delta = \varepsilon'$ and also $\kappa = \varepsilon'$ in Lemma F.7, we have

$$|\theta_j^* - \theta_j(t)| \leq 2\varepsilon', \quad \forall t \geq \bar{T}^{app}(\delta),$$

with

$$\bar{T}^{app}(\delta) \leq T^{\text{sig}} + C_2 D^{\frac{D}{D+2}}(\theta_j^*)^{-\frac{2D+2}{D+2}} \ln^+ \frac{2\theta_j^*}{\delta}, \tag{48}$$

and

$$T^{\text{sig}} \leq \begin{cases} C_1(\theta_j^*)^{-1}b_0^{-D} \ln\left(\frac{eb_0}{a_{0j}\sqrt{D}}\right) & a_{0j} < b_0/\sqrt{D}; \\ C_1(\theta_j^*)^{-1}a_{0j}^{-1} \ln\left(\frac{ea_{0j}\sqrt{D}}{b_0}\right) & a_{0j} > b_0/\sqrt{D}, \text{ and } D = 1; \\ C_1(\theta_j^*)^{-1}D^{-\frac{1}{2}}a_{0j}^{-1}b_0^{-D+1} & a_{0j} > b_0/\sqrt{D}, \text{ and } D > 1, \end{cases} \tag{49}$$

where both $C_1$ and $C_2$ are absolute constants.

For the choice $b_0 = cD^{\frac{D+1}{D+2}}\varepsilon^{\frac{1}{D+2}}$, the second term on the right-hand side of Equation (48) is dominated by the right-hand side of Equation (49), and we can choose $c$ small enough so that the summation of the two terms is bounded by $b_0^{-D}\varepsilon^{-1}\ln n$. This justifies our choice of $t(\varepsilon)$.

$\square$

**Proposition F.2.** *We consider the set R. We let $j \in R$, and suppose all the events $\{E_k\}$ in Lemma F.9 hold. Given $b_0$ and $t(\varepsilon)$ defined in Proposition F.1. For any positive constant $\delta'$, if the eigenvalue $\lambda_j < n^{-5}$, and $n$ is large enough, then we have*

$$\tilde{\lambda}_j(t) < 2\lambda_j^{1-2\delta'}b_0^D \cdot n^{1+\delta'}.$$

*Proof.* Since the events in Lemma F.9 hold, we have $|\xi_j| \leq 2(C_{\text{proxy}})^{-1/2}\sqrt{\frac{\ln \tilde{j} + \ln n}{n}}$, where $\tilde{j} = \frac{\tilde{d}}{\lambda_j}$. Since $j \in R$, we have $|\theta_j^*| \leq \tilde{\sigma} \leq \sqrt{c'}\varepsilon$. Since $b_0^D t(\varepsilon) = \varepsilon^{-1}\ln n$, we can check that $t$ is no more than the hitting time $T_2$ defined by Equation (67) in Lemma F.8 as follows.

Note that

$$\varepsilon^{-1}\ln n(|\theta_j^*| + |\xi_j|) \leq \sqrt{c'}\ln n + \sqrt{\ln n}\sqrt{\frac{n}{\ln n\tilde{d}}} \cdot \sqrt{\frac{\ln n\tilde{d} + \ln \frac{1}{\lambda_j}}{n}}$$

$$< \ln n + \sqrt{\ln n + \ln \frac{1}{\lambda_j}}.$$

Since $b_0 = c \cdot D^{\frac{D+1}{D+2}}\varepsilon^{\frac{1}{D+2}}$, for $n$ large enough, we have $\lambda_j < n^{-5} < \frac{b_0^2}{D^2}$ and also

$$\ln \frac{b_0/D}{\lambda_j^{\frac{1}{2}}} = \ln c + \frac{1}{2(D+2)}(\ln n + \ln\ln n\tilde{d}) - \frac{1}{D+2}\ln D + \frac{1}{2}\ln \frac{1}{\lambda_j} > \ln n + \sqrt{\ln n + \ln \frac{1}{\lambda_j}}.$$

It then follows that

$$\beta_j(t) < \lambda_j^{\frac{1}{2}}\exp\left(b_0^D t(|\theta_j^*| + |\xi_j|)\right)$$

$$\leq \lambda_j^{\frac{1}{2}}\exp\left(\sqrt{\ln n}\sqrt{\frac{n}{\ln n\tilde{d}}} \cdot \sqrt{\frac{\ln n\tilde{d} + \ln \frac{1}{\lambda_j}}{n}}\right) \cdot n$$

$$< \lambda_j^{\frac{1}{2}}\exp\left(\sqrt{\ln \frac{1}{\lambda_j}}\right) \cdot \exp\left(\sqrt{\ln n}\right) \cdot n$$

$$< \lambda_j^{\frac{1}{2}-\delta'}n^{1+\delta'}.$$

Using $\tilde{\lambda}_j(t) = (\beta_j^2(t) + \lambda_j)(b_0^2 + D\beta_j^2(t))^D$, we obtain the desired result for sufficiently large $n$. $\square$

**Remark F.3.** *The above proposition provides a very weak upper bound on $\tilde{\lambda}$, but it is sufficient to show that any eigenvalue $\lambda_j$, such that if given any constant $C$, $\lambda_j < n^{-C}$, then $\tilde{\lambda}_j(t)$ is also less than any polynomial of $n^{-1}$. Therefore, when considering the eigenvalue ordering problem, such signals can be ignored.*

### F.3 CONSERVATION QUANTITY

We omit the subscript $j$ in the following two sections F.3 and F.4 because all the proofs are similar for $j = 1, 2, \cdots, d$. By Equation (38), it is easy to see that

$$\frac{d}{dt}a^2 = \frac{1}{D}\frac{d}{dt}b^2 = \frac{d}{dt}\beta^2 = 2ab^D\beta(\theta^* - \theta + \xi).$$

Consequently, we have

$$a^2(t) - \beta^2(t) \equiv a_0^2, \quad b^2(t) - D\beta^2(t) \equiv b_0^2. \tag{50}$$

Using this, we see that

$$a(t) = \left(\beta^2(t) + a_0^2\right)^{1/2}, \quad b(t) = \left(D\beta^2(t) + b_0^2\right)^{1/2} > 0.$$

Using these conservation quantities, we can prove the following estimations in terms of $\beta$:

$$\max\left(a_0, |\beta|\right) \le a \le \sqrt{2}\max\left(a_0, |\beta|\right)$$
$$\max\left(b_0, \sqrt{D}|\beta|\right) \le b \le \sqrt{2}\max\left(b_0, \sqrt{D}|\beta|\right) \tag{51}$$

which also implies that $|\theta| = \left|ab^D\beta\right| \ge D^{D/2}|\beta|^{D+2}$. The evolution of $\theta$. It is direct to compute that

$$\begin{aligned}
\dot{\theta} &= \dot{a}b^D\beta + aDb^{D-1}\dot{b}\beta + ab^D\dot{\beta} \\
&= \left[\left(b^D\beta\right)^2 + \left(Dab^{D-1}\beta\right)^2 + \left(ab^D\right)^2\right](\theta^* - \theta + \xi) \\
&= \theta^2\left(a^{-2} + D^2b^{-2} + \beta^{-2}\right)(\theta^* - \theta + \xi).
\end{aligned} \tag{52}$$

And we also have

$$|\theta| = \left|ab^D\beta\right| \ge D^{D/2}|\beta|^{D+2} \implies |\beta| \le \left(D^{-D/2}|\theta|\right)^{1/(D+2)}. \tag{53}$$

Therefore,

$$\theta^2\left(a^{-2} + D^2b^{-2} + \beta^{-2}\right) \ge \theta^2\beta^{-2} \ge D^{-\frac{D}{D+2}}|\theta|^{\frac{2D+2}{D+2}}. \tag{54}$$

### F.4 MULTI-LAYER DYNAMIC

We study the dynamic of the ODE for any given $j$. Before the analysis, we streamline some notations.

Assume for some $\kappa_j > 0$, it holds that $|\xi_j| \le \kappa_j$. (Note that this $\kappa_j$ can be the high probability upper bound derived using Lemma F.9.) Since $j$ is given, we drop the the subscript $j$ to simplify the exposition throughout this subsection; for example, we write $\lambda$ for $\lambda_j$ and $\theta^*$ for $\theta_j^*$. We write $\ln^+(x) = \max(1, \ln(x))$ for any $x > 0$.

**Lemma F.4** (Monotonicity from equation). *Consider the equation Equation (38). Suppose $y > 0$.*

1. *$a(t), \beta(t)$, and $\theta(t)$ are all non-negative and increasing.*

2. *We have*
$$y \ge \theta(t) \ge 0 \quad \forall t \ge 0.$$

3. *Since $y = \theta^* + \xi$ and $|\xi| \le \kappa$, we have*
$$|\theta^* - \theta(t)| \le |\theta^*| + \kappa, \quad \forall t \ge 0.$$

4. *$|\theta^* - \theta(t)|$ is decreasing provided that $|\theta^* - \theta(t)| > \kappa$.*

5. *If $|\theta^* - \theta(t_1)| \le \kappa$ for some $t_1$, we have*
$$|\theta^* - \theta(t)| \le \kappa \text{ for all } t \ge t_1.$$

*If $y < 0$, Items 3, 4, and 5 still hold, while Items 1 and 2 can be modified by symmetry.*

*Proof.* Items 1 and 2 are directly implied from Equation (38). Item 3 is implied by Item 2.

To prove Item 4, consider Equation (52), from which we have

$$\dot{\theta} = \theta^2 \left( a^{-2} + D^2 b^{-2} + \beta^{-2} \right) (y - \theta),$$

which implies $\dot{\theta} \geq 0$.

Since $|\theta^* - \theta(t)| > \kappa$, we have either $\theta(t) > \theta^* + \kappa$ or $\theta(t) < \theta^* - \kappa$.

The first case is not possible; otherwise, we have $0 < y = \xi + \theta^* \leq \kappa + \theta^* < \theta \leq y$, which is a contradiction. In the second case, we have $|\theta^* - \theta(t)| = \theta^* - \theta(t)$, which is decreasing because $\dot{\theta} \geq 0$.

Item 5 is implied by Item 4.

$\square$

**Lemma F.5** (Approaching from below). *Consider the equation Equation (38). Suppose $\theta^* \geq 8\kappa$ (similar results hold for $\theta^* \leq -8\kappa$ by symmetry). Suppose $t_0 \geq 0$ such that $0 \leq \theta(t_0) < \frac{1}{4}\theta^*$. Define*

$$T^{sig} = \inf \left\{ s \geq 0 : \theta(t_0 + s) \geq \theta^*/4 \right\}.$$

*This is the extra time needed from $t_0$ for $\theta$ to reach $\theta^*/4$. We have*

$$T^{sig} \leq \begin{cases} 4(\theta^*)^{-1} b_0^{-D} \ln\left( \frac{b_0}{a_0 \sqrt{D}} \right) & a_0 < b_0/\sqrt{D}; \\ 4(\theta^*)^{-1} a_0^{-1} \ln\left( \frac{a_0 \sqrt{D}}{b_0} \right) & a_0 > b_0/\sqrt{D}, \text{ and } D = 1; \\ 4(\theta^*)^{-1} D^{-\frac{1}{2}} a_0^{-1} b_0^{-D+1} & a_0 > b_0/\sqrt{D}, \text{ and } D > 1. \end{cases} \tag{55}$$

*Proof.* Since $|y - \theta^*| = |\xi| \leq \kappa$ and $\theta^* \geq 8\kappa$, we have $y \geq 7\kappa > 0$. Therefore, $\theta(t) \in [0, y]$. For any $t \leq t_0 + T^{sig}$, we use $\theta^* \geq 8\kappa$ to show that

$$y - \theta(t) = \theta^* - \theta(t) + \xi \geq \frac{3}{4}\theta^* - \kappa \geq \frac{1}{2}\theta^*.$$

Let $r = \min\left( a_0, b_0/D^{\frac{1}{2}} \right)$ and $R = \max\left( a_0, b_0/D^{\frac{1}{2}} \right)$. Define the following time point if it exists:

$$T^{pos,1} = \inf \left\{ s \geq 0 : \beta(t_0 + s) \geq r \right\}; \quad T^{pos,2} = \inf \left\{ s \geq 0 : \beta(t_0 + s) \geq R \right\}$$

$$T^{sig} = \inf \left\{ s \geq 0 : |\theta^* - \theta(t_0 + s)| \leq \frac{3}{4}\theta^* \right\}.$$

We will first bound both $T^{pos,1}$ and $T^{pos,2}$.

From Equation (38), we have

$$\dot{\beta}(t) = a(t) b^D(t) [\theta^* + \xi - \theta(t)] \geq \frac{1}{4}\theta^* a(t) b^D(t), \quad \text{for} \quad t \leq t_0 + T^{sig}. \tag{56}$$

**Stage 1:** $0 \leq s \leq T^{pos,1}$. Note that $\sqrt{2}a_0 > a(t) > a_0$, and $e \cdot b_0^D > b(t)^D = (D\beta(t)^2 + b_0^2)^{\frac{D}{2}} > b_0^D$. We have

$$\dot{\beta}(t_0 + s) \geq \frac{1}{4}\theta^* a_0 b_0^D \geq \frac{1}{4}\theta^* a_0 b_0^D,$$

which suggests $\beta$ increases at least linearly. Therefore, we have

$$T^{pos,1} \leq 8r \left( \theta^* a_0 b_0^D \right)^{-1}. \tag{57}$$

**Stage 2:** $T^{pos,1} \leq s \leq T^{pos,2}$. Consider two cases.

Case 1: If $a_0 < b_0/\sqrt{D}$, $r = a_0$ and $R = b_0/\sqrt{D}$. Note $a \geq \beta$ in Equation (51). We use Equation (56) to get

$$\dot{\beta}(t_0 + s) \geq \frac{1}{4}\theta^* b_0^D |\beta(t_0 + s)|,$$

By Grönwall's inequality, we have

$$T^{\text{pos},2} - T^{\text{pos},1} \leq 4\left(\theta^* b_0^D\right)^{-1} \ln \frac{b_0}{a_0\sqrt{D}}. \tag{58}$$

Case 2: If $a_0 > b_0/\sqrt{D}$, $R = a_0$ and $r = b_0/\sqrt{D}$. We use $b \geq \sqrt{D}\beta$ in Equation (51) together with $a > a_0$, $b > \sqrt{D}|\beta|$ and Equation (56) to get that

$$\dot{\beta}(t_0 + s) \geq \frac{1}{4}\theta^* a_0 D^{\frac{D}{2}} |\beta(t_0 + s)|^D.$$

By comparison theorem, we have

$$T^{\text{pos},2} - T^{\text{pos},1} \leq \begin{cases} 4\left(\theta^* a_0\right)^{-1} \ln \frac{a_0\sqrt{D}}{b_0}, & \text{if} \quad D = 1; \\ 4\left((D-1)\theta^* a_0 D^{D/2}\right)^{-1}\left[\left(b_0/\sqrt{D}\right)^{-(D-1)} - a_0^{-(D-1)}\right], & \text{if} \quad D \geq 2. \end{cases} \tag{59}$$

**Stage 3:** If $T^{\text{sig}} \leq T^{\text{pos},2}$, then we can use the for $T^{\text{sig}}$ in Stage 2 as a bound for $T^{\text{sig}}$. Now, we consider the case $T^{\text{pos},2} < T^{\text{sig}}$. We combine Equation (51) with $a > |\beta|$, $b > \sqrt{D}|\beta|$, and Equation (56) to get

$$\dot{\beta}(t_0 + T^{\text{pos},2} + s) \geq \frac{1}{4}\theta^* D^{D/2}|\beta(T^{\text{pos},2} + s)|^{D+1}, \quad \text{for} \quad s \in [0, T^{\text{sig}} - T^{\text{pos},2}].$$

Beside, we have $\beta(t_0 + T^{\text{pos},2}) = R > 0$. By Lemma F.10, we have

$$T^{\text{sig}} - T^{\text{pos},2} \leq 4D^{-\frac{D+2}{2}}(\theta^*)^{-1}R^{-D}. \tag{60}$$

We now bound $T^{\text{sig}}$ using the summation of Equation (57), Equation (60), and Equation (58) if $a_0 < b_0/\sqrt{D}$, or the summation of Equation (57), Equation (60), and Equation (59) if $a_0 > b_0/\sqrt{D}$.

If $a_0 < b_0/\sqrt{D}$, we can bound the right hand sides of Equation (57) and Equation (60) by $8(\theta^*)^{-1}b_0^{-D}$ and $4(\theta^*)^{-1}b_0^{-D}$ respectively.

If $a_0 > b_0/\sqrt{D}$, we can bound the right hand sides of Equation (57) and Equation (60) by $8(\theta^*)^{-1}D^{-\frac{1}{2}}a_0^{-1}b_0^{-D+1}$ and $4(\theta^*)^{-1}D^{-\frac{1}{2}}a_0^{-1}b_0^{-D+1}$ respectively. Furthermore, if $D > 1$, we can bound Equation (59) by $4(\theta^*)^{-1}(D-1)^{-1}D^{-\frac{1}{2}}a_0^{-1}b_0^{-D+1}$.

This leads to

$$T^{\text{sig}} \leq \begin{cases} 4(\theta^*)^{-1}b_0^{-D}\left(3 + \ln\left(\frac{b_0}{a_0\sqrt{D}}\right)\right) & a_0 < b_0/\sqrt{D}; \\ 4(\theta^*)^{-1}a_0^{-1}\left(3 + \ln\left(\frac{a_0\sqrt{D}}{b_0}\right)\right) & a_0 > b_0/\sqrt{D}, \text{ and } D = 1; \\ 16(\theta^*)^{-1}D^{-\frac{1}{2}}a_0^{-1}b_0^{-D+1} & a_0 > b_0/\sqrt{D}, \text{ and } D > 1. \end{cases} \tag{61}$$

$\square$

**Lemma F.6** (Approximation time near $\theta^*$). *Consider the equation Equation (38) with $\theta^* \geq 0$ ( a similar result holds for $\theta^* \leq 0$ ). Suppose $\theta^* > 8\kappa$. Suppose for some $t_0 \geq 0$ such that*

$$\frac{1}{4}\theta^* \leq \theta(t_0) \leq \theta^* - \kappa.$$

*Then, for any $\delta > 0$, we have*

$$|\theta^* - \theta(t)| \leq \kappa + \delta, \quad \forall t \geq t_0 + 4^{\frac{2D+2}{D+2}} D^{\frac{D}{D+2}}(\theta^*)^{-\frac{2D+2}{D+2}} \ln^+ \frac{|\theta^* - \theta(t_0)| - \kappa}{\delta}.$$

*Proof.* Given any $\delta > 0$, if $\theta(t_0) \geq \theta^* - \kappa - \delta$, we have $|\theta^* - \theta(t)| \leq \kappa + \delta$ for all $t \geq t_0$ by Lemma F.4 (Item 4) and the desired result is proved.

Next, suppose $\theta(t_0) < \theta^* - \kappa - \delta$. Define

$$T^{\text{app}} = \inf \left\{ s \geq 0 : |\theta^* - \theta(t_0 + s)| \leq \kappa + \delta \right\}.$$

By Lemma F.4 (Item 4) again, it suffices to provide an upper bound on $T^{\text{app}}$.

For all $t \geq t_0$, we have $\frac{1}{4}\theta^* \leq \theta(t)$ by Lemma F.4 (Item 1). Consequently, Equation (54) implies that

$$\theta^2 \left(a^{-2} + D^2 b^{-2} + \beta^{-2}\right) \geq D^{-\frac{D}{D+2}} |\theta|^{\frac{2D+2}{D+2}} \geq 4^{-\frac{2D+2}{D+2}} D^{-\frac{D}{D+2}} (\theta^*)^{\frac{2D+2}{D+2}} := c_0.$$

Furthermore, by Equation (52), we have

$$\dot{\theta} = \theta^2 \left(a^{-2} + D^2 b^{-2} + \beta^{-2}\right) (\theta^* - \theta + \xi) \geq c_0(\theta^* - \kappa - \theta).$$

Let $x(s) := \theta^* - \kappa - \theta(t_0 + s)$ with $x(0) = \theta^* - \kappa - \theta(t_0)$. Note that $T^{app}$ is the hitting time of $x(s)$ to $\delta$. Applying Lemma F.11 to $x(s)$, we have

$$T^{\text{app}} \leq c_0^{-1} \ln \frac{|\theta^* - \theta(t_0)| - \kappa}{\delta}.$$

$\square$

**Lemma F.7.** *Consider the equation Equation (38) with $\theta^* \geq 0$ (a similar result holds for $\theta^* \leq 0$ ).*

*Suppose $\theta^* \geq 8\kappa$. For two absolute constants $C_1, C_2$, we have*

$$|\theta^* - \theta(t)| \leq \kappa + \delta, \quad \forall t \geq \bar{T}^{app}(\delta),$$

*where*

$$\bar{T}^{app}(\delta) := \bar{T}^{sig} + C_2 D^{\frac{D}{D+2}} (\theta^*)^{-\frac{2D+2}{D+2}} \ln^+ \frac{\theta^*}{\delta}, \tag{62}$$

*and*

$$\bar{T}^{sig} := \begin{cases} C_1 (\theta^*)^{-1} b_0^{-D} \ln\left(\frac{b_0}{a_0\sqrt{D}}\right) & a_0 < b_0/\sqrt{D}; \\ C_1 (\theta^*)^{-1} a_0^{-1} \ln\left(\frac{a_0\sqrt{D}}{b_0}\right) & a_0 > b_0/\sqrt{D}, \text{ and } D = 1; \\ C_1 (\theta^*)^{-1} D^{-\frac{1}{2}} a_0^{-1} b_0^{-D+1} & a_0 > b_0/\sqrt{D}, \text{ and } D > 1. \end{cases} \tag{63}$$

*Proof.* We will repeatedly apply the monotonicity of Lemma F.4.

Recall $T^{\text{sig}}$ defined in Lemma F.5 with $t_0 = 0$ and let $t_1$ be the upper bound on $T^{\text{sig}}$ we found therein. Then $\theta(t_1) \geq \frac{\theta^*}{4}$.

We then apply Lemma F.6 with $t_0 = t_1$, and conclude that $|\theta^* - \theta(t)| \leq \kappa + \delta$ for all $t \geq t_1 + t_2$, where $t_2 = 4^{\frac{2D+2}{D+2}} D^{\frac{D}{D+2}} (\theta^*)^{-\frac{2D+2}{D+2}} \ln^+ \frac{|\theta^* - \theta(t_1)| - \kappa}{\delta}$. Note that $|\theta^* - \theta(t_1)| - \kappa \leq \theta^*$. We complete the proof by defining $\bar{T}^{\text{sig}} = t_1$ and $\bar{T}^{app}(\delta) = t_1 + 4^{\frac{2D+2}{D+2}} D^{\frac{D}{D+2}} (\theta^*)^{-\frac{2D+2}{D+2}} \ln^+ \frac{\theta^*}{\delta}$.

$\square$

**Lemma F.8.** *Consider the equation Equation (38). Denote $r' = \min\{a_0, b_0/D\}$, $R' = \max\{a_0, b_0/D\}$. W.L.O.G., We assume that $\theta^* \geq 0$. Define $T_1 = \inf\{t : |\beta(t)| > r'\}$, and $T_2 = \inf\{t : |\beta(t)| > R'\}$. If $D \geq 1$, and $t$ satisfies the following:*

$$\sqrt{2e} a_0 b_0^D \int_0^t (|\theta^*| + |\xi|) ds \leq \min(a_0, b_0/D),$$

*then we have*

$$|\theta(t)| \le 2e \cdot a_0^2 b_0^{2D} \int_0^t (|\theta^*| + |\xi|) \mathrm{d}s. \tag{64}$$

*Moreover, if $a_0 \le b_0/D$ and $0 \le t \le T_2 - T_1$ satisfies the following:*

$$\sqrt{2e} b_0^D \int_0^t (|\theta^*| + |\xi|) \mathrm{d}s \le \ln \frac{b_0/D}{a_0},$$

*then we have*

$$|\beta(t)| \le a_0 \exp\left( b_0^D \int_0^t (|\theta^*| + |\xi|) \mathrm{d}s \right);$$

$$|\theta(t)| \le \sqrt{2e} a_0^2 b_0^D \exp\left( 2\sqrt{e} b_0^D \int_0^t (|\theta^*| + |\xi|) \mathrm{d}s \right).$$

*Proof.* From Equation (38), we have

$$|\beta(t)| \le \int_0^t a(s) b^D(s) (|\theta^*| + |\xi|) ds.$$

Consider $t \le T_1$. We use Equation (51) to get $a(t) \le \sqrt{2} a_0$ and by Equation (50), $b^2(t) - D\beta^2(t) \equiv b_0^2$. Consequently, we have $b^2(t) \le (1 + \frac{1}{D}) b_0^2$, and thus

$$|\beta(t)| \le \sqrt{2e} a_0 b_0^D \int_0^t (|\theta^*| + |\xi|) dt, \tag{65}$$

which implies Equation (64) by using the fact that $|\theta| = |ab^D\beta|$. Furthermore, Equation (65) implies that

$$T_1 \ge \inf\left\{ t \ge 0 : \sqrt{2e} a_0 b_0^D \int_0^t (|\theta^*| + |\xi|) ds \ge r' \right\}.$$

Then when $t > T_1$, in the following, suppose $a_0 \le b_0/D$. We have $r' = a_0$ and $R' = b_0/D$.

Consider $t \in (T_1, T_2)$. We have $a(t) \le \sqrt{2}|\beta(t)|$ and $b(t)^2 \le (1 + \frac{1}{D}) b_0^2$. Consequently, Equation (38) implies that

$$|\beta(t)| \le a_0 + \sqrt{2e} b_0^D \int_{T_1}^t |\beta(s)| (|\theta^*| + |\xi|) \mathrm{d}s, \tag{66}$$

for any $t \in (T_1, T_2)$. By Grönwall inequality, we have

$$|\beta(t)| \le a_0 \exp\left( \sqrt{2e} b_0^D \int_{T_1}^t (|\theta^*| + |\xi|) \mathrm{d}s \right), \quad t \in (T_1, T_2).$$

By definition of $T_2$, we have

$$T_2 \ge \inf\left\{ t \ge T_1 : \sqrt{2e} b_0^D \int_{T_1}^t (|\theta^*| + |\xi|) \mathrm{d}s = \ln \frac{b_0/D}{a_0} \right\}. \tag{67}$$

The bound for $\theta(t)$ now follows from using the bounds $a(t) \le \sqrt{2}|\beta|, b^2(t) \le (1 + \frac{1}{D}) b_0^2$ to get

$$|\theta(t)| = |ab^D\beta| \le \sqrt{2e} b_0^D |\beta|^2 \le \sqrt{2e} a_0^2 b_0^D \exp\left( 2\sqrt{e} b_0^D \int_{T_1}^t (|\theta^*| + |\xi|) \mathrm{d}s \right), \forall t \in (T_1, T_2).$$

$\square$

## F.5 Auxiliary Lemma

The following lemma provides a choice of $\kappa_j \geq |\xi_j|$ with high probability.

**Lemma F.9.** *Recall $S$ defined by Assumption 5.1 and let $C = 2(C_{proxy})^{-1/2}$. For $k \in S$, we introduce the events $\{E_k\}$ as follows:*

$$E_k := \{|\xi_k| \leq Cn^{-1/2}\sqrt{\ln n}\}. \tag{68}$$

*For $k \in S^C$, we introduce the events $\{E_k\}$ as follows:*

$$E_k := \left\{|\xi_k| \leq Cn^{-1/2}\sqrt{\ln\left(n\tilde{k}\right)}\right\}. \tag{69}$$

*where $\tilde{k} = \sum_j \lambda_j/\lambda_k$.*

*Then, with probability at least $1 - \frac{4}{n}$, all events $E_k, k \in [d]$ hold simultaneously.*

*Proof.* By Assumption 5.1, the noise $\xi_k$ is sub-Gaussian with variance proxy $C_{\text{proxy}}/n$. Therefore, $\mathbf{P}\left(|\xi_k| \geq s\right) \leq 2\exp\left(-(2C_{\text{proxy}})^{-1}ns^2\right)$.

If $k \in S$, we have

$$\mathbf{P}\left\{|\xi_k| \geq 2C_{\text{proxy}}^{-1/2}\sqrt{\frac{\ln n}{n}}\right\} \leq 2\exp\left(-2(\ln n)\right).$$

By the union bound, we have

$$\begin{aligned} \mathbf{P}\left\{\cap_{k \in S} E_k\right\} &\geq 1 - \sum_{k \in S} \mathbf{P}\left\{|\xi_k| \geq 2C_{\text{proxy}}^{-1/2}\sqrt{\frac{\ln n}{n}}\right\} \\ &\geq 1 - |S|2\exp\left(-2(\ln n)\right) \\ &\geq 1 - \frac{2}{n}, \end{aligned} \tag{70}$$

where the last inequality is because $|S|2\exp\left(-2(\ln n)\right) \leq 2n^{-1}$.

If $k \in S^C$, we have

$$\mathbf{P}\left\{|\xi_k| \geq 2(C_{\text{proxy}})^{-1/2}\sqrt{\frac{\ln n\tilde{k}}{n}}\right\} \leq 2\exp\left(-(\ln n + \ln\frac{\sum_j \lambda_j}{\lambda_k})\right) \leq \frac{2}{n} \cdot \frac{\lambda_k}{\sum_j \lambda_j}, \tag{71}$$

where we recall that $\tilde{k} = \frac{\sum_j \lambda_j}{\lambda_k}$.

By the union bound, we have

$$\begin{aligned} \mathbf{P}\left\{\cap_{k \in S^C} E_k\right\} &\geq 1 - \sum_{k \in S^C} \mathbf{P}\left\{|\xi_k| \geq 2(C_{\text{proxy}})^{-1/2}\sqrt{\frac{\ln n\tilde{k}}{n}}\right\} \\ &\geq 1 - \sum_{k \in S^C} \frac{2}{n} \cdot \frac{\lambda_k}{\sum_j \lambda_j} \\ &\geq 1 - \frac{2}{n}. \end{aligned} \tag{72}$$

We combined the Equation (70) and Equation (72), and we derive the results.

$\square$

The following two lemmas provide convenient upper bounds on hitting times of ODE solutions.

**Lemma F.10.** *Let $k > 0$ and $p > 1$.*

- *Consider the ODE*

$$\dot{x} \geq kx^p, \quad x(0) = x_0 > 0$$

*Then we have*

$$x(t) \geq \left( x_0^{-(p-1)} - (p-1)kt \right)^{-\frac{1}{p-1}}$$

*and thus for any $M \geq 0$,*

$$\inf\{t \geq 0 : x(t) \geq M\} \leq \left[ (p-1)kx_0^{p-1} \right]^{-1}. \tag{73}$$

- *Consider the ODE*

$$\dot{x} \leq -kx^p, \quad x(0) = x_0 > 0.$$

*Then we have*

$$x(t) \leq \left( x_0^{-(p-1)} + (p-1)kt \right)^{-\frac{1}{p-1}},$$

*and thus for any $M > 0$,*

$$\inf\{t \geq 0 : x(t) \leq M\} \leq \left[ (p-1)kM^{p-1} \right]^{-1}. \tag{74}$$

**Lemma F.11.** *Let $k > 0$ and $x_0 > 0$.*

    *1. If*

$$\dot{x}(t) \geq k\, x(t), \qquad x(0) = x_0,$$

*then for all $t \geq 0$, it holds that*

$$x(t) \geq x_0\, e^{kt},$$

*and for every $M \geq x_0$, we have*

$$\inf\{t \geq 0 : x(t) \geq M\} \leq \frac{1}{k} \log\!\left(\frac{M}{x_0}\right).$$

    *2. If*

$$\dot{x}(t) \leq -k\, x(t), \qquad x(0) = x_0,$$

*then for all $t \geq 0$, it holds that*

$$x(t) \leq x_0\, e^{-kt},$$

*and for every $0 < M \leq x_0$, we have*

$$\inf\{t \geq 0 : x(t) \leq M\} \leq \frac{1}{k} \log\!\left(\frac{x_0}{M}\right).$$

## G  RELATED WORK ON PRINCIPAL COMPONENT REGRESSION

As discussed in Section 3.1, the PC estimator serves as a motivating example for the concepts of ESD and span profile due to its clear illustration of bias-variance trade-offs. However, the ESD and span profile are designed to characterize the intrinsic difficulty of generalization arising from signal-kernel alignment, and their definitions do not rely on the PC estimator. Nonetheless, the analysis of PC estimators, particularly in high-dimensional linear regression, been an active area of recent research. Below, we briefly summarize some relevant contributions to provide context.

### G.1  PROPORTIONAL ASYMPTOTICS

Several studies analyze Principal Component Regression (PCR) in the proportional asymptotic setting where the dimension $p$ and sample size $n$ grow with $p/n \to \gamma$. In this regime, Xu & Hsu (2019) study the limiting risk of PCR with Gaussian designs with diagonal covariance. They assume polynomially decaying eigenvalues or a convergent empirical spectrum, together with an isotropic prior, and they reveal a "double-descent" risk curve. In a related vein, Wu & Xu (2020) extend the analysis to a general covariance matrix $\Sigma_x$ and an anisotropic prior satisfying $\mathbb{E}\beta_* \beta_*^\top = \Sigma_\beta$. They also derive an exact risk expression and demonstrate how "misalignment" between $\Sigma_x$ and $\Sigma_\beta$ affects risk; here "alignment" refers to concordance between the orderings of their eigenvalues. Both studies assume knowledge of the eigenvectors of the population covariance matrix $\Sigma_x$ to construct the *oracle PCR*. Gedon et al. (2024) analyze the limiting risk of PCR under a latent factor model and explore the effect of distribution shift. Green & Romanov (2024) derive the exact limits of estimation risk, in-sample prediction risk, and out-of-sample prediction risk of PCR under the assumption that both the empirical spectrum distribution and the distribution of the true signal's mass over the eigenspaces of $\Sigma_x$ converges weakly.

### G.2  NON-ASYMPTOTIC ANALYSIS

Complementary research develops non-asymptotic guarantees. Agarwal et al. (2019) derive finite-sample upper bounds on prediction error using $\|\beta^*\|_1^2$ and the rank of the design matrix under latent factor models, and they explore the robustness of PCR to noise and missing values in the observed covariates. Bing et al. (2021) consider PCR with an adaptively selected number of components under latent factor models and provide alternative finite-sample risk bounds using $\|\beta^*\|_2^2$. Huang et al. (2022) derive non-asymptotic risk bounds for PCR in more general settings by analyzing the alignment between population and empirical principal components. Hucker & Wahl (2023) derive non-asymptotic error bounds for PCR in kernel regression.

