# OpenReview forum: "Alignment-Sensitive Minimax Rates for Spectral Algorithms with Learned Kernels"
_ICLR.cc/2026/Conference — Submitted to ICLR 2026_

### Official Review · Reviewer_zuxG · 2025-10-28

**Soundness:** 3
**Presentation:** 2
**Contribution:** 2
**Rating:** 6
**Confidence:** 3

**Summary:**

This paper introduces the Effective Span Dimension (ESD), a new alignment-sensitive complexity measure for spectral learning algorithms. The ESD jointly depends on the signal, the kernel spectrum, and the noise level, and is shown to characterize the minimax excess risk in sequence models, as well as in their linear and RKHS counterparts. The authors further analyze over-parameterized gradient flow dynamics, proving that adaptive eigenvalue learning can reduce the ESD, and thereby improve generalization. The paper concludes with numerical experiments that visualize ESD evolution and empirically support the theoretical claims.

**Strengths:**

The paper offers an original and technically sound contribution to the theory of spectral learning. The introduction of the ESD provides a novel lens through which to understand generalization in adaptive kernel and spectral methods. The ESD allows the analysis of signal–spectrum alignment, bias–variance trade-offs, and minimax optimality within a single framework. The theoretical results are clearly presented, mathematically sound, and supported by illustrative experiments.

**Weaknesses:**

The connection to neural networks is somewhat overstated. Most theoretical results hold only in idealized sequence, linear, or RKHS models, while genuine neural networks involve evolving eigenfunctions, which the paper explicitly leaves for future work. Consequently, the practical relevance to real NN training remains speculative. In addition, while the paper contrasts ESD with classical notions such as the effective dimension, this comparison is not entirely fair: traditional kernel learning theory (e.g., Caponnetto & De Vito, Zhang) already incorporates signal–kernel alignment through the RKHS norm of the target function or source condition constants, albeit indirectly. While the ESD formalism clarifies and extends this dependence, the importance of target-kernel alignment was already recognized as a critical aspect of learning in the past literature.

**Questions:**

1. _Applicability to Neural Networks:_ The paper’s framing emphasizes relevance to neural network generalization, yet all formal results are derived in simplified sequence or RKHS models. Could the authors clarify what specific aspects of neural network training (beyond the stylized over-parameterized gradient flow) they expect the ESD framework to capture? For instance, do they anticipate ESD reductions under standard SGD dynamics in finite-width networks?

2. _Estimating the ESD in Practice:_ Since the ESD is defined at the population level and depends on the unknown signal coefficients, it is unclear how it could be estimated from data. Do the authors foresee practical estimators or proxies for the ESD that could be applied in empirical studies?

3. _Relation to Classical Source Conditions:_ Classical RKHS learning rates already depend on the target’s regularity (e.g., through the source condition constant or RKHS norm). Could the authors elaborate on how their framework fundamentally differs from these forms of target dependence? Is the ESD intended to replace such constants, or does it complement them?

---

> ### Author Response · Authors · 2025-11-21
> **Connection to neural network**
>
> We thank the reviewer for the thoughtful and constructive feedback. We are encouraged that you find our contribution original, technically sound, and mathematically rigorous. We appreciate your recognition of the ESD as a novel lens for understanding generalization.
>
> Below, we address your concerns regarding the connection to neural networks and the comparison with classical kernel theory.
>
> > The connection to neural networks is somewhat overstated. Most theoretical results hold only in idealized sequence, linear, or RKHS models, while genuine neural networks involve evolving eigenfunctions, which the paper explicitly leaves for future work. Consequently, the practical relevance to real NN training remains speculative.
>
> We thank the reviewer for this insightful comment. We agree that distinguishing between the motivating context (neural networks) and the theoretical scope (sequence and RKHS models) is important.
> We have three specific points to address the concern regarding practical relevance and evolving eigenfunctions:
>
>
> 1. _Theoretical Proxy is Standard and Necessary_: We reference neural networks to motivate the problem of feature adaptivity, and we utilize stylized models to rigorously analyze this adaptivity. As noted in recent feature learning literature (e.g., Ba et al., 2022; Li & Lin, 2024; Zhang et al, 2024), these stylized models serve as standard theoretical proxies that allow for a closed-form derivation of feature learning dynamics. Our contribution is bridging this adaptive dynamics to minimax optimal rates via ESD, which has not been done before.
>
> 2. _ESD is Valid for Evolving Eigenfunctions_: You correctly note that genuine NNs involve evolving eigenfunctions. While our result on dynamics (Section 5) focuses on eigenvalues for analytical tractability, the definition and utility of ESD are not limited to fixed eigenfunctions. We explain this aspect using the pathwise ESD framework discussed in Appendix C.2 as follows:
> 	- **Concept**: For any time-dependent kernel path $(k_t)_{t \geq 0}$, the quantity $d^{\dagger}(\sigma^2 ; f^{*}, k_t)$ remains a well-defined population measure of complexity at each training time $t$, even when both eigenvalues and eigenfunctions evolve.
> 	- **Mechanism**: The ESD can explain why a learned kernel generalizes better: an adaptive learning algorithm aligns the kernel leading eigenfunctions with the signal, reduces the ESD, and thus moves the signal into classes with strictly smaller minimax excess risk. This reasoning remains valid regardless of whether that alignment was achieved by re-weighting eigenvalues (as derived in our theory) or rotating eigenfunctions (as in real NNs).
>
>
> 3. _Empirical Evidence in Evolving Regimes_:
> While the theoretical derivation of eigenfunction evolution is challenging, we highlight an experiment to address your concern about practical relevance. In Appendix C.2, we trained a 4-layer Deep Linear Network, where eigenfunctions do evolve. Figure 5 demonstrates that the ESD computed on the evolving kernel accurately tracks the decrease in true risk throughout training. This provides concrete evidence that the ESD framework captures the generalization benefits of feature learning in realistic settings where classical fixed-kernel assumptions break down.
>
> In the revision, we will make the following changes:
> 1. We will move the discussion of the pathwise ESD analysis and the Deep Linear Network experiment (Appendix C.2) into the main text. This directly answers the reader's question about applicability to evolving eigenfunctions.
> 2. We will revise the Introduction to explicitly state that while ESD applies generally (as validated empirically), our theory on the dynamics of ESD focuses on the sequence model.
> 3. We will update the Discussion to frame the theoretical derivation of eigenfunction dynamics as a promising future direction and state the distinction from the eigenvalue dynamics established in the current work.
>
> **References**
>
> [1] Ba J, Erdogdu MA, Suzuki T, Wang Z, Wu D, Yang G. High-dimensional asymptotics of feature learning: How one gradient step improves the representation. Advances in Neural Information Processing Systems. 2022 Dec 6;35:37932-46.
>
> [2] Li Y, Lin Q. Improving adaptivity via over-parameterization in sequence models. Advances in Neural Information Processing Systems. 2024 Dec 16;37:52438-80.
>
> [3] Zhang H, Lai J, Li Y, Lin Q, Liu JS. Towards a statistical understanding of neural networks: Beyond the neural tangent kernel theories. arXiv preprint arXiv:2412.18756.

---

> ### Author Response · Authors · 2025-11-21
> **Comparison to classical notions**
>
> > In addition, while the paper contrasts ESD with classical notions such as the effective dimension, this comparison is not entirely fair: traditional kernel learning theory (e.g., Caponnetto & De Vito, Zhang) already incorporates signal–kernel alignment through the RKHS norm of the target function or source condition constants, albeit indirectly. While the ESD formalism clarifies and extends this dependence, the importance of target-kernel alignment was already recognized as a critical aspect of learning in the past literature.
>
> We thank the reviewer for raising this point.
> We fully agree that source conditions ($\sum \lambda_i^{-s} \langle f^*, \phi_i \rangle^2 < \infty$) implicitly capture alignment (as we have cited in Section 2), and we certainly do not claim to be the first to identify alignment as critical for learning.
>
> Our contribution is not the concept of alignment, but a novel ESD framework that is more flexible to quantify alignment, especially in settings with learned kernels. We will further elaborate on this in the following response to your question.
>
>
> > Relation to Classical Source Conditions: Classical RKHS learning rates already depend on the target’s regularity (e.g., through the source condition constant or RKHS norm). Could the authors elaborate on how their framework fundamentally differs from these forms of target dependence? Is the ESD intended to replace such constants, or does it complement them?
>
> As we response to the last comment, the ESD is intended to complement classical source conditions, especially when the kernel is not static.
>
> We highlight the differences in three aspects:
>
> 1. _Recovering and refining classical rates_: Under the standard assumptions of polynomial eigenvalue decay $\lambda_j \asymp j^{-\gamma}$ and a Hölder source condition $\sum_j \lambda_j^{-s} \langle f^*,\psi_j\rangle^2 \le R_s$, classical results show that the minimax rate scales as $n^{-s\gamma/(1+s\gamma)}$. In our framework, the same assumptions imply that the span profile is bounded by a quota sequence $K_n \asymp n^{1/(1+s\gamma)}$, so the minimax rate $\sigma_0^2 K_n/n$ recovers the classical rate. Thus, ESD subsumes the usual theory when a single smoothness index and decay regime are valid. At the same time, the ESD framework allows more general spectra (we only require an ordering of eigenvalues, not a specific decay law) and more general quota sequences  (for example, $K_n \asymp (\log n)^{1/b}$ in Example 4.4). As a result, the analysis can capture "irregular" rates such as  $\sigma_0^2 (\log n)^{1/b}/n$ that are not naturally expressed by a fixed source exponent.
> 2. _Noise dependent complexity_: Source conditions are typically used to derive rates for a fixed kernel as $n \to \infty$, and do not depend on the noise level. In contrast, the ESD $d^{\dagger}(\sigma^2 ; f^{*}, \mathbf{k})$ is defined at a specific noise level $\sigma^2$ (in our regression reductions $\sigma^2 = \sigma_0^2/n$) and directly characterizes the minimax risk that remains meaningful in finite samples.
> 3. _Dynamic analysis for learned spectra_: This is the most important difference. Classical source conditions are formulated relative to a fixed kernel: they assume an eigenbasis and decay pattern that do not change during training. In kernel learning and feature learning, the eigenvalues and eigenfunctions evolve over time, and it is unclear how to track a smoothness parameter $s_t$ for the learned kernel along this trajectory. In contrast, the ESD $d^{\dagger}(\sigma^2 ; f^{*}, \mathbf{k}_t)$ remains well defined for each time $t$ and continues to quantify alignment relative to the current kernel, as we discuss in our previous comment "ESD is Valid for Evolving Eigenfunctions." This allows us to mathematically track the improved kernel-signal alignment during training, a dynamic that classical notions cannot easily describe.
>
> To clarify the novelty, we will make the following changes in the revision:
> 1. We will rewrite the Introduction to acknowledge the precedent of alignment in the literature and summarize the differences between ESD and the classical notions.
> 2. We will move the pathwise ESD analysis (currently in Appendix C.2) into the main text to demonstrate explicitly how ESD offers a tractable metric for evolving kernels where classical notions are difficult to apply.

---

> ### Author Response · Authors · 2025-11-21
> **Response to the related question**
>
> > Applicability to Neural Networks: The paper’s framing emphasizes relevance to neural network generalization, yet all formal results are derived in simplified sequence or RKHS models. Could the authors clarify what specific aspects of neural network training (beyond the stylized over-parameterized gradient flow) they expect the ESD framework to capture? For instance, do they anticipate ESD reductions under standard SGD dynamics in finite-width networks?
>
> We thank the reviewer for this thoughtful question. We have clarified in our response that all of our formal results are proved in stylized sequence, linear, and RKHS models, together with the OP-GF setting, and that we do not currently claim guarantees for finite-width networks. Here we clarify what aspects of neural network training we expect the ESD framework to capture, and what remains open.
>
> As explained in our response on ESD with evolving eigenfunctions, for any time dependent kernel path $\left(\mathbf{k}\_t\right)\_{t \geq 0}$ induced by a network during training, one can consider $d^{\dagger}\left(\sigma^2 ; f\_{\star}, \mathbf{k}\_t\right)$ at each time $t$. This remains a meaningful population quantity even when both eigenvalues and eigenfunctions evolve.
>
> Regarding the reviewer's specific question about standard SGD in finite-width networks, we do not currently have theoretical guarantees that ESD must decrease under such dynamics, and we will state this explicitly in the revised discussion. Our view is that the present work isolates and analyzes one concrete spectral mechanism in a tractable proxy model and provides a definition of ESD that can be measured empirically for trained networks. Establishing rigorous ESD monotonicity under SGD for finite-width neural networks is an important open problem that we hope this framework will help to formulate and investigate.
>
> Nonetheless, we provide empirical evidence in this direction by training a four-layer deep linear network in Appendix C.2, where the empirical feature covariance (and thus the eigenfunctions) changes over time. In this experiment we observe that the ESD computed from the evolving kernel decreases along training and closely tracks the decay of the prediction risk (Figure 5), suggesting that the alignment mechanism captured by ESD is relevant more broadly.
>
> > Estimating the ESD in Practice: Since the ESD is defined at the population level and depends on the unknown signal coefficients, it is unclear how it could be estimated from data. Do the authors foresee practical estimators or proxies for the ESD that could be applied in empirical studies?
>
> Thank you for this question. We view the ESD as a population level complexity measure that quantifies signal–kernel alignment, in the same way that the sparsity level plays this role in high dimensional regression: it determines the minimax rate, but it is typically unknown and estimating it can be at least as hard as the original prediction problem. From this perspective, our main goal in the present work is to understand minimax rates and algorithmic behavior given an ESD level, rather than to develop a fully data driven estimator of the ESD.
>
> That said, we do foresee several natural practical proxies. In sequence models, one can formulate penalized risk estimates at different truncation points and then define an estimated ESD as the truncation point that minimizes the risk estimate. For example, the penalized projection method in the spirit of Birgé and Massart (2001) provide a promising way to formulate such a risk estimate and select such a truncation level adaptively. A rigorous analysis of these procedures is beyond the scope of the current paper, so we choose to present the ESD as a population quantity here and leave a thorough study of data driven ESD estimation for future work.
>
> **Reference**
>
> Birgé, L. and Massart, P., 2001. Gaussian model selection. Journal of the European Mathematical Society, 3(3), pp.203-268.

---

### Official Review · Reviewer_1UBu · 2025-10-29

**Soundness:** 3
**Presentation:** 3
**Contribution:** 2
**Rating:** 4
**Confidence:** 3

**Summary:**

In the context of kernel regression, the paper introduces a new characteristic, Effective Span Dimension (ESD), discusses its properties, and uses it to analyze the spectral alignment between the target (represented by its spectral expansion coefficients) and the learning algorithm (represented by the kernel eigenvalues). First, the original regression problem is reduced to a spectral "sequence model" setting. Then, the ESD is introduced, motivated by the Principal Component algorithm. It is then argued that the ESD is relevant for any estimator (Theorem 3.3) and captures properties not captured by alternative alignment measures and effective dimensions. Then, a span profile reflecting the dependence of ESD on the noise level is introduced and discussed. A minimax rate result analogous to the earlier Theorem 3.3 is established. Then, Section 5 considers gradient flow in a deep linear network-type models and shows, under certain assumptions, a dynamic signal-spectrum alignment in terms of ESD. Finally, section 6 provides numerical demonstrations of spectral alignment and associated evolutions of span profiles.

**Strengths:**

The paper is carefully and thoughfully written. The mathematical level of the paper is high, and the explanations are good. The authors are very familiar with spectral algorithms and respective literature.

The paper has conceptual novelty in introducing a new charactiristic, Effective Span Dimension (ESD), applicable to general spectral distributions in contrast to the standard power-law source/capacity framework. This characteristic is relevant for target-spectrum alignment studies and might be useful in future research (though I'm not convinced in its importance; see Weaknesses below). In particular, the authors demonstrate, in terms of this characteristic, an alignment under gradient flow in some special setting (Theorem 5.2).

The paper contains both rigorous results and experiments. Several lengthy appendices provide additional discussion, numerical illustrations and the proofs. It appears that the authors have put a significant effort into this work.

**Weaknesses:**

I don't see why the Effective Span Dimension - the key concept proposed and studied in the paper - is as fundamental and important as the authors claim ("*the ESD is a fundamental measure for signal-spectrum alignment*").

1. ESD is introduced essentially as the critical value of the parameter $d$ in the PC method at which the squared bias matches the variance. A critical value of the regularization parameter can presumably also be considered for any other spectral method, e.g. previously mentioned Ridge and GF. For other methods the connection between the critical parameter and the risk may be not so explicit, but one can nevertheless define and study profiles analogous to the span profile of definiton 3.4. Each method would then have a different profile; there is no obvious connection between profiles of different methods.

2. The authors argue that ESD is fundamental because, while it is introduced using a specific estimator (PC), by Theorem 3.3 it quantifies the best performance of any estimator. However, this argument is misleading, since PC plays a preferred role in Theorem 3.3. Namely, the class $\mathcal F_{K,\lambda}^{(n)}$ is effectively defined there as the class of all problems on which the risk of the PC estimator is sufficiently low. The theorem then states that no other estimator can uniformly outperform PC on this class. But this is hardly surprising, since different estimators are efficient on different classes of problems. Likewise, I would expect analogs of theorem 3.3 to hold for other classes of problems, introduced using baseline estimators other than PC. In any case, Theorem 3.3 does not imply that the ESD is as relevant for other estimators as for PC.

3.  I don't see how ESD can be used to efficiently compare different spectral algorithms and clearly reveal their important theoretical properties. In the standard source/capacity framework one can, for example, identify a specific range of exponents where GF achieves the minimax rate, but Ridge fails to do that due to saturation (Math´e, 2004; Bauer et al., 2007; Pillaud-Vivien et al., 2018). While limited to the standard source/capacity setting, this result provides a concise, sharp and contrastive characterization of the two algorithms and so has a clear theoretical value. I don't see any comparably interesting results in the present paper. Theorems 3.3 and 4.3 basically say that PC cannot be uniformly improved on the problems where PC has an upper bounded risk. Theorem 5.2 does demonstrate a signal-spectrum alignment by GF in terms of ESD. However, I don't find this theorem very illuminating. It is established in a very special linear network setting and under some complicated assumptions. The effect is likely non-universal, and it's not very clear which properties of the algorithm and the problem are essential for it. The theorem only tells us something about the ESD, which is an auxiliary quantity, rather than the risk which is the primary object of interest. In particular, the theorem doesn't demonstrate acceleration of learning thanks to the spectral alignment.

**Questions:**

N/A

---

> ### Author Response · Authors · 2025-11-24
> **Response to Weaknesses 1&2**
>
> We thank you for your careful reading of our paper and for the constructive comments.
> We are glad that the reviewer finds the paper carefully written, mathematically solid, and conceptually novel. We appreciate the opportunity to clarify the nature of the Effective Span Dimension (ESD) and its role. Below, we address your concerns point-by-point.
>
>
>
> > "1. ESD is introduced essentially as the critical value of the parameter $d$ in the PC method ... Each method would then have a different profile ..."
>
>
> > "2. The authors argue that ESD is fundamental because, while it is introduced using a specific estimator (PC), ... Likewise, I would expect analogs of theorem 3.3 to hold for other classes of problems, introduced using baseline estimators other than PC. "
>
>
> We thank the reviewer for raising these points regarding the relationship between the Effective Span Dimension (ESD), the Principal Component (PC) method, and alternative estimators. We understand the concern that ESD might appear to be merely a tuning parameter for PC.
>
>
> We wish to clarify that ESD is a population-level measure of signal-kernel alignment that characterizes the information-theoretic difficulty of the problem, independent of the algorithm used to solve it. Our response is structured into three parts: (1) the logical independence of ESD from the PC algorithm, (2) the interpretation of the minimax result, and (3) the limitations of alternative "profiles" (e.g., Ridge) compared to ESD.
>
>
> First, we would like to clarify the logical order of our definitions and results. the definition of ESD depends only on the problem geometry, not the estimator. The measure is defined as:
> $$
> d^{\dagger}(\sigma^2 ; \theta, \lambda)=\min \\{k: \frac{1}{k} \sum\_{i>k} \theta\_{\pi\_i}^2 \leq \sigma^2\\},
> $$
> where $\{\pi\_i\}$ indices eigenvalues in decreasing order. This definition relies exclusively on the ordering of the spectrum, the signal coefficients $\theta$, and the noise level $\sigma^2$. It does not invoke any estimator.
> The associated class
> $$
> \mathcal F^{(n)}\_{K,\lambda} = \\{\theta: d^\dagger(\sigma^2;\theta,\lambda)\le K\\}$$
> is defined by the geometry of the signal relative to the kernel eigenbasis. This geometry exists regardless of whether PC is used to estimate the signal.
>
>
>
> Second, we would like to clarify the  interpretation of the minimax result regarding the ESD. The reviewer suggests that Theorem 3.3 implies ESD is only relevant for "PC-favorable" problems. We respectfully disagree. Theorem 3.3 establishes a minimax lower bound:
> $$\inf_{\hat{\theta}} \sup_{\theta \in \mathcal{F}_{K,\lambda}^{(n)}} \mathcal{R}(\hat{\theta}, \theta) \gtrsim K \sigma^2.$$
> This lower bound holds uniformly over all measurable estimators, not just PC. This proves that the quantity $K$ (and thus the ESD) characterizes the information-theoretic limit of the problem class. No other method can achieve a better rate on this class.
> PC serves merely as the "witness" (upper bound) to show that this fundamental limit is achievable.
> Thus, ESD quantifies the intrinsic difficulty of the estimation task, akin to how the sparsity level $s$ quantifies difficulty in high-dimensional regression regardless of whether one uses Ridge, the lasso, or the Dantzig selector.
>
>
> Third, we would like to point out the limitation of the reviewer's suggestion of using a profile derived from other estimators, such as the Ridge estimator. While possible, such a profile lacks the sharp characterizing power of ESD for two reasons:
>
> - **Suboptimality**: Ridge estimation is known to be minimax suboptimal for signals with high smoothness (saturation effect when smoothness $\alpha > 1$, see Zhang et al 2023). A complexity measure derived from the "Ridge profile" would encode this deficiency, failing to capture the true difficulty of smoother signals.
> - **Lack of Transparency**: The risk of Ridge depends on the entire eigenvalue sequence through the effective degrees of freedom, $df(\nu) = \sum \frac{\lambda_i}{\lambda_i + \nu}$. Unlike ESD, which offers a clear "truncation" interpretation, a Ridge-based profile obscures the precise boundary of learnability.
>
>
> Lastly, we acknowledge that our use of the word "fundamental" should be contextualized. We will revise the text to clarify that ESD is fundamental in the sense that it acts as the minimax-optimal dimension parameter for the class $\mathcal F^{(n)}_{K,\lambda}$, analogous to how the sparsity level controls the optimal rate in high-dimensional sparse regression. We do not claim the ESD is the only possible measure, but rather a concrete signal-kernel alignment metric that encompasses classical spectral decay frameworks while remaining valid to explain the performance of adaptive kernels where classical assumptions fail.
>
> **Reference**
>
> Li, Zhang, and Lin. "On the Saturation Effect of Kernel Ridge Regression." The Eleventh International Conference on Learning Representations. 2023.

---

> ### Author Response · Authors · 2025-11-24
> **Response to Weakness 3 (part 1)**
>
> > "3. I don't see how ESD can be used to efficiently compare different spectral algorithms ... In the standard source/capacity framework one can identify a specific range where GF achieves the minimax rate, but Ridge fails due to saturation… I don't see any comparably interesting results in the present paper. "
>
> We thank the reviewer for this comment, which highlights the distinction between classical fixed-kernel analysis and our alignment-based framework. We agree that classical theory provides valuable insights for fixed kernels. However, our work addresses a different fundamental question: **How do we characterize statistical complexity when the kernel itself is learned?**
>
> We wish to clarify the following key points regarding the utility and novelty of our results.
>
> **ESD Subsumes and Extends Classical Rates**
>
> The reviewer states that classical theory identifies specific minimax rates. We emphasize that our ESD framework strictly recovers these classical results while extending them to regimes where classical theory is silent.
> 	- Recovering Standard Rates: As detailed in Equation (9) and Appendix D.3, if eigenvalues follow a power law $\lambda\_j \asymp j^{-\gamma}$ and the signal satisfies a source condition with index $\alpha$ (i.e., $\sum \lambda\_i^{-\alpha} \langle f^*, \phi\_i \rangle^2 < \infty$), the ESD yields a quota $K\_n \asymp n^{\frac{1}{1+\alpha \gamma}}$. Our Theorem 4.3 then immediately recovers the optimal rate $n^{-\frac{\alpha \gamma}{1+\alpha \gamma}}$.
> 	- Handling Irregular Signals: Crucially, ESD applies where classical power-law/source assumptions fail. In Example 4.4, we analyze "irregular" signals where $K\_n \asymp (\log n)^{1/b}$. These cases do not fit standard source conditions, yet ESD provides a sharp minimax rate $\asymp \sigma\_0^2 (\log n)^{1/b} / n$. This demonstrates that ESD provides "interesting" theoretical characterizations for a broader class of problems than the standard framework.
>
>
> **Why ESD is Necessary for Learned Kernels**
>
> The reviewer mentions comparing Ridge vs. GF under fixed source conditions. We respectfully point out that such comparisons under source conditions are ill-defined in the context of feature learning, where the eigenfunctions $\psi\_j(t)$ evolve during training.
> - Under the classical framework, a signal $f^*$ might satisfy a source condition with smoothness index $\alpha_0$ relative to the kernel at initialization ($t=0$), but have a completely different  smoothness index $\alpha_t$ at time $t$. It is unclear how to track a time-dependent smoothness parameter $\alpha_t$ when the eigenfunctions $\psi_j(t)$ evolve over time.
> - In contrast, the ESD $d^\dagger(\sigma^2; f^*, \mathbf{k}\_t)$ remains a well-defined scalar summary of the geometry between the signal and the current kernel $\mathbf{k}\_t$. This allows us to mathematically track how the ESD decreases under kernel learning, which is a dynamic that the classical framework do not address.
>
>
> We hope this clarifies that the primary benefit of ESD is not to rank estimators, but to provide a complexity measure for adaptive kernels.

---

> ### Author Response · Authors · 2025-11-24
> **Response to Weakness 3 (part 2)**
>
> > "Theorem 5.2 does demonstrate a signal-spectrum alignment by GF in terms of ESD. However, I don't find this theorem very illuminating. It is established in a very special linear network setting and under some complicated assumptions. The effect is likely non-universal, and it's not very clear which properties of the algorithm and the problem are essential for it."
>
> We appreciate this feedback and we agree that Theorem 5.2 should be interpreted as a case study rather than a universal theory. However, we wish to clarify its specific role in our framework and how the reduction of ESD implies a concrete improvement in risk.
>
> **On the Scope of Theorem 5.2**
>
> We view Theorem 5.2 as a mechanistic prototype. While the over-parameterized diagonal model is a simplified setting, it allows us to rigorously track the interplay between optimization dynamics and statistical complexity.
>
> More concretely, Theorem 5.2 reveals that the gradient flow dynamics increase the effective eigenvalues much more rapidly along directions where the signal energy $|\theta^*\_j|^2$ is large and slower along directions where the signal is small.
> This dynamics explicitly reorders the spectrum to match the signal structure. As a result, the ESD $d^\dagger(t)$ decreases. This provides a rigorous existence proof that "feature learning" can be mathematically formalized as "ESD compression."
>
> We supplement this theoretical result with numerical experiments on deep linear networks (Appendix C.2), which exhibit the same ESD reduction trend. A general theory of ESD dynamics for realistic neural networks is an ambitious goal and is open to future studies. We see Theorem 5.2 as a first step in that direction, and we will revise the text to explicitly frame it as an "existence proof" of this phenomenon rather than a claim of universality.
>
>
>
> **Acceleration of Learning**
>
> The reviewer argues that ESD is merely an auxiliary quantity and that the theorem does not demonstrate acceleration. We respectfully disagree. Regarding Theorem 5.2 specifically, Li and Lin (2024) have already analyzed the risk of the OP-GF algorithm, but they did not provide a complexity measure to explain why the risk improves. Our work fills this gap by using ESD as the explanatory variable, showing that the risk improvement results from the explicit minimization of the ESD $d^\dagger$.
>
>
>
> Regarding the general phenomenon of "ESD compression," our minimax results (Theorems 3.3 and 4.3) imply that a reduction in ESD corresponds to statistical acceleration:
> 	- We established that the minimax risk scales as $\sigma^2 d^\dagger$. Therefore, if the learning dynamics reduce $d^\dagger$ from the class with quota $K\_{start}$ to the class with quota $K\_{end}$ (where $K\_{end} \ll K\_{start}$), the learner has effectively moved the problem from a "hard" complexity class to an "easy" one.
> 	- By reducing the effective dimension $d^\dagger$, the adaptive algorithm reduces the sample complexity required to attain a specific error tolerance.
>
> To make this connection concrete, we offer the following interpretation:
> Conceptually, one can view the training process as learning a kernel $\mathbf{k}\_T$ such that the ESD is minimized as $d^\dagger\_T$. If one were to freeze this learned kernel and apply a standard spectral estimator (like PC) on fresh data of the same sample size, the resulting risk would be bounded by $O(\sigma^2 d^\dagger\_T)$.
> Thus, a result like Theorem 5.2 would guarantee that the "potential for generalization" improves over time. We will add a remark after Theorem 5.2 to clarify this interpretation.
>
> ---
>
> We thank the reviewer again for these constructive comments. We believe that by clarifying the independent definition of ESD, its connection to minimax lower bounds, and its unique utility in tracking the complexity of learned kernels, we have addressed the core concerns regarding the significance of our framework.

---

### Official Review · Reviewer_MgNF · 2025-11-01

**Soundness:** 2
**Presentation:** 3
**Contribution:** 2
**Rating:** 6
**Confidence:** 3

**Summary:**

This paper introduces a novel, signal-aware complexity measure for kernel regression, the Effective Span Dimension (ESD). This measure, depends on the interplay between the true signal, the kernel's spectrum and the noise level. The authors first approximate the RKHS regression problem with a sequence model. They then analyze this sequence model and prove that their proposed ESD is a fundamental measure of the problem's difficulty by showing that the risk of an oracle Principal Component estimator is tightly bounded by the ESD and proving a matching minimax lower bound. Finally, the authors connect this framework to the algorithm of Li & Lin (2024), showing that throughout training, this adaptive process actively reduces the ESD.

**Strengths:**

The authors correctly identify the shortcomings of "signal-agnostic" measures which fail to capture the critical interplay between the signal and the kernel. This idea of the EDS is sound. Aside minor typos, the paper is well-written.

**Weaknesses:**

While the premise of the paper is sound, the execution is not entirely convincing. My biggest criticism is that the entire framework is built upon an "oracle" estimator (the PC estimator in the sequence model and the KPCE estimator in the RKHS model) that assumes access to unknowable, population-level quantities (the eigenvalues and eigenvectors of the integral operator). It makes it unclear if we can use the ESD to study the standard kernel ridge regression estimator or the kernel PC ones (the non oracle version of Eq. (22)). The paper characterizes the difficulty of an idealized problem that no practitioner ever gets to solve.

**Questions:**

Typos/notations:
- Eq. (3) isn't the equation with the covariance exact? Why did you use \approx?
- The use of psi_j for eigenfunctions and psi_nu for filter functions is confusing
- Eq (8) it should be theta and not theta^\star in the set
- The use of sigma^2 for sigma^2_0/n is confusing
- Example 4.2: where is L_k defined?
- On page 17 Gkj (d×d matrix) is referred to as the "empirical Gram matrix." In kernel literature, "Gram matrix" has a standard n×n meaning.

---

> ### Author Response · Authors · 2025-11-20
> **Response to Main Concern**
>
> We thank the reviewer for recognizing the value of our signal-aware complexity measure and for the positive comments on the writing. We first address the main concern about oracle estimators, and then respond to the specific questions and notation issues in the following post.
>
>
> > My biggest criticism is that the entire framework is built upon an "oracle" estimator ...
>
> We thank you for raising this important conceptual question. We wish to clarify that the Effective Span Dimension (ESD) is designed as a statistical complexity measure, rather than a tuning parameter for an algorithm. In this sense, it plays the same role as the sparsity level $s$ in high-dimensional regression or the smoothness index $\alpha$ in nonparametric regression.
>
> First, while we motivate ESD via oracle PC estimators, the definition of ESD itself is not tied to any specific estimator, just as the sparsity level is not tied exclusively to Lasso or best subset selection.
>
> Second, defining complexity via population-level quantities is standard practice in statistical learning theory. For example:
>
> - In sparse regression, the optimal rate depends on the sparsity $s$, which is defined through the (unknown) support of the coefficients.
>
> - In RKHS regression, the optimal rate depends on smoothness $\alpha$, which is defined through the (unknown) sequence of inner products between $f^*$ and the kernel eigenfunctions as well as the kernel eigenvalues.
>
> Even though $s$ and $\alpha$ are not directly available to the practitioner, they characterize the minimax risk. Our ESD is defined and used in the same way: Theorems 3.3 and 4.3 establish that no estimator (whether oracle PC, ridge regression, or empirical PC) can achieve a risk lower than the rate determined by the ESD. Thus, ESD characterizes the fundamental difficulty of the learning task itself, regardless of the method used.
>
>
> > It makes it unclear if we can use the ESD to study the standard kernel ridge regression estimator or the kernel PC ones...
>
> We appreciate this question about how ESD applies to specific practical estimators.
> We respectfully clarify that our aim is to identify a measure of the kernel-signal alignment and to establish the corresponding theoretical limit of generalization. Only then is it meaningful to discuss algorithms relative to this limit. The utility of ESD manifests in two ways:
>
> 1. **Establishing the Universal Baseline**: Since our minimax lower bounds apply to all estimators, ESD provides a universal baseline that every practical method must respect.
>
> 2. **Explaining Adaptive Mechanisms**: Crucially, ESD is not merely a static oracle quantity; it is a dynamic metric we use to explain feature learning. In Section 5, we explicitly use ESD to analyze Over-Parameterized Gradient Flow (OP-GF). This demonstrates that adaptive algorithms succeed by modifying the population geometry (the kernel's alignment with the signal) to actively reduce the ESD and thus lower the theoretical limit.
>
> Regarding empirical estimators, the fact that a specific empirical estimator might not attain the rate predicted by the ESD does not invalidate ESD as a measure. We illustrate this with two analogies:
>
> - In sparse regression, sparsity $s$ is the right complexity parameter, but ridge regression does not achieve the optimal rate for sparse models (Ref. [1]).
>
> - In RKHS regression, kernel ridge regression (KRR) is known to suffer from the saturation effect and does not attain the optimal rate implied by the source condition if the signal is very smooth (Ref. [2]).
>
> The saturation effect suggests that KRR cannot, in general, be described by a single ESD. This is a limitation of KRR, rather than a limitation of ESD as a complexity measure.
>
> For empirical kernel principal component estimators (eKPCE), a matching upper bound in terms of ESD is plausible, but proving it would require detailed control of spectral gaps and operator fluctuations. We have cited relevant works on eKPCE in Appendix G, and a full ESD-based analysis would build on these works. However, establishing such an upper bound is orthogonal to the main aim of the current paper. Our primary contribution is proposing ESD as a metric to quantify signal-kernel alignment and using it to mechanistically explain adaptivity (as demonstrated in Section 5). The technical derivation for eKPCE would not advance this central theme, so we regard it as a separate technical direction beyond the scope of the current work.
>
> **References**
>
> [1] Raskutti, Wainwright, & Yu. "Minimax rates of estimation for high-dimensional linear regression over $\ell_q$-balls." IEEE transactions on information theory (2011).
>
> [2] Li, Zhang, & Lin. "On the Saturation Effect of Kernel Ridge Regression." ICLR (2023).

---

> ### Author Response · Authors · 2025-11-20
> **Response to Questions**
>
> > Eq. (3) isn't the equation with the covariance exact? Why did you use \approx?
>
> Equation (3) represents the reduction of the random-design RKHS regression to a sequence model. This reduction involves approximations because $n$ is finite and the covariates $x_i$ are random, meaning the empirical basis functions are not perfectly orthogonal. As derived in Appendix B.1 (Page 17), there are two distinct approximations involved:
>
> 1. The Signal Approximation: We have $z_j = \theta_j^* + \Delta_j + \xi_j$, where $\Delta_j = (G_{jj}-1)\theta_j^* + \sum_{k \neq j} G_{kj}\theta_k^*$. Here, $G_{kj} := \frac{1}{n}\sum_{i=1}^n \psi_k(x_i)\psi_j(x_i)$ is the empirical feature correlation. While $\mathbb{E}[G_{kj}] = \delta_{kj}$, for finite $n$, $G_{jj} \approx 1$ and $G_{kj} \approx 0$, making $\Delta_j$ small but non-zero.
>
> 2. The Noise Approximation: Conditional on the covariates, the noise covariance is $\text{Cov}(\xi_j, \xi_k) = n^{-1}\sigma_0^2 G_{jk}$. Again, this is only exactly diagonal in the limit $n \to \infty$ (by the Law of Large Numbers) or in the specific case of a fixed orthonormal design.
>
> We hope this clarify the notation $\approx$ in Equation (3).
>
> > The use of psi_j for eigenfunctions and psi_nu for filter functions is confusing
>
> We agree. We will rename the filter functions to $g_\nu(\lambda)$ to clearly distinguish them from the eigenfunctions $\psi_j$.
>
> > Eq (8) it should be theta and not theta^\star in the set
>
> Thank you for catching this. We will correct this in the revision.
>
> > The use of sigma^2 for sigma^2_0/n is confusing
>
> We use the notation $\sigma^2$ to denote the noise level of the sequence model and use $\sigma_0^2$ to denote the observation noise variance in linear models and RKHS regression. As seen in Equation (3) as well as Appendices A.2 and B.1, it is clear that taking $\sigma^2$ to be $\sigma_0^2/n$ builds the connection between the sequence model and the regression models.
>
> To avoid confusion, we will explicitly denote the effective variance as $\sigma_{eff}^2=\sigma_0^2/n$ when we connect the regression models to the sequence model, while retaining the cleaner $\sigma^2$ notation for the theoretical analysis of the sequence model itself.
>
> > Example 4.2: where is L_k defined?
>
> Thank you for catching this typo. We will remove this notation.
>
> > On page 17 Gkj (d×d matrix) is referred to as the "empirical Gram matrix." In kernel literature, "Gram matrix" has a standard n×n meaning.
>
> Thank you for pointing this out. We agree that "Gram matrix" usually refers to the $n \times n$ matrix $K(x_i, x_j)$. Since $G$ is a $d\times d$ matrix, we will rename it to "Empirical Feature Correlation Matrix" to avoid confusion.

---

### Meta-Review · Area_Chair_cvdN · 2026-01-07

**Summary:**

This is a borderline paper which is well written and contains some novelty. The main criticism I see is that the complexity measure does not appear to be a fundamental quantity and could easily be replaced by other complexity measures; the authors try to cancel that criticism by arguing that the reviewer misunderstands the complexity measure as a tuning parameter of PC. But, in my reading, that is not what the reviewer says. Their argument is that you could come up with a variety of other complexity measures by using other algorithms as a guideline. In any case, I agree with this line of criticism and think that some stronger motivation for the complexity measure is required.

**Reviewer Concerns:**

The choice of complexity measure is somewhat arbitrary.

**Reviewer Scores:**

In my reading the rebuttal misses the point and I don't think that the scores would have changed.

---

### Decision · Program_Chairs · 2026-01-26

Reject